# Are Domain Generalization Benchmarks with Accuracy on the Line Misspecified?

**Olawale Salaudeen**[*]                                                       *olawale@mit.edu*
*Massachusetts Institute of Technology*

**Nicole Chiou**[†]                                                           *nicchiou@stanford.edu*
*Stanford University*

**Shiny Weng**[†]                                                            *shinyw@stanford.edu*
*Stanford University*

**Sanmi Koyejo**                                                            *sanmi@cs.stanford.edu*
*Stanford University*

**Reviewed on OpenReview:** *https://openreview.net/forum?id=fNywRyqPQo*

## Abstract

Spurious correlations—unstable statistical shortcuts a model can exploit—are expected to degrade performance out-of-distribution (OOD). However, across many popular OOD generalization benchmarks, vanilla empirical risk minimization (ERM) often achieves the highest OOD accuracy. Moreover, gains in in-distribution accuracy generally improve OOD accuracy—a phenomenon termed *accuracy on the line*—which contradicts the expected harm of spurious correlations. We show that these observations are an artifact of *misspecified OOD datasets* that do not include shifts in spurious correlations that harm OOD generalization; the setting they are meant to evaluate. Consequently, current practice evaluates "robustness" without truly stressing the spurious signals we seek to eliminate; our work pinpoints when that happens and how to fix it. **Contributions.** (i) We derive *necessary and sufficient conditions* for a distribution shift to reveal a model's reliance on spurious features; when these conditions hold, "accuracy on the line" disappears. (ii) We audit leading OOD datasets and find that most still display accuracy on the line, suggesting they are misspecified for evaluating robustness to spurious correlations. (iii) We catalog the few well-specified datasets and summarize generalizable design principles, such as identifying datasets of natural interventions such as a pandemic, to guide future well-specified benchmarks.

## 1 Introduction

Domain generalization aims to develop classifiers that generalize to new and potentially *worst-case* unobserved distributions (Arjovsky et al., 2019; Rosenfeld et al., 2020). Spurious correlations, also referred to as shortcuts, are coincidental statistical associations between features and labels in training data that fail to generalize beyond the training distribution, hindering domain generalization (Nagarajan et al., 2020; Geirhos et al., 2020; Makar et al., 2022). Thus, many algorithms for domain generalization have focused on learning classifiers that ignore these unreliable patterns—e.g., *invariance* or *feature disentanglement* methods (Arjovsky et al., 2019; Wang et al., 2019; Parascandolo et al., 2020; Salaudeen & Koyejo, 2022; Creager et al., 2021; Krueger et al., 2021; Ahuja et al., 2021; Shi et al., 2021; Zhou et al., 2022; Wang et al., 2022; Li et al., 2022; Salaudeen & Koyejo, 2024). Zhou et al. (2022); Wang et al. (2022) provide a comprehensive survey on domain

---

[*]Correspondence to olawale@mit.edu. Work was partly done while a PhD student at the University of Illinois at Urbana-Champaign and a visiting PhD student at Stanford University.
[†]Equal contribution; listed in alphabetical order.

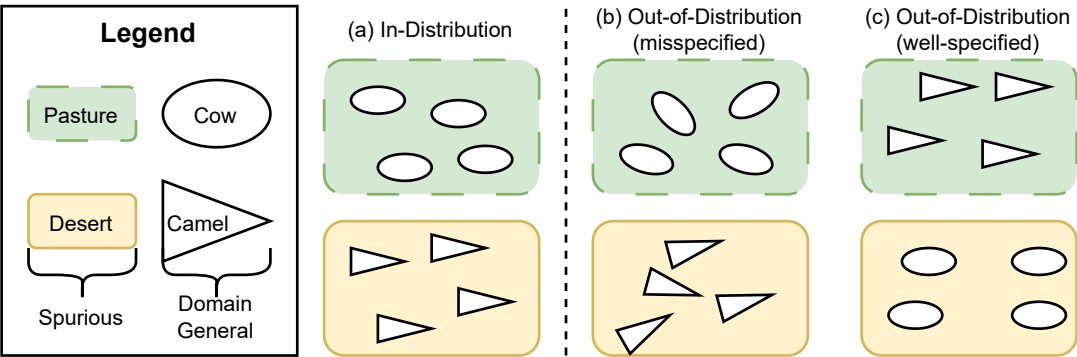

Figure 1: Simplified illustration of misspecified vs. well-specified domain generalization benchmarks. Consider a classification task of predicting cows vs. camels. In (a) (*in-distribution*), most cows appear on green pasture backgrounds and most camels on yellow desert backgrounds, so the background is spuriously correlated with the label. In (b) (*OOD, misspecified*), the dataset differs from (a) only in image rotation while preserving the pasture→cow and desert→camel association. A classifier that relies on background therefore still performs well. In (c) (*OOD, well-specified*), the background–label pairing is reversed (camels on pasture, cows on desert). A background-based classifier now misclassifies most examples, so models that depend on the spurious correlations underperform those that use only domain-general (animal) features.

generalization. Notably, domain generalization is distinct from domain adaptation, where one aims to adapt unstable correlations rather than remove them (Saenko et al., 2010; Long et al., 2018; Ganin et al., 2016; Wilson & Cook, 2020; Alabdulmohsin et al., 2023; Gupta et al., 2023; Tsai et al., 2024).

To ground this problem setting, there are many real-world settings where spurious correlations lead to adverse real-world behavior. For instance, when predicting medical diagnoses from chest X-rays, the relationship between physiological features and diagnoses is expected to be stable from site to site (domain-general) while the relationship between site-specific markings on X-rays and diagnoses is unstable (spurious). In practice, classifiers learn to rely on site-specific markings and fail out-of-distribution (Zech et al., 2018). Furthermore, clinical classifiers trained on data from one hospital may be less effective in a different hospital with distinct patient demographics, equipment, or clinical practices, inducing spurious correlations (Yang et al., 2024b). Similarly, an animal detection classifier trained on data from one environment and sensor type may be less effective in classifying animals in images captured under a different environment and sensor configuration-induced spurious correlations (Beery et al., 2018; Xiao et al., 2020).

Consequently, the standard approach to benchmarking domain generalization algorithms involves training classifiers on a set of in-distribution data and subsequently evaluating their performance on distinct out-of-distribution datasets (Gulrajani & Lopez-Paz, 2020; Koh et al., 2021). The underlying hypothesis is that classifiers that rely on spurious correlations will exhibit poorer transfer performance when evaluated on new domains, whereas classifiers that avoid these correlations will generalize better. Consequently, effective algorithms for mitigating or eliminating spurious correlations during training should give classifiers with improved transferability and higher out-of-distribution accuracy. Importantly, standard empirical risk minimizers (ERM) leverage spurious correlations if doing so minimizes empirical risk (Xiao et al., 2020; Vapnik, 1991; 1999). In some settings, they may even prefer the spurious correlations (Jain et al., 2024; Yang et al., 2024a).

Still, empirical evidence suggests that standard empirical risk minimizers often achieve the highest out-of-distribution (OOD) accuracy on widely used domain generalization benchmarks (Gulrajani & Lopez-Paz, 2020; Yao et al., 2022; Gagnon-Audet et al., 2022; Yang et al., 2023). Additionally, Taori et al. (2020); Miller et al. (2020; 2021); Wenzel et al. (2022); Baek et al. (2022); Saxena et al. (2024); Nastl & Hardt (2024) demonstrate a strong correlation between in- and out-of-distribution accuracy across several state-of-the-art domain generalization benchmarks, suggesting that better in-distribution (ID) performance generally predicts better

out-of-distribution performance. On popular tabular distribution shift datasets (Gardner et al., 2023), Nastl & Hardt (2024) further show that a classifier that uses all available, and potentially spurious, features generally results in better OOD performance than a classifier that uses a select subset, assumed to be causal and stable (Peters et al., 2016; Heinze-Deml et al., 2018; Schölkopf et al., 2012). These observations seemingly contradict the thesis that there can be spurious statistical associations in training data that can improve ID performance but worsen OOD performance (Geirhos et al., 2020; Makar et al., 2022).

These findings raise a critical question: *Does a better in-distribution classifier imply a better out-of-distribution classifier, challenging the necessity of targeted algorithms for domain generalization?*

We propose a reconciling perspective and provide supporting evidence. Drawing on the concept of underspecification in modern machine learning pipelines (D'Amour et al., 2022), we propose that the in-distribution empirical risk minimizer's apparent superiority out-of-distribution stems from the misspecification of popular domain generalization benchmarks for assessing robustness to spurious correlations.

The core of this work studies a benchmark's ability to distinguish between two types of classifiers: one that ignores spurious correlations (domain-general or invariant classifiers) and another that leverages all available correlations (including spurious) that maximize in-distribution accuracy (domain-specific classifiers), Figure 1. Specifically, we consider the setting where correlations between features and labels exist in training data and a subset (spurious) shift at test time. We investigate the types of shifts under which classifiers that rely on these spurious correlations generalize worse out-of-distribution than classifiers that only use stable (domain-general) correlations. We call these types of shifts *well-specified*. We propose that when the best in-distribution classifier is also the best out-of-distribution classifier on a benchmark, this benchmark does not represent the types of settings domain generalization is concerned with. Thus, our contributions are as follows:

## 1.1 Our Contributions

- We show that, with high probability, misaligned spurious correlation pre and post-distribution shift is necessary and sufficient for well-specified domain generalization benchmarks, Theorem 1-2.

- We demonstrate that these well-specified benchmarks and those with *accuracy on the line* are provably at odds, Theorem 3. Thus, *accuracy on the line is a test and sign of misspecified benchmarks*.

- With over 40 total ID/OOD[*] data splits across state-of-the-art benchmarks, we show that many state-of-the-art benchmarks exhibit accuracy on the line and may be misspecified for domain generalization. We also identify well-specified splits that the field should prioritize. Code[†]. We also provide a visualization tool to study future datasets: Link.

- We outline design principles that give well-specified ID/OOD splits for future curation of domain generalization benchmarks.

- Additionally, our findings have implications for related tasks such as benchmarking algorithms for algorithmic fairness, causal representation learning, etc., in the context of both predictive and generative models.

## 2 Theoretical Analysis: (Mis)specification of Domain Generalization Benchmarks

### 2.1 Preliminaries

Following previous work (Wang et al., 2019; Rosenfeld et al., 2020; Ahuja et al., 2021; Salaudeen & Koyejo, 2024), we define domain-general (dg) features $\mathcal{Z}_{\mathrm{dg}} \subseteq \mathbb{R}^k$, where the optimal classifier that only uses these domain-general features is desired. Conversely, spurious features (spu or domain-specific) $\mathcal{Z}_{\mathrm{spu}} \subseteq \mathbb{R}^l$ contain additional domain-specific information that improves the prediction task in-distribution, but their use can

---

[*]In many domain generalization benchmarks, a set of domains is available. This set is typically split into two disjoint subsets, where one is for training (in-distribution/ID), and the other is for testing (out-of-distribution/OOD).

[†]`https://github.com/olawalesalaudeen/misspecified_DG_benchmarks`.

degrade performance out-of-distribution. The observed features $\mathcal{X} \subseteq \mathbb{R}^d$ are a concatenation of $\mathcal{Z}_{\text{dg}}$ and $\mathcal{Z}_{\text{spu}}$, where $d = k + l$. We also define $\mathcal{Y} = \{\pm 1\}$. $P$'s represent probability distributions over $X, Y$. $\mathcal{E}$, where $|\mathcal{E}| > 1$, denotes the set of distributions of interest. $P_i \in \mathcal{E}$ implies that marginals without $Z_{\text{spu}}$ are preserved across $P_i$'s.

Then, we consider classifiers $f \in \mathcal{F} : \mathcal{X} \mapsto \mathcal{Y}$ of the form $f(X) = X\top w = Z_{\text{dg}}^\top w_{\text{dg}} + Z_{\text{spu}}^\top w_{\text{spu}}$ where $w_{\text{dg}} \in \mathbb{R}^k$ and $w_{\text{spu}} \in \mathbb{R}^l$. We note empirical findings support our parameterization of $Z_{\text{dg}}$ and $Z_{\text{spu}}$ from *non-linear transformation* when a final linear mapping is applied, e.g., kernel regressors or common deep neural networks. Rosenfeld et al. (2022a) demonstrates that improving out-of-distribution performance can be done with new linear classifiers on learned (non-linear) representations. Additional work shows that last-layer retraining improves robustness to spurious correlations (Kirichenko et al., 2022; LaBonte et al., 2023). These findings suggest that the non-spurious and spurious representations learned by deep models are often linearly separable, suggesting that our assumption is reasonable. Importantly, we allow for a nonlinear shift in spurious correlation.

Additionally, within $\mathcal{F}$, $\mathcal{F}_{\text{dg}} \subset \mathcal{F}$ comprises functions $f_{\text{dg}}$ that only use domain-general features $f_{\text{dg}}(X) = f_{\text{dg}}([Z_{\text{dg}}; 0])$. Also, denote $f_X \in \mathcal{F} \backslash \mathcal{F}_{\text{dg}} := \mathcal{F}_X$, where $f_X$ uses both domain-general and domain-specific features. The function $\ell(\cdot, \cdot) \mapsto \mathbb{R}$ denotes a loss function, and $\mathcal{R}^e(f) = \mathbb{E}_{P_e}[\ell(Y, f(X))]$ defines the expected loss for a function $f \in \mathcal{F}$. Additionally, we define the accuracy of $f \in \mathcal{F}$ on distribution $P$ as

$$\text{acc}_P(f) = \mathbb{E}_{(X,Y) \sim P}\big[\mathbf{1}(f(X) \cdot Y > 0)\big] = \Pr\big(f(X) \cdot Y > 0\big), \tag{1}$$

where $\mathbf{1}(\cdot)$ is the indicator function.

Next, we enumerate some assumptions and definitions for our theoretical analysis.

We first formally define spurious features and domain-general features (Definitions 1-2). The relationship between spurious features and the label we want to predict is allowed to change across domains and can negatively impact out-of-distribution performance, while the relationship between domain-general features and the label is stable across domains.

For example, consider the canonical example of classifying cows and camels. Here, domain general features are animal features, while domain-specific (spurious) features are backgrounds. Predictions derived for features of the cow are generally expected to be stable, while predicting cow vs. camel based on pasture vs. desert may be accurate for some data distributions and catastrophically inaccurate in others. Another example is domain-general physiological features rather than spurious hospital-specific markings in clinical chest X-rays diagnostic tasks (Zech et al., 2018).

**Definition 1** (Domain-General Features $Z_{\text{dg}}$)**.** *For all $P_i$, $P_j \in \mathcal{E}$,*

$$\mathbb{E}_{P_i}[Y \mid Z_{dg}] = \mathbb{E}_{P_j}[Y \mid Z_{dg}]. \tag{2}$$

**Definition 2** (Spurious Features $Z_{\text{spu}}$)**.** *There exists $P_i$, $P_j \in \mathcal{E}$ such that,*

$$\mathbb{E}_{P_i}[Y \mid Z_{spu}] \neq \mathbb{E}_{P_j}[Y \mid Z_{spu}] \quad and \quad \mathbb{E}_{P_i}[Y \mid Z_{dg}, Z_{spu}] \neq \mathbb{E}_{P_j}[Y \mid Z_{dg}, Z_{spu}]. \tag{3}$$

We assume both types of features are informative about labels (Assumption 1) and are not redundant (Assumption 2). Clearly, if Assumption 1 does not hold, then the learning problem is trivial; features are uncorrelated with labels. When Assumption 2 does not hold, spurious features are redundant and have no unique information about the labels. Ahuja et al. (2021) study this setting (*Fully Informative Invariant Features (FIIF)*; we use 'domain-general' instead of 'invariant') and give conditions under which classifiers using spurious correlations can achieve equal OOD accuracy as the optimal invariant classifier. Hence, we focus on the partially informative domain-general features setting, i.e., under Assumption 2. For example, we assume that, in the training domain, there are examples that a classifier is more likely to get correct if it uses spurious features.

For instance, there may be occluded animals in their natural habitat included in the training set, so using background improves the prediction accuracy on the training set (Beery et al., 2018). Similarly, the physiological features of a disease, like COVID-19, may not be fully represented in some chest X-rays, but

hospital-specific markings correlated with the diagnosis of COVID-19 serve as a predictive shortcut for the label in the training set. We assume both types of features are correlated with and contain unique information about the label.

**Assumption 1** (Informative Domain-General and Domain-Specific Features). *For all observed training distributions $P$,*

$$\mathbb{E}_P[Y \mid Z_{dg}] \neq \mathbb{E}_P[Y] \ and \ \mathbb{E}_P[Y \mid Z_{spu}] \neq \mathbb{E}_P[Y]. \tag{4}$$

**Assumption 2** (Non-Redundant Features).

$$Z_{spu} \not\perp\!\!\!\perp Y \mid Z_{dg} \ and \ Z_{dg} \not\perp\!\!\!\perp Y \mid Z_{spu}. \tag{5}$$

*Also referred to as partially informative domain-general features.*

Since we consider any feature whose inclusion decreases worst-case performance on the set of distributions of interest $\mathcal{E}$ as spurious, the definition of spurious is strongly tied to the $\mathcal{E}$ 'worst-case' is with respect to, such as the set of hospitals one expects a classifier to have to operate in. What is considered spurious for $\mathcal{E}' \neq \mathcal{E}$ may differ, even if $\mathcal{E}' \subset \mathcal{E}$ or $\mathcal{E} \subset \mathcal{E}'$. Notably, this observation implies that domain generalization cannot practically be divorced from domain expertise in defining $\mathcal{E}$. Clearly, too narrow a $\mathcal{E}$ decreases expected robustness (potentially catastrophically), and too broad a $\mathcal{E}$ may excessively and unnecessarily decrease overall utility (Shen et al., 2024).

We define two types of classifiers: (i) the optimal domain-general classifier, which depends on $\mathcal{E}$ (Definition 3) and (ii) the optimal domain-specific classifier for a given $P \in \mathcal{E}$ (Definition 4).

**Definition 3** (Optimal Domain General Classifier $f_{dg}^{\mathcal{E}}$). *Given a set of distributions of interest $\mathcal{E} = \{P_i(X, Y) : i = 1, \ldots\}$,*

$$f_{dg}^{\mathcal{E}} = \underset{f \in \mathcal{F}_{dg}}{\operatorname{argmax}} \min_{P_i \in \mathcal{E}} acc_{P_i}(f). \tag{6}$$

*By construction, $f_{dg}^{\mathcal{E}} \in \mathcal{F}_{dg}$ does not use spurious features.*

**Definition 4** (Optimal Domain Specific Classifier $f_X^P$). *Given a distribution $P$ and*

$$f_X^P = \underset{f \in \mathcal{F}}{\operatorname{argmax}} \ acc_P(f). \tag{7}$$

*By construction, $f_X^P \in \mathcal{F}_X$ uses spurious features (Lemma 1).*

Our first result, Lemma 1, shows that, given informative and non-redundant features (Assumptions 1-2), for any distribution $P \in \mathcal{E}$, the optimal $\mathcal{E}$-domain-general and $P$-domain-specific classifier are different, and the optimal domain-specific classifier achieves a lower (strongly convex) loss in-distribution than the optimal domain-general classifier. This result contextualizes the rest of our results in that optimal domain-specific classifiers use all non-redundant features that improve in-distribution performance.

**Lemma 1** (Domain-General and Domain-Specific In-Distribution Error Gap). *Assume non-redundant/non-trivial features, partially informative domain-general features (Assumptions 1-2), and strongly convex loss $\ell$.*

$$\min_{f \in \mathcal{F}} \mathcal{R}^e(f) < \min_{f \in \mathcal{F}_{dg}} \mathcal{R}^e(f), \tag{8}$$

*where $\mathcal{F} : \mathcal{X} \to \mathbb{R}$ where $f(x) = w^\top x = w_{dg}^\top z_{dg} + w_{spu}^\top z_{spu}$, $f \in \mathcal{F}$. For $f \in \mathcal{F}_{dg}$, $f(x) = w^\top x = w_{dg}^\top z_{dg}$. The proof of Lemma 1 is provided in Appendix A.1.*

We typically train the sort of classifiers we study in this work with cross-entropy loss, which is generally not strongly convex. However, with common regularizers, like weight decay, the objective can become strongly convex (Shalev-Shwartz & Ben-David, 2014). Given that the in-distribution risk minimizer and the domain-general classifier differ, we now show that, given training and test distributions $P_{\text{ID}} \neq P_{\text{OOD}} \in \mathcal{E}$, respectively, the domain-general classifier $f_{dg}^{\mathcal{E}}$ may also not achieve a higher OOD accuracy than the in-distribution risk minimizer $f_X^{P_{\text{ID}}}$ on $P_{\text{OOD}}$.

Next, we define *well-specified benchmarks* (Definition 5). Consider the running example of predicting cows and camels. Suppose we are given a set of training distributions and an OOD distribution. Then we are given two classifiers: a domain-general classifier that only uses animal features for prediction and a domain-specific classifier that also uses background information—both classifiers are trained on the training distributions. We consider this a well-specified benchmark if and only if the domain general classifier achieves a higher accuracy on the OOD dataset. Effectively, well-specified benchmarks penalize classifiers for using spurious correlations; hence, they can identify algorithms that give domain-general classifiers.

**Definition 5** (Well-Specified Domain Generalization Benchmark). *Two ID/OOD splits, $P_{ID}, P_{OOD} \in \mathcal{E}$, are 'well-specified' if and only if*

$$acc_{P_{OOD}}\big(f_X^{P_{ID}}\big) < acc_{P_{OOD}}\big(f_X^{\mathcal{E}}\big), \tag{9}$$

*where $f_{dg}^{\mathcal{E}}$, $f_X^{P_{ID}}$ are from Definitions 3 and 4, respectively—note that this definition is with respect to accuracy.*

**Analysis Setting.** For the rest of this work, we consider sub-Gaussian $Z_{\mathrm{spu}}^{\mathrm{ID}}$ with mean $\mu_{\mathrm{spu}}$, covariance $\Sigma_{\mathrm{spu}}$, and sub-Gaussian parameter $\kappa$—importantly, *our results also apply more generally to other classes of random variables (Remark 1)*. We define an $L_\phi$-Lipschitz function which parametrizes the distribution shift w.r.t. $Z_{\mathrm{spu}}$; $\phi : \mathbb{R}^{l \times l} \to \mathbb{R}^{l \times l}$ such that $Z_{\mathrm{spu}}^{\mathrm{OOD}} = \phi(Z_{\mathrm{spu}}^{\mathrm{ID}})$ and $\mathbb{E}[Z_{\mathrm{spu}}^{\mathrm{OOD}}] = M \mathbb{E}[Z_{\mathrm{spu}}^{\mathrm{ID}}] = M\mu_{\mathrm{spu}}$ where $M \in \mathbb{R}^{l \times l}$ and $\Sigma_\phi$ is the covariance of $Z_{\mathrm{spu}}^{\mathrm{OOD}}$.

*Our overall goal is to identify shifts where achieving higher transfer accuracy is informative about the reliance of classifiers on spurious correlations.* Identifying the class of such shifts is necessary as they describe ID/OOD splits for domain-generalization benchmarks where achieving the higher OOD accuracy meaningfully maps to learning domain-general classifiers, i.e., the shift is well-specified for the task. Without such well-specified shifts, we would incorrectly assume that a domain-general classifier is ineffective because it achieves a lower accuracy out-of-distribution than a different empirical risk minimizer. Thus, our next result shows that when shifts ($\phi$) result in a sufficient misalignment between spurious correlations before and after the shift, the domain-general classifier achieves a higher out-of-distribution accuracy than a domain-specific classifier.

**Remark 1.** *Our results consider sub-Gaussian spurious features. However, our results hold for other classes of random variables, e.g., sub-exponential or other random variables in Orlicz spaces (Krasnoselskii, 1960). In these cases, our proofs remain largely unchanged, with only the constant factors adjusted to account for the different concentration properties of the spurious features.*

## 2.2 Main Results

**Overview.** We formalize when an out-of-distribution (OOD) benchmark is well-specified for evaluating robustness to spurious correlations. Theorems 1-2 give sufficient, and under symmetry necessary, conditions for well-specification: the spurious–feature–label correlation must be sufficiently misaligned between in- and out-of-distribution splits, a property we call *spurious correlation reversal*. Theorem 3 proves that a strong positive correlation between in- and out-of-distribution accuracies ("accuracy on the line") is essentially incompatible with spuriously correlated correlation reversal. Specifically, the probability of observing a well-specified benchmark with accuracy on the line is zero. Together, these results provide a test of the absence of accuracy on the line to identify benchmarks that meaningfully penalize reliance on spurious correlations.

Our first results give sufficient conditions for well-specified domain-generalization splits. Theorem 1 demonstrates that a domain generalization split is well-specified if (i) the OOD spurious correlation is misaligned with the ID spurious correlation and (ii) the variance of spurious features is sufficiently controlled not to undo the effect of misalignment. For symmetrically distributed features, such as Gaussians, Theorem 2 gives similar conditions that are both necessary and sufficient.

**Theorem 1** (Sufficient Conditions for Well-Specified Domain Generalization Splits). *Assume $Z_{spu}^{ID}$ is sub-Gaussian with mean $\mu_{spu}$, covariance $\Sigma_{spu}$, and parameter $\kappa$. Define an $L_\phi$–Lipschitz nonlinear transformation*

$$\phi : \mathbb{R}^l \to \mathbb{R}^l, \quad and \ let \quad Z_{spu}^{OOD} = \phi(Z_{spu}^{ID}).$$

*Assume further that*

$$\mathbb{E}[Z_{spu}^{OOD}] = \mathbb{E}[\phi(Z_{spu}^{ID})] = M\,\mu_{spu},$$

*for a matrix $M \in \mathbb{R}^{l \times l}$. In-distribution, $Z_{spu}^{ID} \sim P_{ID}$ and out-of-distribution, $Z_{spu}^{OOD} \sim P_{OOD}$. Additionally, denote $w_{spu}$ the contribution of $Z_{spu}^{ID}$ to the optimal $P_{ID}$ classifier $f_X^{P_{ID}}$. Then, for any $\delta \in (0, 1)$, if*

$$w_{spu}^{\top}(M\,\mu_{spu}) + \sqrt{2}\,L_\phi\,\kappa\,\|w_{spu}\|_2\sqrt{\log(1/\delta)} < 0,$$

*(with the understanding that under the Lipschitz assumption, the sub-Gaussian property carries over with parameter $L_\phi\,\kappa$), then with probability at least $1 - \delta$ over $Z_{spu}^{OOD}$, we have*

$$acc_{P_{OOD}}(f_X^{P_{ID}}) < acc_{P_{OOD}}(f_{dg}^{\mathcal{E}}),$$

*where $f_{dg}^{\mathcal{E}}$ and $f_X^{P_{ID}}$ are the optimal domain–general and domain–specific predictions (Definitions 3–4). These conditions give a well-specified shift, Definition 5. Proof provided in Appendix A.2.*

We assume the distribution of spurious features is symmetric; a similar form of Theorem 1's results are both necessary and sufficient, Theorem 2.

**Theorem 2** (Necesary and Sufficient Conditions for Well-Specified Domain Generalization Splits under Distributional Symmetry)**.** *Assume $Z_{spu}^{ID}$ is a random variable with mean $\mu_{spu}$, covariance $\Sigma_{spu}$, and $w_{dg}^{\top}Z_{dg}^{OOD}$ is symmetric about its mean. Define an $L_\phi$–Lipschitz nonlinear transformation*

$$\phi : \mathbb{R}^l \to \mathbb{R}^l, \quad \text{and let} \quad Z_{spu}^{OOD} = \phi(Z_{spu}^{ID}).$$

*Assume further that*

$$\mathbb{E}[Z_{spu}^{OOD}] = \mathbb{E}[\phi(Z_{spu}^{ID})] = M\,\mu_{spu}, \qquad \mathrm{Cov}(Z_{spu}^{OOD}) = \Sigma_\phi, \quad and \quad Z_{dg} \perp\!\!\!\perp Z_{spu} \mid Y$$

*for some matrix $M \in \mathbb{R}^{l \times l}$, and that the scalar and $w_{spu}^{\top}Z_{spu}^{OOD}$ is symmetric about its mean. Then,*

$$acc_{P_{OOD}}(f_X^{P_{ID}}) < acc_{P_{OOD}}(f_{dg}^{\mathcal{E}}) \iff \frac{w_{dg}^{\top}\mu_{dg} + w_{spu}^{\top}M\mu_{spu}}{\sqrt{w_{dg}^{\top}\Sigma_{dg}w_{dg} + w_{spu}^{\top}\Sigma_\phi w_{spu}}} < \frac{w_{dg}^{\top}\mu_{dg}}{\sqrt{w_{dg}^{\top}\Sigma_{dg}w_{dg}}}. \tag{10}$$

*Note that the RHS of Equation 10 implies that either:*

- Spurious Correlation Reversal: $w_{spu}^{\top}M\mu_{spu} < 0$ *(related to sufficiency condition without the symmetry assumption in Theorem 1), or*

- *Small Signal-to-Noise Ratio (SNR): sufficiently large variance for $w_{spu}^{\top}M\mu_{spu} > 0$.*

*These conditions are necessary and sufficient for a well-specified shift, Definition 5. Proof provided in Appendix A.3.*

As an example, consider the classic examples of classifiers using background, pasture, or desert to predict cows and camels, respectively. Our results show that the correlation between background and label has to lead to sufficient disagreement (reversed) between training and test distributions such that their use harms OOD (test) predictions. Without this feature, classifiers without spurious correlations are not expected to transfer better than classifiers with spurious correlations.

Importantly, we are concerned with benchmarking classifiers designed not to rely on spurious correlations for prediction, even if they are useful and potentially reliable classifiers *most of the time*. Figure 2 verifies these conditions empirically with simulation experiments. Additionally, semi-synthetic examples with variants of the ColoredMNIST dataset is provided in Appendix B.1 (Arjovsky et al., 2019).

**Remark 2** (Multisource Domain-Generalization.)**.** *We often have access to a set of distributions for evaluation, and the norm is to evaluate leave-one-domain-out ID and OOD splits (Gulrajani & Lopez-Paz, 2020; Koh et al., 2021). In the context of Definition 5, $P_{ID}$ represents the mixture of ID distributions, and the test split is $P_{OOD}$ (which can also be a mixture). Importantly, a finite mixture of sub-Gaussians is also sub-Gaussian (Wainwright, 2019), so our results directly apply without loss of generality. Notably, we focus on distinct ID/OOD splits. A set of domains may give many splits, yet only a subset of the splits, if any, may be well-specified.*

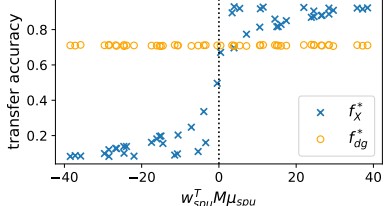 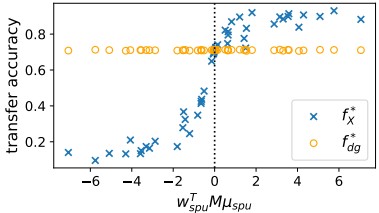 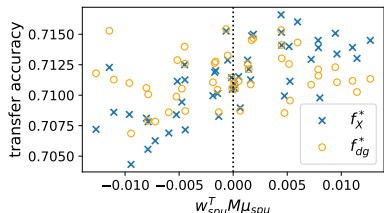

(a) **All random variables are Gaussian.** When Theorem 1-2 conditions are satisfied OOD (x-axis: $\mathrm{w}_{\mathrm{spu}}^\top M\mu_{\mathrm{spu}}$ + c < 0; $c > 0$), the domain general classifiers outperform the domain specific classifiers OOD. This result verifies that there needs to be sufficient misalignment between in- and out-of-distribution spurious correlations for the domain-general features to outperform the domain-specific classifiers in OOD accuracy.

(b) $\mathbf{Z}_{\mathrm{spu}}^{\mathrm{ID}}$ **is defined to be a mixture of 4 Gaussian distributions (sub-Gaussian) such that the test distributions are not a mixture of the same 4 Gaussians.** The same conclusions in Figure 2a hold for sub-Gaussian random variables when the test distribution is an extrapolation.

(c) Rosenfeld et al. (2022b) studies domain interpolation, where the target domain is a mixture of the training domain; a variant of ERM is provably worst-shift optimal. Spurious features $(\mathbf{Z}_{\mathrm{spu}}^{\mathrm{ID}})$ **are defined to be a mixture of 4 Gaussian distributions such that the test distributions are a different mixture of the 4 Gaussians.** Overall, there is minimal difference in OOD accuracy between domain-general and domain-specific classifiers in this setting (domain interpolation).

Figure 2: In our simulation, domain-general classifiers ($f_{\mathrm{dg}}$) are trained on only domain general features, while domain-specific classifiers are trained on both domain general and domain-specific features from the same distribution. Features are sub-Gaussian. We evaluate these classifiers on 50 test distributions generated by randomly sampling the distribution shift ($M$) such that all marginals without spurious features are the same ID and OOD in the test distribution, but for spurious features: $Z_{\mathrm{spu}}^{\mathrm{OOD}} = MZ_{\mathrm{spu}}^{\mathrm{ID}}$ and, thus, the spurious correlation shifts from $\mu_{\mathrm{spu}}$ to $M\mu_{\mathrm{spu}}$. Details on the experiments can be found in Appendix B.

So far, we have shown that the learned domain-general and domain-specific classifiers on a given training distribution differ, and the domain-general classifier achieves higher OOD accuracy (well-specified) when spurious correlations ID and OOD are sufficiently misaligned. For symmetrically distributed spurious features, such misalignment is necessary and sufficient for the evaluation goal. Next, suppose we observe such misalignment and the split is well-specified. Then, we evaluate a set of diverse classifiers on held-out ID and OOD test examples. We should observe a weak correlation or a strong negative correlation between the classifiers' ID and OOD accuracy, i.e., no positive *accuracy on the line. When we observe accuracy on the line, with a high probability, the ID/OOD split is misspecified, Theorem 3.*

### 2.2.1 Tradeoffs between Well-Specification and Accuracy on the Line

First, we formally define *accuracy on the line*, the correlation strength between in- and out-of-distribution accuracy, Definition 6.

**Definition 6** (Accuracy on the Line; Miller et al. (2021))**.** *Define $a \in \mathbb{R}$, $\epsilon \geq 0$, and $\Phi^{-1}$ as the inverse Gaussian cumulative density function. The* correlation property *is defined as*

$$\left| \Phi^{-1} \left( acc_{P_{ID}}(f) \right) - a \cdot \Phi^{-1} \left( acc_{P_{OOD}}(f) \right) \right| \leq \epsilon \, \forall \, f, \tag{11}$$

*where $f$'s are distinct classifiers.*

If there exists an $a$ such that $\epsilon = 0$, then there is a perfect correlation between ID and OOD accuracy. As $\epsilon$ grows, the strength of the correlation decreases. If $a > 0$, then the correlation is positive, and if $a < 0$, the correlation is negative. We will call the setting where $a > 0$ positive accuracy on the line and $a < 0$ accuracy on the inverse line. Theorem 3 shows that the smaller the $\epsilon$, the smaller the probability that the ID/OOD split is well-specified. The probability of a well-specified ID/OOD split when $\epsilon = 0$ is also 0.

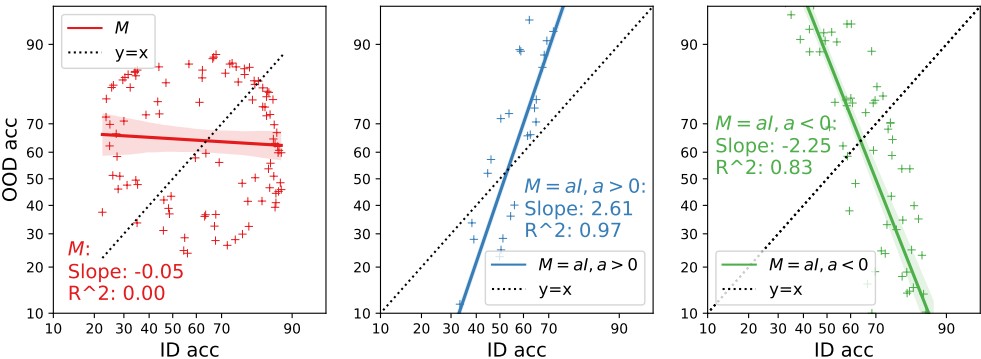

Figure 3: ID vs. OOD accuracy on probit scale. When $M$ is sampled arbitrarily, such that the SNR condition is satisfied (Theorem 2), the strong correlation between ID and OOD accuracy disappears. (*left figure*). For $M_{\text{ID}} = I$ and $M_{\text{OOD}} = aI$, there is only a linear scaling in the spurious correlation and spurious feature variance. In this setting, we observe a correlation between ID and OOD accuracy. However, when spurious correlation reversal is not satisfied, $a > 0$ (*middle figure*), we have positive accuracy on the line. When $a < 0$ (*right figure*), we have the spurious correlation reversal condition and have accuracy on the inverse line, where there is a strong but negative correlation between in- and out-of-distribution accuracy. More experimental details can be found in Appendix B.

**Theorem 3** (Benchmarks with Accuracy on the Line are Misspecified Almost Everywhere). *Define*

$$\mathcal{W}_\epsilon = \left\{ M \in \mathbb{R}^{l \times l} : \quad \begin{aligned} & w_{spu}^\top (M\,\mu_{spu}) + \sqrt{2\,(L_\phi\,\kappa)^2\,\|w_{spu}\|_2^2\,\log(1/\delta)} < 0 \quad \textit{(well-specified; Theorem 1),} \\ & \text{and} \\ & \left| \Phi^{-1}\big(acc_{P_{OOD}}(f_X)\big) - a\,\Phi^{-1}\big(acc_{P_{ID}}(f_X)\big) \right| \leq \epsilon \quad \textit{(accuracy on the line; Definition 6)} \end{aligned} \right\}$$
(12)

*Then:*

(i) $\mathcal{W}_0$ *has Lebesgue measure zero in* $\mathbb{R}^{l \times l}$ *(i.e., probability zero for a random draw).*

(ii) *For any* $0 \leq \epsilon_i \leq \epsilon_j$, *we have* $\mathcal{W}_{\epsilon_i} \subseteq \mathcal{W}_{\epsilon_j}$.

*The proof of Theorem 3 and supporting Lemmas are provided in Appendix A.6.*

Theorem 3 demonstrates that the two conditions in Equation 12 are at odds. In particular, as $\epsilon \to 0$ (i.e., perfect accuracy on the line), almost every shift is misspecified, and the Lebesgue measure (probability) of the set of well-specified shifts grows monotonically with $\epsilon$, i.e., inversely with accuracy on the line. This means that when we observe accuracy on the line, with a high probability, the ID/OOD split is misspecified. In Appendix A.7, we construct an intuitive example of such (zero-measure) shifts, where both the conditions for well-specified splits and accuracy on the line hold.

To summarize, Theorem 1-2 give necessary and sufficient conditions for well-specified domain generalization ID/OOD splits, and Theorem 3 demonstrates that there is zero probability that such conditions and perfect accuracy on the line for the ID/OOD split simultaneously hold—Figure 3. Moreover, accuracy on the line is at odds with well-specified shifts. Our results suggest that datasets with accuracy on the line may be misspecified for benchmarking domain generalization. In contrast, benchmarks with *accuracy on the inverse line*, or weak correlation between ID and OOD accuracy, are better suited to benchmark domain generalization. The accuracy on the line property now gives a test for well-specified benchmarks. Building on this insight, we apply this test to state-of-the-art domain generalization benchmarks (Section 4).

# 3 Domain Generalization Datasets and Related Work

We first consider two influential domain generalization benchmark suites: DomainBed (Gulrajani & Lopez-Paz, 2020) and WILDS (Koh et al., 2021). **DomainBed** is a collection of object recognition domain generalization benchmarks. For example, PACS (Khosla et al., 2012; Li et al., 2017) includes images of seven classes across four domains: Photos, Art Paintings, Cartoons, and Sketches. Another benchmark is ColoredMNIST (Arjovsky et al., 2019), a semi-synthetic binary classification variation of MNIST (Deng, 2012), which introduces color as a spurious correlation and defines domains by specific color-label associations. Gulrajani & Lopez-Paz (2020) found that empirical risk minimization achieved the best transfer performance compared to state-of-the-art domain generalization algorithms across PACS, ColoredMNIST, and other DomainBed benchmarks. They also observed differences in their findings based on their choice of model selection.

**WILDS** was designed to better represent real-world shifts across vision and language. For example, Camelyon17 (Zech et al., 2018; AlBadawy et al., 2018) includes images of tissue that may contain tumor tissue (classes) from different hospitals with varying conditions (domains). CivilComments Borkan et al. (2019) includes comments to an online article that may be toxic (class) for different subpopulations defined by demographic identities (domains). These benchmarks illustrate the suite's focus on practical and natural real-world applications. However, even across WILDS benchmarks, no state-of-the-art methods have demonstrated consistent superiority over ERM (Koh et al., 2021).

For **Subpopulation shift**, which we consider a special case of the broader domain generalization task, benchmarks are designed specifically to evaluate distribution shift robustness in scenarios where spurious correlations lead to worse performance on underrepresented subgroups out-of-distribution. For example, the Waterbirds benchmark (Sagawa et al., 2019) introduces a spurious correlation between bird species and backgrounds, such as waterbirds predominantly appearing in water environments. Classifiers that rely on backgrounds rather than bird features when predicting bird type (classes) generalize poorly to new domains where backgrounds are urban areas—birds in urban backgrounds are undersampled in the training data.

Benchmarks addressing **other types of distribution shifts** where domain generalization is desired have also been proposed, e.g., temporal shifts (Yao et al., 2022; Joshi et al., 2023; Zhang et al., 2023).

Other conceptualizations of distribution shift datasets have been extended to benchmark domain generalization. However, not all are designed to evaluate robustness to spurious correlations, nor do they represent multi-source domain generalization settings. For instance, some benchmarks are designed to assess robustness against typical data quality variations encountered in real-world settings, like variations of ImageNet (Deng et al., 2009; Hendrycks & Dietterich, 2019; Hendrycks et al., 2021). Such benchmarks are out of scope for our study as they are not standard benchmarks for the domain generalization tasks we study. Nevertheless, we refer to previous work studying accuracy on the line for these datasets (Taori et al., 2020).

*Developing and curating benchmarks is an arduous process and is of immense service to the research community.* DomainBed, WILDS, and other benchmarks are useful when applied in the appropriate scope. Our work demonstrates that robustness to spurious correlations and related tasks are inappropriate applications of many of these benchmarks. In the next section, we outline design principles to develop new benchmarks for spurious correlations. Furthermore, we identify benchmarks that our analysis suggests are well-specified.

## 3.1 Accuracy on the Line

The accuracy on the line phenomena has been observed empirically in previous work for many benchmarks in these suites (Recht et al., 2019; Miller et al., 2021; 2020; Taori et al., 2020; Baek et al., 2022; Saxena et al., 2024). However, Liu et al. (2023) identify real-world tabular datasets with weak or negative linear correlation, and Teney et al. (2023) identify non-tabular real-world datasets where ID and OOD performance exhibit other patterns between in- and out-of-distribution accuracy beyond strongly positive and linear. Notably, they also found that correlations can be inverted in some datasets. Sanyal et al. (2024) derive noise conditions to achieve shifts with accuracy on the inverse line. Our work uniquely specifies the implications of accuracy on the line (or lack thereof) on the utility of datasets as domain generalization benchmarks.

Closest to our work is Bell et al. (2024), which studies spurious correlation benchmarks and shows that different datasets often disagree on which methods perform best, hindering reliable conclusions about spurious correlation mitigation. They propose three desiderata: (i) ERM Failure, (ii) Discriminative Power, and (iii) Convergent Validity[‡], arguing that good benchmarks must produce group-wise errors under standard empirical risk minimization, distinguish among methods, and agree with other benchmarks that test the same phenomenon. However, they focus primarily on subpopulation shifts and systematically assessing the agreement between existing benchmarks (e.g., Waterbirds). Although both studies highlight the importance of benchmark design, our primary goal is to formally characterize when distribution shifts fail to penalize reliance on spurious correlations, resulting in an ill-posed domain generalization setup, rather than to compare a wide range of algorithms across multiple datasets to demonstrate agreement or lack thereof.

Other works have studied conditions where the empirical risk minimizer for observed distributions is sufficient for domain generalization. Rosenfeld et al. (2022b) show this to be the case under domain interpolation, i.e., OOD distributions are convex combinations of observed distributions (in an online setting), see example in Figure 2c. Additionally, when domain-general features are fully informative, i.e., spurious correlations are redundant, and under some conditions, Ahuja et al. (2021) also show this to be the case. Our results corroborate their findings.

**Worst Case Consideration.** This notion of worst-case stress testing to establish robustness under distribution shift is not new. When evaluating the out-of-distribution generalization of deep learning methods developed on the ImageNet task (Deng et al., 2009) (*ImageNet Large Scale Visual Recognition Challenge (ILSVRC)* (Russakovsky et al., 2015)), Kornblith et al. (2019) assess generalization by applying said methods to contemporary datasets with presumably different distributions than ImageNet, such as CIFAR-10 (Krizhevsky, 2009), however, the datasets they investigated could be considered quite similar in distribution to ImageNet. Alternatively, Salaudeen & Hardt (2024) adversarially constructed a dataset, ImageNot, designed to shift spurious correlations present in the original ImageNet construction strongly. Notably, the findings of Salaudeen & Hardt (2024) align with those of Kornblith et al. (2019) despite the deliberately adversarial construction of ImageNot. We argue that constructing datasets with such adversarial spurious correlation shifts is essential for rigorously probing a classifier's use of spurious correlations.

More generally, other works have proposed alternative approaches to developing domain generalization benchmarks. Satisfying our conditions introduces an additional necessary dimension for creating more meaningful evaluations. For example, Zhang et al. (2023) argues that many existing benchmarks are limited by having too few domains and overly simplistic settings, which restrict their ability to simulate the significant distribution shifts observed in real-world scenarios. Similarly, Lynch et al. (2023) contend that benchmarks inadequately capture the complex, many-to-many spurious correlations that can arise in practical applications.

Like previous work, we next investigate the accuracy of the line properties of state-of-the-art domain generalization benchmarks to take an inventory of well-specified datasets. Table 1 provides a summary.

## 4 Empirical Results

We evaluate the correlation between in-domain (ID) and out-of-domain (OOD) accuracy for benchmarks in the popular DomainBed (Gulrajani & Lopez-Paz, 2020) and WILDS (Koh et al., 2021) benchmark suites, as well as subpopulation shift benchmarks, e.g., WaterBirds (Sagawa et al., 2019).

**Datasets.** Specifically, our results include the following datasets: **Camelyon** (Bandi et al., 2018; Koh et al., 2021), **CivilComments** (Borkan et al., 2019; Koh et al., 2021), **ColoredMNIST** (Arjovsky et al., 2019; Gulrajani & Lopez-Paz, 2020), **Covid-CXR** (Alzate-Grisales et al., 2022; Cohen et al., 2020b; Tabik et al., 2020; Tahir et al., 2021; Suwalska et al., 2023), **FMoW** (Christie et al., 2018; Koh et al., 2021), **PACS** (Li

---

[‡]The extent to which a test or measure correlates with other tests or measures that are designed to assess the same or similar constructs, indicating that the different measures are capturing the same underlying concept. To satisfy Convergent Validity, a benchmark should agree with similar benchmarks and disagree with those that are dissimilar (Bell et al., 2024; Salaudeen et al., 2025)

Table 1: ✓: well-specified ($R < 0.3$); ✗: misspecified ($R > 0.3$). We train on a set of ID distributions and test on a left-out OOD distribution. We present the Pearson R of ID and OOD probit-transformed accuracies and the slope and intercept of OOD accuracy regressed on ID accuracy. (*) OOD for waterbirds refers to the group where $y = 0$ and $a = 0$. The ID dataset is a mixture of groups at train time. Additional datasets and analysis are provided in Appendix C, which also includes complete tables with all splits for each dataset. With a conservative threshold of $R < 0.3$, only a subset of datasets satisfies our derived conditions. We note that 0.3 is commonly regarded as the threshold for a "weak" correlation, which is the sole justification for our choice.

| Dataset | OOD | $R < 0.3$ | slope | offset | R | p-value | std error |
|---|---|---|---|---|---|---|---|
| ColoredMNIST | Env 2 acc | ✓ | -1.56 | 0.47 | -0.74 | 0.00 | 0.01 |
| CXR | Env 1 acc | ✓ | -0.60 | 0.56 | -0.48 | 0.00 | 0.03 |
| SpawriousO2O hard | Env 0 acc | ✗ | 0.32 | -0.21 | 0.50 | 0.00 | 0.05 |
| SpawriousM2M hard | Env 0 acc | ✗ | 0.76 | -0.26 | 0.94 | 0.00 | 0.01 |
| SpawriousO2O easy | Env 0 acc | ✗ | 0.48 | -0.29 | 0.74 | 0.00 | 0.04 |
| SpawriousM2M easy | Env 0 acc | ✗ | 0.34 | 0.26 | 0.60 | 0.00 | 0.00 |
| PACS | Env 1 acc | ✗ | 0.68 | -0.68 | 0.84 | 0.00 | 0.01 |
| TerraIncognita | Env 1 acc | ✗ | 0.83 | -1.41 | 0.74 | 0.00 | 0.02 |
| Camelyon | Env 2 acc | ✗ | 0.62 | 0.49 | 0.78 | 0.00 | 0.01 |
| Subpopulation Shift Datasets | | | | | | | |
| CivilComments | Env 1 acc | ✓ | -0.49 | 0.16 | -0.47 | 0.00 | 0.03 |
| WaterBirds | Env 0 (*) acc | ✓ | -0.13 | 1.58 | -0.13 | 0.00 | 0.03 |
| FMoW | Env 5 acc | ✗ | 0.76 | -0.61 | 0.87 | 0.00 | 0.01 |

et al., 2017; Gulrajani & Lopez-Paz, 2020), **Spawrious** (Lynch et al., 2023), **TerraIncognita** (Beery et al., 2018; Gulrajani & Lopez-Paz, 2020), and **Waterbirds** (Sagawa et al., 2019).

**Model Architectures.** For vision datasets, we leverage pretrained deep learning architectures such as **ResNet-18/50** (He et al., 2016), **DenseNet-121** (Huang et al., 2017), **Vision Transformers** (Dosovitskiy et al., 2020), and **ConvNeXt-Tiny** (Liu et al., 2022). For language datasets, we utilize pretrained embeddings from **BERT** (Devlin et al., 2019) and **DistilBERT** (Sanh et al., 2020), and apply lower-capacity machine learning classifiers, such as logistic regression, for downstream classification tasks.

**Experimental Setup.** The benchmarks we consider include a set of domains, where domains are distinct data distributions. As standard in the literature (Gulrajani & Lopez-Paz, 2020; Koh et al., 2021), the in-distribution (ID) data is defined as a mixture of a subset of the data domains, and the out-of-distribution (OOD) data is not included in the training domains, that is, we perform a leave-one-domain-out ID/OOD splits for our experiments, where we train on all but one domain and use the left-out domain as OOD. We generate classifiers for our experiments by varying classifier and training hyperparameters for each architecture, including the number of training epochs (Appendix C Table 2). Our experiments include training these classifiers end-to-end with varying hyperparameters and data augmentations, as well as finetuning pretrained classifiers and transfer learning.

Table 1 highlights the prevalence of widely-used domain generalization benchmarks with accuracy on the line, a signature of potential misspecification, while Figure 4 qualitatively illustrates benchmarks with weak or strongly negative correlation between in and out-of-distribution accuracy. We provide a detailed account of our experiments, along with benchmark-specific discussions, in Appendix C.

**Selecting Number of classifiers.** To ensure a robust estimate of ID-OOD correlation, we follow and extend the approach of prior work by training a diverse set of classifiers that vary in architecture, random seed, data order, and hyperparameters. This diversity mitigates the risk of single-classifier artifacts influencing our findings. We adopt a simple heuristic: continue adding classifiers until the ID-OOD Pearson correlation changes by less than 1% with the addition of new classifiers. As shown in Table 3, this threshold is consistently reached well before exhausting our pool of trained classifiers. In practice, we train 2–10× more classifiers than

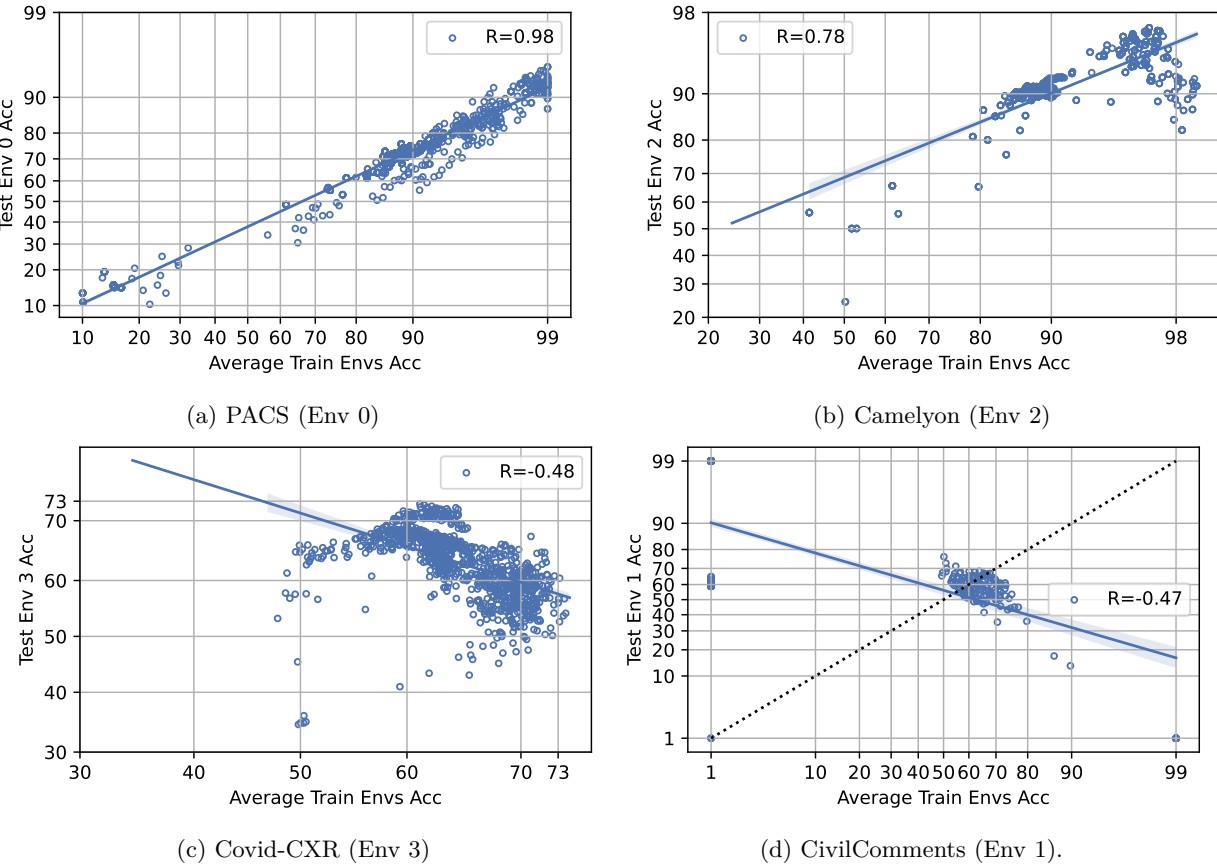

Figure 4: We show some ID/OOD splits of popular domain-generalization benchmarks with a strong positive, weak, or strong negative correlation between in-distribution and out-of-distribution accuracy. Our results suggest that algorithms that consistently provide models with the best transfer accuracies for these splits are at least partially successful in removing spurious correlations. For Camelyon (b), within some accuracy range, we have accuracy on the inverse line, indicating the importance of a qualitative assessment on these trends.

previous studies, often exceeding thousands per dataset and ensuring our correlation estimates are stable and conservative (Miller et al., 2021). For instance, we train over 10,000 classifiers for a single CivilComments split, and our criterion is satisfied with less than 7,000 classifiers. We train over 5,000 classifiers for a PACS environment, and our criterion is satisfied with 910. We trained as many classifiers as computationally feasible to ensure the robustness of our findings.

## 4.1 Findings

We find that semisynthetic datasets more reliably satisfy our derived conditions. Particularly, semisynthetic here means real-world datasets that either (i) have real-world but artificial spurious correlations introduced (ColoredMNIST/color, Spawrious/background, and Waterbirds/background) or (ii) have some selection process that introduces spurious correlations (CivilComments). Other datasets without careful consideration with respect to spurious correlations generally have accuracy on the line. The Covid CXR datasets from different regions sometimes exhibit a relationship that Teney et al. (2024) refers to as 'No Transfer,' where the OOD accuracy is near constant despite variance in the ID accuracy. Our results further echo Teney et al. (2024)'s notes on potentially misleading advice from past studies: particularly the recommendation of focusing on ID performance to improve OOD robustness (Wenzel et al., 2022), which our work suggests is only a reasonable strategy in settings where distribution shifts are relatively simple and weak.

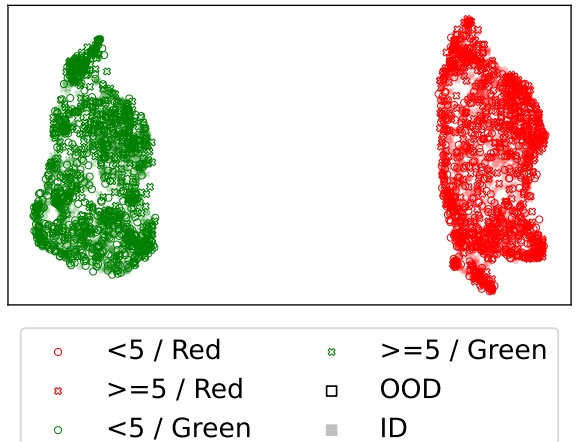

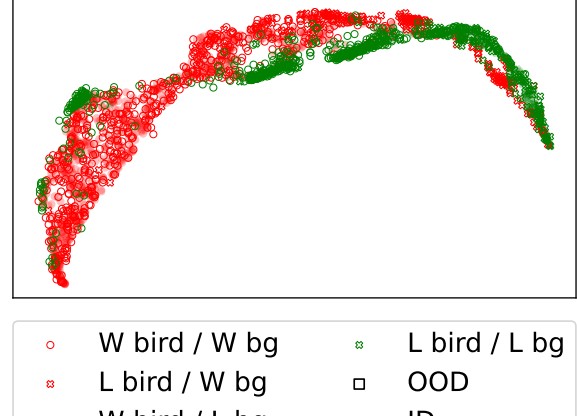

(a) ColoredMNIST. Embeddings of a model trained on Env 1 and 2. The labels are $< 5, \geq 5$ and the spurious feature is color (red, green). The UMAP of the embeddings clusters based on color (spurious) rather than digit features (domain-general). This leads to incorrect predictions OOD where the color-label correlation reverses, as observed in our experiments (Section C.2).

(b) Waterbirds. Embeddings trained on examples with spurious correlation between bird type and background. The bird type is the label, and the background is the spurious feature. The UMAP of the embeddings clusters based on background type (spurious) more strongly than bird features (domain-general). This leads to incorrect predictions OOD where background-label correlation changes, as observed in our experiments (Section C.9).

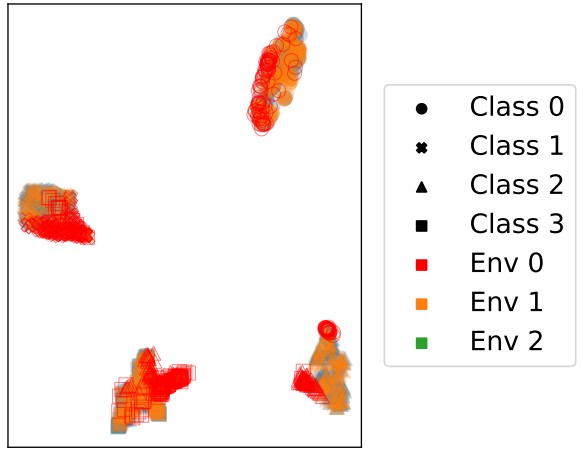

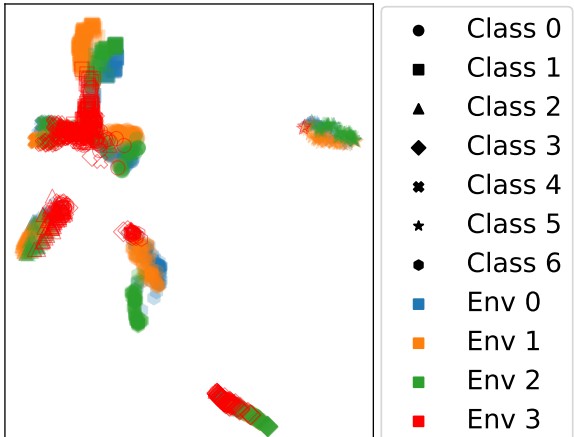

(c) Spawrious O2O Hard. Embeddings of a model trained on Env 1 and 2. The label is animal type ,and the spurious feature is background. Generally, for some classes (e.g., Class 0), some subset of OOD (Env 0) examples cluster very closely to the same classes in in-distribution data. However, for the same class, many OOD examples incorrectly cluster more closely to class 2; there is also smaller variance in these examples. We observe this for most classes, demonstrating that the model is relying on spurious correlations. This matches our observations of OOD performance degradation due to spurious correlations (Section C.3.2)

(d) PACS. Embeddings of a model trained on Env 0-2. The spurious feature is unknown, but it is purported to exist because each environment has a different image style. We find that there is a relatively strong alignment between the in-distribution and out-of-distribution embeddings for each class. This matches our finding that there is not a strong spurious correlation that would cause models to embed examples based on features that have an unstable relationship between labels (Section C.4), like we see in the other examples we analyzed with UMAP.

Figure 5: We use a UMAP to visualize the embedding clusters for examples. We fit the UMAP to held-out examples of the in-distribution data and also apply it to the out-of-distribution data.

Our results and findings should not be surprising given that the spurious correlations we aim to mitigate for real-world decision-making tend not to be localized. For instance, correlations between gender and occupation likely persist across 'naturally' collected datasets (Teresa-Morales et al., 2022). Hence, they may not harm performance across different data sources. We find that the subpopulation shift benchmarks we assess often have the desired properties derived in this work (Yang et al., 2023). Notably, these benchmarks are (i) constructed so that the spurious correlation from training is no longer accurate at testing and (ii) evaluated based on worst-group (worst-case) performance. Our work shows that in less explicitly defined domain generalization contexts, such types of shifts are also necessary. We provide detailed results and discussion in Appendix C.

**Embeddings Analysis.** We qualitatively verify that classifiers learn spurious correlations without targeted algorithms. We use UMAP (Uniform Manifold Approximation and Projection), a fast, scalable dimensionality reduction technique of feature representations that preserves both local and global structure of data (McInnes et al., 2018). Figure 5 demonstrates the concordance between our findings and the embeddings learned by classifiers. For instance, for ColoredMNIST, Waterbirds, and Spawrious, embeddings of classes are consistently clustered together in the training distribution; however, out of distribution, the same classes' embeddings are aligned with the wrong class from the in-distribution data, and would likely be labeled incorrectly. The influence of these spurious correlations leads to clustering examples with the wrong class out of distribution when the spurious correlations change. We find that these datasets also have a lower (negative) correlation between in- and out-of-distribution accuracy.

For PACS, however (Figure 5d), the alignment between in- and out-of-distribution embeddings is consistent, meaning that the same classes in- and out-of-distribution are closely embedded. Our analysis also finds that PACS has a strong correlation between in- and out-of-distribution accuracy. Overall, our qualitative analysis of the embeddings learned by in-distribution optimal classifiers supports our theoretical and empirical findings.

## 4.2 Discussion

Domain generalization considers worst-case shifts (Arjovsky et al., 2019; Rosenfeld et al., 2022b). While worst-case shifts may have a trade-off with average utility (Salaudeen & Koyejo, 2024; Miller, 2024), it is crucial in population-sensitive and high-stakes applications like healthcare (Angwin et al., 2016; Chen et al., 2019; Oakden-Rayner et al., 2020). However, domain generalization benchmarks have not been rigorously assessed for their utility in such scenarios. We address this gap by deriving conditions for well-specified benchmarks, helping practitioners align benchmark choices with their goals. We showed that well-specified benchmarks, i.e., those that are aligned with worst-case shifts, will not have accuracy on the line. Hence, benchmark users and curators with a goal of addressing such shifts should seek out benchmarks with weak or negative ID and OOD accuracy correlation.

We emphasize that **dataset availability reflects selection bias.** The frequency and importance of real-world scenarios where spurious correlations degrade OOD performance, particularly in high-stakes domains like healthcare, policing, and finance (see previously cited examples), cannot be dismissed simply because they are underrepresented in public benchmarks. In these domains, spurious correlations often stem from historical biases and can lead to harmful outcomes if left unmitigated. However, privacy constraints frequently limit the release of relevant datasets (Cohen & Mello, 2018).

Many benchmark curators define their intended scope carefully. For example, WILDS (Koh et al., 2021) focuses on common real-world shifts rather than worst-case shifts. Our findings often validate these intended scopes, yet benchmark users may not always adhere to them. To clarify benchmark suitability, we categorize benchmarks into (i) worst-case and (ii) natural shifts (Koh et al., 2021; Taori et al., 2020). We emphasize that worst-case benchmarks are particularly valuable for auditing uninterpretable classifiers, such as detecting demographic biases in classifier decisions (Ferrer et al., 2021), whereas natural shifts may suffice when prioritizing average OOD performance. Our conditions enable assessing whether spurious correlations (e.g., race in Chest X-Rays (Gichoya et al., 2022)) impact predictions.

Importantly, many benchmarks that do not satisfy our conditions reflect other types of shifts observed in the real world, where spurious correlations, if they exist, are preserved, but some other source of shift occurs. In such contexts, non-worst-case benchmarks have significant utility. However, these benchmarks do not evaluate strong shifts in spurious correlations, such as when they are removed or inverted via interventions like a pandemic or policy.

Building on our insight, for robust algorithm development, both worst-case and natural shifts should be considered to ensure broad applicability. Additionally, new spurious correlation benchmarks should undergo *accuracy on the line* evaluations to support their reliability—ideally, there is no positive accuracy on the line.

In general, it is crucial to benchmark classifiers on both natural and worst-case benchmarks to comprehensively characterize the robustness spectrum. This requires identifying and categorizing benchmarks as natural or worst-case, a framework for which this work carefully develops conditions.

Finally, **domain expertise is critical in defining spurious correlations**: a narrow potential distribution set may limit robustness, while an overly broad definition of potential distributions can unnecessarily reduce utility (Shen et al., 2024). For instance, Chiou et al. (2024) report that training a classifier on multiple recording sessions degrades OOD (new session) brain-computer interface (BCI) classification performance compared to training on single-session data.

### 4.3 Applying Our Findings

**Constructing Benchmarks Without Positive Accuracy on the Line.** Indeed, some of the datasets we study empirically that satisfy our desired conditions are primarily semisynthetic (e.g., ColoredMNIST, Spawrious, Waterbirds). In contrast, datasets such as Covid-CXR and WILDSCamelyon have in-distribution (ID) and out-of-distribution (OOD) splits that do not exhibit positive accuracy on the line. This suggests that careful and intentional data collection and curation are likely necessary to obtain naturally occurring datasets without positive accuracy on the line.

One approach to constructing such datasets is to **identify settings where a natural experiment or intervention has occurred**. For example, the Covid-CXR dataset leverages the natural intervention of the pandemic and regional variations. Such interventions can remove confounds that induce spurious correlations or alter the directions of spurious correlations. Without such interventions, there may be little reason to expect that spurious correlations would fail to generalize in the real-world datasets that are most readily available.

Another approach is to split datasets along axes where there are strong spurious correlation effects, such as demographics in human decision-making. An example of this is CivilComments, where data is split across demographic attributes that affect human-generated labels of toxicity. not toxic.

Before finalizing a benchmark, we should measure the correlation between ID and OOD performance to **ensure that the dataset does not exhibit strong positive accuracy on the line**. This validation can help confirm that the benchmark is well-specified and truly tests robustness to spurious correlations.

**Qualitatively Assessing Accuracy on the Line.** For the Camelyon dataset (Section C.6), we observe a shift in correlation patterns based on classifier accuracy. Specifically, classifiers with greater than 90% accuracy exhibit a negative correlation between ID and OOD performance, whereas classifiers below this threshold show a positive correlation. Since accuracy on the line (Definition 3) is a global property, this dataset does not meet the criteria for accuracy on the line since there is a strong deviation from the positive correlation in some regimes. This underscores a key limitation: evaluating overall correlation may not sufficiently identify well-specified benchmarks. While this approach is robust against false negatives of misspecification, it may introduce false positives, necessitating qualitative inspection as a complementary assessment tool.

**On the Slope vs. the Correlation Coefficient.** In this work, we focus on the correlation between ID and OOD accuracy rather than the slope between the two quantities. Recall that for a set of pairs $x, y$, which have a Pearson R correlation of $r$, the slope when $y$ is regressed on $x$ is $s = r * (\sigma_y/\sigma_x)$. $s$ depends

on variances $\sigma_y$ and $\sigma_x$, which may or may not be relevant to the spurious correlation problem. The slope depends strongly on the definition of $\mathcal{F}$ (Appendix C). This observation is also related to limitations in attributing the accuracy drop across distribution only to distributional systematic bias (Salaudeen & Hardt, 2024). While $s$ is informative, its relationship with our derived conditions is not as obvious.

**Reconciling Worst-Case and Average-Case Generalization.** A strict worst-case approach to generalization can conflict with developing locally (sites) beneficial classifiers, particularly in healthcare (Futoma et al., 2020; Miller, 2024). However, failures even within the same site suggest this tension is unavoidable (Oakden-Rayner et al., 2020), as spurious correlations remain brittle due to other non-local factors like temporal drifts (Ji et al., 2023) or interventions (Birkmeyer et al., 2020). Reliable worst-case robustness benchmarks remain essential, even with a narrower scope. Still, when worst-case focus severely impacts average utility, additional evaluation on alternative benchmarks is warranted. A practical approach is to narrow the classifier's deployment scope to a smaller set of distributions, which can improve worst-case performance (e.g., when the shifts in the smaller set are weaker) without sacrificing as much average utility. However, practically, maintaining reliable predictions may require scope-dependent classifier monitoring and updates.

## 4.4 Downstream Implications

**Implications on Key Domain Generalization and Evaluation Practices.** ID/OOD splits within the same dataset can exhibit varying Pearson R correlations, meaning some splits provide more reliable benchmarks than others. However, **averaging** over all ID/OOD splits (Gulrajani & Lopez-Paz, 2020) dilutes this reliability, particularly when only one split is well-specified. The issue worsens when averaging across multiple datasets to compare domain generalization methods (Gulrajani & Lopez-Paz, 2020). In subpopulation shifts, **worst-group accuracy** is the standard evaluation metric (Koh et al., 2021); adopting a similar norm more generally for domain generalization improves the robustness of evaluation.

A related issue arises in **classifier selection** via cross-validation, where selecting classifiers based on held-out accuracy, whether ID or a held-out domain, can lead to overfitting to spurious correlations specific to that set. Alternative selection criteria are implied conditional independencies (Salaudeen & Koyejo, 2024), cross-risk minimization (Pezeshki et al., 2023), and confidence-based ensemble aggregation (Chen et al., 2023). However, classifier selection under distribution shifts remains a challenge.

Furthermore, accuracy on the line can act as a pre-check to decide whether algorithmic invariance penalties are worth tuning. If there are spurious correlations between a set of observed environments, for any single test environment, it may be the case that classifiers that ignore these spurious correlations perform worse than a classifier that utilizes them. Such settings will exhibit accuracy on the line. However, in other environments, the same spurious correlations can lead to failure. We expect such settings not to exhibit accuracy on the line.

**Implications on Benchmarking Causal Representation Learning** aims to uncover underlying causal structures. One approach to this is *independent causal mechanisms* (Pearl, 2009; Schölkopf et al., 2021), which has motivated many domain generalization algorithms (Arjovsky et al., 2019; Salaudeen & Koyejo, 2024; Peters et al., 2016). Since both tasks require distinguishing stable from spurious correlations, our results on evaluating domain generalization also apply to benchmarking causal representation learning. Specifically, when assessing classifiers, including disentangled causal classifiers, based on OOD accuracy, our framework helps determine when domain generalization reliably reflects success in learning causal representations, as discussed in Salaudeen et al. (2024).

**Implications on Benchmarking Algorithmic Fairness** which aims to mitigate biases that cause disparate performance across demographic groups; some definitions of fairness are closely linked to domain generalization (Creager et al., 2021). Group sufficiency particularly aligns with the principle of invariance (Chouldechova, 2017; Liu et al., 2019). We emphasize a straightforward but key insight: when using OOD accuracy to benchmark whether classifiers avoid relying on group information, the benchmark must

ensure that group information hinders out-of-distribution performance. In this case, a strong positive correlation between training and worst-group test accuracy suggests that group information generalizes. In contrast, a weak or negative correlation between ID and OOD accuracy is preferable.

Modern **foundation classifiers** are susceptible to spurious correlations (Alabdulmohsin et al., 2024; Zhu et al., 2023; Gerych et al., 2024; Hamidieh et al., 2024). Analyzing the CivilComments dataset, we find that spurious correlation shifts in language datasets exhibit similar patterns to vision datasets within our framework, showing strong positive, negative, and weak correlations between ID and OOD accuracy. Furthermore, Saxena et al. (2024) report strong positive correlations in large language classifiers for predictive tasks (Q/A) under distribution shift. Our results highlight that the benchmark conditions we establish are also crucial for evaluating spurious correlations in foundation classifiers.

**State-of-the-Art Algorithms.** Many algorithms have been proposed for domain generalization Gulrajani & Lopez-Paz (2020); Koh et al. (2021); Yang et al. (2023). On datasets we identify as misspecified, such as PACS, Terra Incognita, WILDSCamelyon, and WILDFMoW, empirical risk minimization (ERM) performs comparably to state-of-the-art domain generalization algorithms (Gulrajani & Lopez-Paz, 2020; Koh et al., 2021), particularly when controlling for factors like the use of unconstrained, large-scale pretrained classifiers (e.g., CLIP (Radford et al., 2021), trained on internet-scale data rather than ImageNet). Specific results for WILDS datasets are available at https://wilds.stanford.edu/leaderboard/, maintained by the WILDS team.

In contrast, on datasets we identify as well-specified, several algorithms have consistently outperformed ERM, even under independent validation. For example,(Yang et al., 2023) demonstrates improvements over ERM on Waterbirds and CivilComments.

## Limitations and Future Work

Yang et al. (2023) demonstrates that while accuracy on the line may hold, other metrics or a combination of metrics may not simultaneously have the same strong positive linear trend. Investigating metrics other than accuracy is left for future work. Additionally, our theoretical analysis focuses on binary classification, though our empirical results include both binary and multi-class classification. We leave extending our theoretical analysis to multiclass (and multilabel) classification for future work. Furthermore, our results are only necessary and sufficient for symmetric distributions where spurious and domain general features are conditionally independent given the label. We leave studying settings with less restrictive assumptions to future work.

Another direction for future work is to develop a more robust automated method for assessing accuracy on the line, beyond simply computing correlation across all training accuracy levels. Currently, qualitative assessment remains necessary to account for cases where classifiers with higher training accuracy exhibit negative accuracy on the line. Empirically, we found that standard change-point detection methods (Killick et al., 2012) are highly sensitive to noise in accuracy measurements. However, incorporating more robust heuristics could improve their reliability, making them a viable approach for automation.

Additionally, while we have evaluated a wide variety of classifiers in this work, continuing to collect data points of ID/OOD accuracies for benchmarks improves the accuracy of the true relationship. Finally, we leave curating additional benchmarks without accuracy on the line—from diverse, real-world scenarios with high-dimensional spurious features—for future work. This work, along with others (Recht et al., 2019; Taori et al., 2020; Miller et al., 2021), has characterized this property for a variety of popular benchmarks.

## Broader Impact

Our work aims to help the community build evaluation tools that assess robustness to spurious correlations. By formalizing when a benchmark is *well-specified* for this task, we give researchers a simple test for deciding whether gains in-distribution can, or cannot, predict gains out-of-distribution. Well-specified benchmarks will push methodological advancements toward developing classifiers that remain reliable and robust once deployed in critical real-world applications, such as in medical imaging and ecology.

Finally, well-specified benchmarks are only one step toward safe deployment. Practitioners must still test for domain-specific failure modes and monitor classifiers in production. We believe the overall impact of this work is positive: exposing benchmark misspecification moves the field toward classifiers that succeed for the right reasons and reduces unexpected failures in real applications.

## 5 Conclusion

Robustness to spurious correlations under worst-case distribution shifts is a critical challenge in machine learning, essential for ensuring the reliability and fairness of classifiers (Pfohl et al., 2025). In this work, we identify significant limitations in current benchmarks designed to address this problem. Specifically, many state-of-the-art benchmarks, which evaluate out-of-distribution (OOD) accuracy by training classifiers on an in-distribution (ID) split and testing on an OOD split, fail to guarantee that classifiers free of spurious correlations will transfer better. We define a benchmark as *well-specified* if such a guarantee exists.

Previous work observed that many benchmarks exhibit the phenomenon of *accuracy on the line*, where improved ID performance directly correlates with improved out-of-distribution performance. Our theoretical findings suggest that this behavior indicates that such benchmarks are misspecified for evaluating domain generalization and emphasize the importance of prioritizing benchmarks that do not exhibit accuracy on the line when addressing worst-case distribution shifts. We aim to provide a clearer path toward developing classifiers robust to spurious correlations by addressing the evaluation ambiguity. Reliable benchmarks for robustness to spurious correlations contribute to the broader efforts to develop meaningful and valid evaluations of AI systems (Weidinger et al., 2025; Salaudeen et al., 2025; Wallach et al., 2025).

## Acknowledgements

OS was partly supported by the UIUC Beckman Institute Graduate Research Fellowship, NSF-NRT 1735252, GEM Associate Fellowship, and the Alfred P. Sloan MPhD Program. SK acknowledges support from NSF 2046795 and 2205329, the MacArthur Foundation, Stanford HAI, and Google Inc. We thank A. Anas Chentouf, Tyler LaBonte, Vivian Nastl, Haoran Zhang, and Yibo Zhang for their comments on an earlier draft.

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

## Appendix Table of Contents

## A Proofs

### A.1 Proof of Lemma 1—Domain-Specific Classifiers have Lower In-Domain Error under Partially Informative Domain-General Features

Assume Non-trivial and non-redundant features (Assumption 1-2), and strongly convex $\ell$.

$$\min_{f \in \mathcal{F}} \mathbb{E}_{(X,Y) \sim P}\big[\ell(f(X), Y)\big] < \min_{f \in \mathcal{F}_{\mathrm{dg}}} \mathbb{E}_{(X,Y) \sim P}\big[\ell(f(X), Y)\big], \tag{13}$$

where $\mathcal{F} : \mathcal{X} \to \mathbb{R}$ where $f(x) = \mathrm{w}^\top x = \mathrm{w}_{\mathrm{dg}}^\top z_{\mathrm{dg}} + \mathrm{w}_{\mathrm{spu}}^\top z_{\mathrm{spu}}$, $f \in \mathcal{F}$. For $f \in \mathcal{F}_{\mathrm{dg}}$, $f(x) = (\mathrm{w})^\top x = \mathrm{w}_{\mathrm{dg}}^\top z_{\mathrm{dg}}$.

*Proof.* From Assumption 1-2, non-trivial and non-redundant features, a classifier that uses both domain-general and spurious features is more expressive than one that does not. Let $w$ be the Bayes optional By the

Bayes optimality of $w^*$, any $w$ achieving the same risk agrees with $w^*$ almost everywhere, i.e.,

$$\mu\left(\left\{x \in \mathcal{X} \mid w^{*\top}x \neq w^\top x\right\}\right) = 0,$$

where $\mu$ denotes the Lebesgue measure on $\mathcal{X}$.

Let $x = z_{\mathrm{dg}} \oplus z_{\mathrm{spu}}$ and $w = w_{\mathrm{dg}} \oplus w_{\mathrm{spu}}$ such that

$$w^{*\top}x = w_{\mathrm{dg}}^\top z_{\mathrm{dg}} + w_{\mathrm{spu}}^\top z_{\mathrm{spu}}.$$

If we only consider values of $x$ where $w_{\mathrm{dg}}^\top z_{\mathrm{dg}} \neq 0$ and $w_{\mathrm{spu}}^\top z_{\mathrm{spu}} \neq 0$, then without loss of generality we have that

$$w^{*\top}x = w_{\mathrm{dg}}^\top z_{\mathrm{dg}} + w_{\mathrm{spu}}^\top z_{\mathrm{spu}} \neq w_{\mathrm{dg}}^\top z_{\mathrm{dg}}.$$

Given Assumption 1-2,

$$\mu\left(\left\{x \mid w_{\mathrm{dg}}^\top z_{\mathrm{dg}} \neq 0\right\}\right) > 0,$$

the risk of $w_{\mathrm{dg}}^\top z_{\mathrm{dg}}$ is strictly greater than that of $w^{*\top}x$. Equation 13 follows from the strong convexity of the loss. □

## A.2 Proof of Theorem 1—Sufficient Conditions for Well-Specified Domain Generalization Benchmark Splits

Assume $Z_{\mathrm{spu}}^{\mathrm{ID}}$ is sub-Gaussian with mean $\mu_{\mathrm{spu}}$, covariance $\Sigma_{\mathrm{spu}}$, and parameter $\kappa$. Define a nonlinear transformation

$$\phi : \mathbb{R}^l \to \mathbb{R}^l,$$

that is $L_\phi$–Lipschitz, and let

$$Z_{\mathrm{spu}}^{\mathrm{OOD}} = \phi(Z_{\mathrm{spu}}^{\mathrm{ID}}).$$

Assume further that

$$\mathbb{E}[Z_{\mathrm{spu}}^{\mathrm{OOD}}] = \mathbb{E}[\phi(Z_{\mathrm{spu}}^{\mathrm{ID}})] = M\,\mu_{\mathrm{spu}},$$

for some matrix $M \in \mathbb{R}^{l \times l}$. In-distribution, $Z_{\mathrm{spu}}^{\mathrm{ID}} \sim P_{\mathrm{ID}}$ and out-of-distribution, $Z_{\mathrm{spu}}^{\mathrm{OOD}} \sim P_{\mathrm{OOD}}$. Additionally, denote $w_{\mathrm{spu}}$ the contribution of $Z_{\mathrm{spu}}^{\mathrm{ID}}$ to the optimal $P_{\mathrm{ID}}$ predictor $f_X^{P_{\mathrm{ID}}}$. Then, for any $\delta \in (0,1)$, if

$$w_{\mathrm{spu}}^\top(M\,\mu_{\mathrm{spu}}) + \sqrt{2}L_\phi\,\kappa\,\|w_{\mathrm{spu}}\|_2\sqrt{\log(1/\delta)} < 0,$$

(with the understanding that under the Lipschitz assumption the sub-Gaussian property carries over with parameter $L_\phi\,\kappa$), then with probability at least $1 - \delta$ over $Z_{\mathrm{spu}}^{\mathrm{OOD}}$, we have

$$\mathrm{acc}_{P_{\mathrm{OOD}}}(f_X^{P_{\mathrm{ID}}}) < \mathrm{acc}_{P_{\mathrm{OOD}}}(f_{\mathrm{dg}}^{\mathcal{E}}),$$

where $f_{\mathrm{dg}}^{\mathcal{E}}$ and $f_X^{P_{\mathrm{ID}}}$ are the optimal domain–general and domain–specific predictions (Definitions 3–4).

*Proof.* Define

$$Z_{\mathrm{spu}}^{\mathrm{OOD}} = \phi(Z_{\mathrm{spu}}^{\mathrm{ID}}).$$

From Equation 1 and the law of total probability, the out-of-distribution (OOD) accuracy of $f_X^{P_{\mathrm{ID}}}$ is equivalently

$$\mathrm{acc}_{\mathrm{OOD}}(f_X^{P_{\mathrm{ID}}}) = \Pr\left(w_{\mathrm{dg}}^\top Z_{\mathrm{dg}} + w_{\mathrm{spu}}^\top Z_{\mathrm{spu}} > 0\right),$$

and

$$\mathrm{acc}_{\mathrm{OOD}}(f_{\mathrm{dg}}^{\mathcal{E}}) = \Pr\left(w_{\mathrm{dg}}^\top Z_{\mathrm{dg}} > 0\right).$$

It suffices to show that

$$\Pr\left(w_{\mathrm{dg}}^\top Z_{\mathrm{dg}} > -w_{\mathrm{spu}}^\top Z_{\mathrm{spu}}^{\mathrm{OOD}}\right) < \Pr\left(w_{\mathrm{dg}}^\top Z_{\mathrm{dg}} > 0\right) \tag{14}$$

with high probability.

Since $Z_{\text{spu}}^{\text{ID}}$ is sub-Gaussian with parameter $\kappa$, by the Lipschitz property of $\phi$ the random variable $\text{w}_{\text{spu}}^{\top} Z_{\text{spu}}^{\text{OOD}}$ is sub-Gaussian with mean

$$\mathbb{E}\left[\text{w}_{\text{spu}}^{\top} Z_{\text{spu}}^{\text{OOD}}\right] = \text{w}_{\text{spu}}^{\top} \mathbb{E}\left[Z_{\text{spu}}^{\text{OOD}}\right] = \text{w}_{\text{spu}}^{\top}(M\,\mu_{\text{spu}})$$

and sub-Gaussian parameter at most $L_\phi\,\kappa$ (i.e., with variance proxy bounded by $(L_\phi\,\kappa)^2\,\|\text{w}_{\text{spu}}\|_2^2$). Thus, for any $t > 0$,

$$\Pr\left(\text{w}_{\text{spu}}^{\top} Z_{\text{spu}}^{\text{OOD}} > \text{w}_{\text{spu}}^{\top}(M\,\mu_{\text{spu}}) + t\right) \le \exp\left(-\frac{t^2}{2(L_\phi\,\kappa)^2\,\|\text{w}_{\text{spu}}\|_2^2}\right).$$

Choose

$$t = \sqrt{2}L_\phi\kappa)\|\text{w}_{\text{spu}}\|_2\sqrt{\log(1/\delta)}.$$

Then,

$$\Pr\left(\text{w}_{\text{spu}}^{\top} Z_{\text{spu}}^{\text{OOD}} > \text{w}_{\text{spu}}^{\top}(M\,\mu_{\text{spu}}) + t\right) \le \delta.$$

Therefore, with probability at least $1 - \delta$,

$$\text{w}_{\text{spu}}^{\top} Z_{\text{spu}}^{\text{OOD}} < \text{w}_{\text{spu}}^{\top}(M\,\mu_{\text{spu}}) + \sqrt{2}L_\phi\kappa\|\text{w}_{\text{spu}}\|_2\sqrt{\log(1/\delta)}.$$

Assume that

$$\text{w}_{\text{spu}}^{\top}(M\,\mu_{\text{spu}}) + \sqrt{2}L_\phi\,\kappa\|\text{w}_{\text{spu}}\|_2\sqrt{\log(1/\delta)} < 0.$$

Then, with probability at least $1 - \delta$, we have

$$\text{w}_{\text{spu}}^{\top} Z_{\text{spu}}^{\text{OOD}} < 0.$$

In this case,

$$\left\{\text{w}_{\text{dg}}^{\top} Z_{\text{dg}} > -\text{w}_{\text{spu}}^{\top} Z_{\text{spu}}^{\text{OOD}}\right\} \subset \left\{\text{w}_{\text{dg}}^{\top} Z_{\text{dg}} > 0\right\},$$

which implies

$$\Pr\left(\text{w}_{\text{dg}}^{\top} Z_{\text{dg}} > -\text{w}_{\text{spu}}^{\top} Z_{\text{spu}}^{\text{OOD}}\right) < \Pr\left(\text{w}_{\text{dg}}^{\top} Z_{\text{dg}} > 0\right).$$

Equivalently,

$$\begin{aligned}
\text{acc}_{\text{OOD}}(f_X^{P_{\text{ID}}}) &= \Pr\left(\text{w}_{\text{dg}}^{\top} Z_{\text{dg}} + \text{w}_{\text{spu}}^{\top} Z_{\text{spu}}^{\text{OOD}} > 0\right) \\
&= \Pr\left(\text{w}_{\text{dg}}^{\top} Z_{\text{dg}} > -\text{w}_{\text{spu}}^{\top} Z_{\text{spu}}^{\text{OOD}}\right) \\
&< \Pr\left(\text{w}_{\text{dg}}^{\top} Z_{\text{dg}} > 0\right) \\
&= \text{acc}_{\text{OOD}}(f_{\text{dg}}^{\mathcal{E}}).
\end{aligned}$$

Therefore, with probability at least $1 - \delta$,

$$\text{acc}_{\text{OOD}}(f_X^{P_{\text{ID}}}) < \text{acc}_{\text{OOD}}(f_{\text{dg}}^{\mathcal{E}}).$$

$\square$

**Lemma 2** (Monotonicity of 0–1 Accuracy under Symmetry). *Let $W_0$ be a continuous, nondegenerate random variable satisfying*

$$W_0 \overset{d}{=} -W_0,$$

*i.e. centrally symmetric about zero, and assume its CDF $F_{W_0}$ is strictly increasing on $\mathbb{R}$ with $F_{W_0}(0) = \frac{1}{2}$. Fix $p = \Pr(Y = 1) \in (0,1)$, and define the conditional scores*

$$U \mid (Y = 0) = \sigma W_0, \qquad U \mid (Y = 1) = \mu_1 + \sigma W_0,$$

*with $\sigma > 0$ and $\mu_1 \in \mathbb{R}$. Let*

$$r = \frac{\mu_1 - 0}{\sigma} = \frac{\mu_1}{\sigma}.$$

*Then the 0–1 accuracy of the threshold-at-zero classifier $\widehat{Y} = \mathbf{1}\{U > 0\}$ can be written as*

$$acc = p\, F_{W_0}(r) + (1 - p)\, \Pr(W_0 \le 0) = p\, F_{W_0}(r) + (1 - p)\, \tfrac{1}{2},$$

*and this accuracy is a* strictly increasing *function of the signal-to-noise ratio $r$.*

*Proof.* We first define the accuracy decomposition. Writing $\widehat{Y} = 1 \iff U > 0$, we have

$$acc = \Pr(\widehat{Y} = Y) = \Pr(\widehat{Y} = 1, Y = 1) + \Pr(\widehat{Y} = 0, Y = 0)$$
$$= p\, \Pr(U > 0 \mid Y = 1) + (1 - p)\, \Pr(U \le 0 \mid Y = 0).$$

Then we use a change of variables using symmetry. Since $U \mid Y = 1 = \mu_1 + \sigma W_0$,

$$\Pr(U > 0 \mid Y = 1) = \Pr\!\left(W_0 > -\tfrac{\mu_1}{\sigma}\right) = 1 - F_{W_0}(-r) = F_{W_0}(r),$$

and since $U \mid Y = 0 = \sigma W_0$ with $\Pr(W_0 \le 0) = F_{W_0}(0) = \tfrac{1}{2}$,

$$\Pr(U \le 0 \mid Y = 0) = \tfrac{1}{2}.$$

Substituting into the decomposition gives

$$acc = p\, F_{W_0}(r) + (1 - p)\, \tfrac{1}{2} = G(r).$$

Because $F_{W_0}$ is strictly increasing and $p > 0$, $G$ is strictly increasing in $r$. Hence accuracy increases in the signal-to-noise ratio $r = \mu_1/\sigma$. $\qquad\square$

## A.3 Proof of Theorem 2—Necessary and Sufficient Conditions for Well-Specified Domain Generalization Benchmark Splits under Distribution Symmetry

Assume $Z_{\text{spu}}^{\text{ID}}$ is a random variable with mean $\mu_{\text{spu}}$, covariance $\Sigma_{\text{spu}}$, and $\mathrm{w}_{\text{dg}}^\top Z_{\text{dg}}^{\text{OOD}}$ is symmetric about its mean. Define a nonlinear transformation

$$\phi : \mathbb{R}^l \to \mathbb{R}^l,$$

that is $L_\phi$–Lipschitz, and let

$$Z_{\text{spu}}^{\text{OOD}} = \phi(Z_{\text{spu}}^{\text{ID}}).$$

Assume further that

$$\mathbb{E}[Z_{\text{spu}}^{\text{OOD}}] = \mathbb{E}[\phi(Z_{\text{spu}}^{\text{ID}})] = M\, \mu_{\text{spu}}, \qquad \mathrm{Var}(Z_{\text{spu}}^{\text{OOD}}) = \Sigma_\phi, \quad \text{and} \quad Z_{\text{dg}} \perp\!\!\!\perp Z_{\text{spu}} \mid Y$$

for some matrix $M \in \mathbb{R}^{l \times l}$, and that the scalar and $\mathrm{w}_{\text{spu}}^\top Z_{\text{spu}}^{\text{OOD}}$ is symmetric about its mean. Then,

$$acc_{P_{\text{OOD}}}(f_X^{P_{\text{ID}}}) < acc_{P_{\text{OOD}}}(f_{\text{dg}}^{\mathcal{E}}) \iff \frac{\mathrm{w}_{\text{dg}}^\top \mu_{\text{dg}} + \mathrm{w}_{\text{spu}}^\top M \mu_{\text{spu}}}{\sqrt{\mathrm{w}_{\text{dg}}^\top \Sigma_{\text{dg}} \mathrm{w}_{\text{dg}} + \mathrm{w}_{\text{spu}}^\top \Sigma_\phi \mathrm{w}_{\text{spu}}}} < \frac{\mathrm{w}_{\text{dg}}^\top \mu_{\text{dg}}}{\sqrt{\mathrm{w}_{\text{dg}}^\top \Sigma_{\text{dg}} \mathrm{w}_{\text{dg}}}}. \qquad (15)$$

Note that the RHS of Equation 15 implies that either:

- *Spurious Correlation Reversal:* $\mathrm{w}_{\text{spu}}^\top M \mu_{\text{spu}} < 0$ (related to sufficiency condition without the symmetry assumption in Theorem 1), or

- sufficiently large variance when $\mathrm{w}_{\text{spu}}^\top M \mu_{\text{spu}} > 0$, i.e., small SNR.

*Proof.* Directly applying Lemma 2,

$$acc_{P_{\text{OOD}}}(f_X^{P_{\text{ID}}}) < acc_{P_{\text{OOD}}}(f_{\text{dg}}^{\mathcal{E}}) \iff \frac{\mathrm{w}_{\text{dg}}^\top \mu_{\text{dg}} + \mathrm{w}_{\text{spu}}^\top M \mu_{\text{spu}}}{\sqrt{\mathrm{w}_{\text{dg}}^\top \Sigma_{\text{dg}} \mathrm{w}_{\text{dg}} + \mathrm{w}_{\text{spu}}^\top \Sigma_\phi \mathrm{w}_{\text{spu}}}} < \frac{\mathrm{w}_{\text{dg}}^\top \mu_{\text{dg}}}{\sqrt{\mathrm{w}_{\text{dg}}^\top \Sigma_{\text{dg}} \mathrm{w}_{\text{dg}}}}.$$

$\qquad\square$

### A.4 Lemma 4—Accuracy on the Line

**Lemma 3.** *Assume $Z_{spu}^{ID}$ is sub-Gaussian with mean $\mu_{spu}$, covariance $\Sigma_{spu}$, and parameter $\kappa$. Define a nonlinear mapping*

$$\phi : \mathbb{R}^l \to \mathbb{R}^l,$$

*that is $L_\phi$–Lipschitz, and let*

$$Z_{spu}^{OOD} = \phi(Z_{spu}^{ID}).$$

*Assume further that*

$$\mathbb{E}[Z_{spu}^{OOD}] = \mathbb{E}[\phi(Z_{spu}^{ID})] = M\,\mu_{spu}, \ \ and \ \Sigma_\phi = Z_{spu}^{OOD}(Z_{spu}^{OOD})^\top$$

*and that*

$$\|M\,\mu_{spu} - \mu_{spu}\| \le \epsilon_1, \tag{16}$$

$$\left\| w_{spu}^\top \Sigma_\phi^\top\, w_{spu} - w_{spu}^\top \Sigma_{spu}\, w_{spu} \right\| \le \epsilon_2. \tag{17}$$

*Moreover, assume there exists a constant $B > 0$ such that for sufficiently small $t$ (a Tsybakov-type condition),*

$$\Pr\Big( |f_X(X)| \le t \Big) \le Bt.$$

*Then, for any $\delta > 0$, with probability at least $1 - \delta$ over $Z_{spu}^{ID}$, the following holds for any classifier $f_X \in \mathcal{F}$:*

$$\left| acc_P(f_X) - acc_{P_\phi}(f_X) \right| \le B\,\epsilon,$$

*where*

$$\epsilon = \|w_{spu}\|\epsilon_1 + C\sqrt{\log(1/\delta)} + \sqrt{\epsilon_2},$$

*and for some small constant $c > 0$,*

$$C = c\kappa \cdot \max\Big\{ \|w_{spu}\|,\ \|M\| \cdot \|w_{spu}\| \Big\}.$$

*Proof.* Define

$$\Delta(X) = f(X) - f(X'),$$

where $X \sim P$ and $X' \sim P_\phi$, respectively, (i.e. $X'$ is obtained by replacing $Z_{\text{spu}}^{\text{ID}}$ with $Z_{\text{spu}}^{\text{OOD}}$). Since

$$f(X) = \mathrm{w}_{\text{dg}}^\top Z_{\text{dg}} + \mathrm{w}_{\text{spu}}^\top Z_{\text{spu}}^{\text{ID}} \quad \text{and} \quad f(X') = \mathrm{w}_{\text{dg}}^\top Z_{\text{dg}} + \mathrm{w}_{\text{spu}}^\top Z_{\text{spu}}^{\text{OOD}},$$

we have

$$\Delta(X) = \mathrm{w}_{\text{spu}}^\top Z_{\text{spu}}^{\text{ID}} - \mathrm{w}_{\text{spu}}^\top Z_{\text{spu}}^{\text{OOD}} = \mathrm{w}_{\text{spu}}^\top \Big( Z_{\text{spu}}^{\text{ID}} - Z_{\text{spu}}^{\text{OOD}} \Big).$$

We now decompose $\Delta(X)$ into a deterministic part $g(X)$ and a stochastic part $h(X)$:

$$\Delta(X) = \underbrace{\Big[ \mathrm{w}_{\text{spu}}^\top \mathbb{E}[Z_{\text{spu}}^{\text{ID}}] - \mathrm{w}_{\text{spu}}^\top \mathbb{E}[Z_{\text{spu}}^{\text{OOD}}] \Big]}_{g(X)} + \tag{18}$$

$$\underbrace{\Big( (\mathrm{w}_{\text{spu}}^\top Z_{\text{spu}}^{\text{ID}} - \mathbb{E}[\mathrm{w}_{\text{spu}}^\top Z_{\text{spu}}^{\text{ID}}]) - (\mathrm{w}_{\text{spu}}^\top Z_{\text{spu}}^{\text{OOD}} - \mathbb{E}[\mathrm{w}_{\text{spu}}^\top Z_{\text{spu}}^{\text{OOD}}]) \Big)}_{h(X)}. \tag{19}$$

Since

$$\mathbb{E}[Z_{\text{spu}}^{\text{ID}}] = \mu_{\text{spu}} \quad \text{and} \quad \mathbb{E}[Z_{\text{spu}}^{\text{OOD}}] = M\,\mu_{\text{spu}},$$

we have

$$|g(X)| = \left| w_{spu}^\top \mu_{spu} - w_{spu}^\top (M\,\mu_{spu}) \right| \tag{20}$$

$$= \left| w_{spu}^\top (\mu_{spu} - M\,\mu_{spu}) \right| \tag{21}$$

$$\leq \|w_{spu}\| \, \|\mu_{spu} - M\,\mu_{spu}\| \tag{22}$$

$$\leq \|w_{spu}\| \, \epsilon_1. \tag{23}$$

Next, consider the stochastic term $h(X)$. Since $Z_{spu}^{ID}$ is sub-Gaussian with parameter $\kappa$, both $w_{spu}^\top Z_{spu}$ and $w_{spu}^\top Z_{spu}^{OOD}$ are sub-Gaussian with parameters $\kappa\|w_{spu}\|$ and $L_\phi \kappa\|w_{spu}\|$ respectively.

Therefore, for any $t > 0$,

$$\Pr\Big( |w_{spu}^\top Z_{spu}^{ID} - \mathbb{E}[w_{spu}^\top Z_{spu}]| > t \Big) \leq 2\exp\Big( -\frac{t^2}{2(\kappa\|w_{spu}\|)^2} \Big),$$

and

$$\Pr\Big( |w_{spu}^\top Z_{spu}^{OOD} - \mathbb{E}[w_{spu}^\top Z_{spu}^{OOD}]| > t \Big) \leq 2\exp\Big( -\frac{t^2}{2(L_\phi \kappa\|w_{spu}\|)^2} \Big).$$

Applying the union bound, with probability at least $1 - \delta$ we have

$$\left| w_{spu}^\top Z_{spu} - \mathbb{E}[w_{spu}^\top Z_{spu}] \right| + \left| w_{spu}^\top Z_{spu}^{OOD} - \mathbb{E}[w_{spu}^\top Z_{spu}^{OOD}] \right| \leq C\sqrt{\log(1/\delta)},$$

where with a small constant factor $c > 0$,

$$C = c\kappa \cdot \max\Big\{ \|w_{spu}\|, \; L_\phi \cdot \|w_{spu}\| \Big\}.$$

Additionally, by assumption,

$$\left| w_{spu}^\top (M\,\mu_{spu}) - w_{spu}^\top \mu_{spu} \right| \leq \|w_{spu}\| \, \epsilon_1,$$

and

$$\left| w_{spu}^\top (\Sigma_\phi w_{spu}) - w_{spu}^\top (\Sigma_{spu} w_{spu}) \right| \leq \epsilon_2.$$

Combining these bounds, with probability at least $1 - \delta$ we obtain

$$\Delta(X) \leq \|w_{spu}\|\epsilon_1 + C\sqrt{\log(1/\delta)} + \sqrt{\epsilon_2} = \epsilon.$$

Finally, the classifier's accuracy difference is determined by the probability that $f(X)$ and $f(X')$ disagree in sign. Given a Tsybakov-type condition, $\Pr(|f_X(X)| \leq t) \leq Bt$, this probability is controlled by $B\epsilon$. It follow that,

$$\left| \mathrm{acc}_P(f) - \mathrm{acc}_{P_\phi}(f) \right| \leq B\epsilon.$$

This completes the proof. $\qquad\square$

**Lemma 4.** *Assume $Z_{spu}^{ID}$ is sub-Gaussian with mean $\mu_{spu}$, covariance $\Sigma_{spu}$, and parameter $\kappa$. Define a nonlinear mapping*

$$\phi : \mathbb{R}^l \to \mathbb{R}^l,$$

*which is $L_\phi$–Lipschitz, and let*

$$Z_{spu}^{OOD} = \phi(Z_{spu}^{ID}).$$

*Assume further that*

$$\mathbb{E}[Z_{spu}^{OOD}] = \mathbb{E}[\phi(Z_{spu}^{ID})] = M\,\mu_{spu}, \;\; and \;\; \Sigma_\phi = Z_{spu}^{OOD}(Z_{spu}^{OOD})^\top$$

*and that*

$$\|M\,\mu_{spu} - \mu_{spu}\| \leq \epsilon_1, \tag{24}$$

$$\left\| w_{spu}^\top \Sigma_\phi w_{spu} - w_{spu}^\top \Sigma_{spu}\, w_{spu} \right\| \leq \epsilon_2, \tag{25}$$

*where the second inequality is understood to control the difference in the covariance (or concentration) of the spurious features after transformation. Moreover, assume there exists a constant $B > 0$ such that for sufficiently small $t$ (a Tsybakov-type condition),*

$$\Pr\Big(|f_X(X)| \le t\Big) \le Bt,$$

*and there exists $\alpha > 0$ such that*

$$acc_P(f_X) \in [\alpha, 1 - \alpha] \quad and \quad a\, acc_{P_\phi}(f_X) \in [\alpha, 1 - \alpha].$$

*Then for any $\delta \in (0,1)$, with probability at least $1 - \delta$,*

$$\Big|\Phi^{-1}\big(acc_P(f_X)\big) - a\,\Phi^{-1}\big(acc_{P_\phi}(f_X)\big)\Big| \le \widetilde{\epsilon}, \tag{26}$$

*for any classifier $f_X \in \mathcal{F}$, where*

$$\widetilde{\epsilon} = LB\Big(\|w_{spu}\|\epsilon_1 + C\sqrt{\log(1/\delta)} + \sqrt{\epsilon_2}\Big) + \zeta,$$

*with—for a small constant factor $c > 0$—*

$$C = c\kappa \cdot \max\Big\{\|w_{spu}\|,\ \|M\| \cdot \|w_{spu}\|\Big\},$$

*and*

$$\zeta = a|1 - a| \max_{x \in [\alpha, 1-\alpha]}\Big|\Phi^{-1}(x)\Big|,$$

*where $\Phi$ is the Gaussian cumulative distribution function, and $L$ is its Lipschitz constant on $[\alpha, 1 - \alpha]$, i.e., for $p, q \in [\alpha, 1 - \alpha]$,*

$$\Big|\Phi^{-1}(p) - \Phi^{-1}(q)\Big| \le L|p - q|.$$

*Proof.* By Lemma 3, with probability at least $1 - \delta$ we have

$$\Big|acc_P(f_X) - a\, acc_{P_\phi}(f_X)\Big| \le B\Big(\|w_{spu}\|\epsilon_1 + C\sqrt{\log(1/\delta)} + \sqrt{\epsilon_2}\Big).$$

Since $acc_P(f_X)$ and $a\, acc_{P_\phi}(f_X)$ lie in $[\alpha, 1 - \alpha]$, the function $\Phi^{-1}$ is Lipschitz on this interval with constant $L$. Therefore, if we set $p = acc_P(f_X)$ and $q = acc_{P_\phi}(f_X)$, then

$$\Big|\Phi^{-1}(p) - \Phi^{-1}(q)\Big| \le L|p - q|.$$

Taking into account the scaling factor $a$ yields

$$\Big|\Phi^{-1}(p) - a\,\Phi^{-1}(q)\Big| \le L|p - q| + |1 - a|\Big|\Phi^{-1}(q)\Big|.$$

Since $|p - q|$ is bounded by the result of Lemma 3, we obtain

$$\Big|\Phi^{-1}\big(acc_P(f_X)\big) - a\,\Phi^{-1}\big(acc_{P_\phi}(f_X)\big)\Big| \le LB\Big(\|w_{spu}\|\epsilon_1 \tag{27}$$

$$+ C\sqrt{2\log(2/\delta)} + \sqrt{\epsilon_2}\Big) + |1 - a| \max_{x \in [\alpha, 1-\alpha]}\Big|\Phi^{-1}(x)\Big|. \tag{28}$$

Defining the right-hand side as $\widetilde{\epsilon}$ completes the proof. $\square$

## A.5 Lemma 5—Tradeoff Between Accuracy on The Line and Well-Specification

**Lemma 5.** *Assume $Z_{spu}^{ID}$ is sub-Gaussian with mean $\mu_{spu}$, covariance $\Sigma_{spu}$, and parameter $\kappa$. Define a nonlinear mapping*

$$\phi : \mathbb{R}^l \to \mathbb{R}^l,$$

*which is $L_\phi$–Lipschitz, and let*

$$Z_{spu}^{OOD} = \phi(Z_{spu}^{ID}).$$

*Assume further that*

$$\mathbb{E}[Z_{spu}^{OOD}] = \mathbb{E}[\phi(Z_{spu}^{ID})] = M\,\mu_{spu}, \ \ and \ \Sigma_\phi = Z_{spu}^{OOD}(Z_{spu}^{OOD})^\top$$

*and that*

$$\|M\,\mu_{spu} - \mu_{spu}\| \le \epsilon_1, \tag{29}$$

$$\left\| w_{spu}^\top \Sigma_\phi w_{spu} - w_{spu}^\top \Sigma_{spu}\,w_{spu} \right\| \le \epsilon_2. \tag{30}$$

*Fix $w_{spu} \in \mathbb{R}^l$ so that $w_{spu}^\top \mu_{spu} > 0$. Suppose that $M$ satisfies the* spurious correlation reversal condition

$$w_{spu}^\top(M\,\mu_{spu}) + \sqrt{2(L_\phi\kappa)^2\|w_{spu}\|_2^2 \log(1/\delta)} \le -\gamma < 0,$$

*for some margin $\gamma > 0$. Moreover, assume there exists a constant $B > 0$ such that for sufficiently small $t$ (a Tsybakov-type condition),*

$$\Pr\Big(|f_X(X)| \le t\Big) \le Bt,$$

*and that there exists some $\alpha > 0$ such that*

$$acc_P(f_X), \ a\,acc_{P_\phi}(f_X) \in [\alpha, 1 - \alpha].$$

*Then, with probability at least $1 - \delta$,*

$$\left|\Phi^{-1}\big(acc_P(f_X)\big) - a\,\Phi^{-1}\big(acc_{P_\phi}(f_X)\big)\right| \ge C\,\|w_{spu}\|\sqrt{\log(1/\delta)}\,\|M\,\mu_{spu} - \mu_{spu}\| - \zeta, \tag{31}$$

*where*

$$\zeta = |1 - a| \max_{x \in [\alpha, 1-\alpha]}\left|\Phi^{-1}(x)\right|,$$

*and $C$ is a positive constant (depending on $\alpha$ and the local slope of $\Phi^{-1}$, Lipschitzness of $\phi$, and concentration of $Z_{spu}^{OOD}$). Moreover,*

$$\|M\,\mu_{spu} - \mu_{spu}\| \ge \|w_{spu}\|^{-1}\Big(\gamma + w_{spu}^\top \mu_{spu}\Big),$$

*so that the right-hand side of* (31) *is strictly positive whenever $\gamma + w_{spu}^\top \mu_{spu} > 0$.*

*Proof.* Since the spurious correlation reversal condition yields

$$\mathrm{w}_{spu}^\top(M\,\mu_{spu}) \le -\gamma < 0,$$

and since $\mathrm{w}_{spu}^\top \mu_{spu} > 0$, we have

$$\mathrm{w}_{spu}^\top(M\,\mu_{spu} - \mu_{spu}) = \mathrm{w}_{spu}^\top(M\,\mu_{spu}) - \mathrm{w}_{spu}^\top\mu_{spu} \le -\gamma - \mathrm{w}_{spu}^\top\mu_{spu}.$$

By the Cauchy–Schwarz inequality,

$$\|\mathrm{w}_{spu}\|\,\|M\,\mu_{spu} - \mu_{spu}\| \ge \left|\mathrm{w}_{spu}^\top(M\,\mu_{spu} - \mu_{spu})\right| \ge \gamma + \mathrm{w}_{spu}^\top\mu_{spu},$$

so that

$$\|M\,\mu_{spu} - \mu_{spu}\| \ge \frac{\gamma + \mathrm{w}_{spu}^\top\mu_{spu}}{\|\mathrm{w}_{spu}\|}.$$

Next, note that

$$\mathrm{acc}_P(f_X) = \mathrm{Pr}\left(\mathrm{w}_{\mathrm{dg}}^\top Z_{\mathrm{dg}} > -\mathrm{w}_{\mathrm{spu}}^\top Z_{\mathrm{spu}}^{\mathrm{ID}}\right) \quad \text{and} \quad \mathrm{acc}_{P_\phi}(f_X) = \mathrm{Pr}\left(\mathrm{w}_{\mathrm{dg}}^\top Z_{\mathrm{dg}} > -\mathrm{w}_{\mathrm{spu}}^\top Z_{\mathrm{spu}}^{\mathrm{OOD}}\right).$$

Since $\mathrm{w}_{\mathrm{spu}}^\top(M\,\mu_{\mathrm{spu}})$ is very negative relative to the random fluctuations of $\mathrm{w}_{\mathrm{spu}}^\top Z_{\mathrm{spu}}^{\mathrm{OOD}}$ (by the sub-Gaussian concentration inequality with parameter $\kappa$) and since $\mathrm{w}_{\mathrm{spu}}^\top \mu_{\mathrm{spu}} > 0$, one can apply standard concentration arguments to show that with probability at least $1-\delta$

$$\left|\mathrm{acc}_P(f_X) - a\,\mathrm{acc}_{P_\phi}(f_X)\right| \geq C_0 \kappa \,\|\mathrm{w}_{\mathrm{spu}}\| \sqrt{\log(1/\delta)}\,\|M\,\mu_{\mathrm{spu}} - \mu_{\mathrm{spu}}\|$$

for some constant $C_0 > 0$. Since by assumption $\mathrm{acc}_P(f_X)$ and $a\,\mathrm{acc}_{P_\phi}(f_X)$ lie in $[\alpha, 1-\alpha]$, the inverse Gaussian CDF $\Phi^{-1}$ is $L$-Lipschitz on this interval. Thus, we have

$$\left|\Phi^{-1}\big(\mathrm{acc}_P(f_X)\big) - a\,\Phi^{-1}\big(\mathrm{acc}_{P_\phi}(f_X)\big)\right| \geq L\left|\mathrm{acc}_P(f_X) - a\,\mathrm{acc}_{P_\phi}(f_X)\right| - |1-a|\max_{x\in[\alpha,1-\alpha]}\left|\Phi^{-1}(x)\right|$$

$$\geq L\left(C_0\kappa\,\|\mathrm{w}_{\mathrm{spu}}\|\sqrt{\log(1/\delta)}\,\|M\,\mu_{\mathrm{spu}} - \mu_{\mathrm{spu}}\|\right)$$

$$- |1-a|\max_{x\in[\alpha,1-\alpha]}\left|\Phi^{-1}(x)\right|.$$

Defining $C = C_0 L\kappa$ and $\zeta = |1-a|\max_{x\in[\alpha,1-\alpha]}|\Phi^{-1}(x)|$ completes the proof:

$$\left|\Phi^{-1}\big(\mathrm{acc}_P(f_X)\big) - a\,\Phi^{-1}\big(\mathrm{acc}_{P_\phi}(f_X)\big)\right| \geq C\,\|\mathrm{w}_{\mathrm{spu}}\|\sqrt{\log(1/\delta)}\,\|M\,\mu_{\mathrm{spu}} - \mu_{\mathrm{spu}}\| - \zeta.$$

Since $\|M\,\mu_{\mathrm{spu}} - \mu_{\mathrm{spu}}\| \geq \frac{\gamma + \mathrm{w}_{\mathrm{spu}}^\top \mu_{\mathrm{spu}}}{\|\mathrm{w}_{\mathrm{spu}}\|}$, the right-hand side is strictly positive whenever $\gamma + \mathrm{w}_{\mathrm{spu}}^\top\mu_{\mathrm{spu}} > 0$. $\quad\square$

### A.6 Proof of Theorem 3—Benchmarks with Accuracy on the Line are Misspecified Almost Everywhere.

Assume that $Z_{\mathrm{spu}}^{\mathrm{ID}}$ is sub-Gaussian with mean $\mu_{\mathrm{spu}}$, covariance $\Sigma_{\mathrm{spu}}$, and parameter $\kappa$. Define a nonlinear mapping

$$\phi : \mathbb{R}^l \to \mathbb{R}^l,$$

which is $L_\phi$–Lipschitz, and let

$$Z_{\mathrm{spu}}^{\mathrm{OOD}} = \phi(Z_{\mathrm{spu}}^{\mathrm{ID}}).$$

Assume further that

$$\mathbb{E}[Z_{\mathrm{spu}}^{\mathrm{OOD}}] = \mathbb{E}[\phi(Z_{\mathrm{spu}}^{\mathrm{ID}})] = M\,\mu_{\mathrm{spu}},$$

and that

$$\|M\,\mu_{\mathrm{spu}} - \mu_{\mathrm{spu}}\| \leq \epsilon_1, \tag{32}$$

$$\left\|\mathrm{w}_{\mathrm{spu}}^\top \Sigma_\phi\,\mathrm{w}_{\mathrm{spu}} - \mathrm{w}_{\mathrm{spu}}^\top \Sigma_{\mathrm{spu}}\,\mathrm{w}_{\mathrm{spu}}\right\| \leq \epsilon_2. \tag{33}$$

Suppose that $M \in \mathbb{R}^{l\times l}$ satisfies the *spurious correlation reversal condition*

$$\mathrm{w}_{\mathrm{spu}}^\top(M\,\mu_{\mathrm{spu}}) + \sqrt{2\,(L_\phi\,\kappa)^2\,\|\mathrm{w}_{\mathrm{spu}}\|_2^2\,\log(1/\delta)} < 0,$$

and assume that there exists a constant $B > 0$ such that for sufficiently small $t$ (a Tsybakov-type condition),

$$\mathrm{Pr}\Big(|f_X(X)| \leq t\Big) \leq Bt,$$

and there exists some $\alpha > 0$ such that

$$\mathrm{acc}_P(f_X),\ a\,\mathrm{acc}_{P_\phi}(f_X) \in [\alpha, 1-\alpha],$$

where $\mathrm{acc}_{P_\phi}(f_X)$ is the accuracy when the out-of-distribution features are given by $\phi(Z_{\mathrm{spu}}^{\mathrm{ID}})$, and $\Phi$ denotes the Gaussian cumulative distribution function.

Define

$$\mathcal{W}_\epsilon = \left\{ M \in \mathbb{R}^{l \times l} : \begin{array}{l} \mathrm{w}_{\mathrm{spu}}^\top (M\,\mu_{\mathrm{spu}}) + \sqrt{2\,(L_\phi\,\kappa)^2\,\|\mathrm{w}_{\mathrm{spu}}\|_2^2\,\log(1/\delta)} < 0, \\ \quad and \\ \left| \Phi^{-1}\big(\mathrm{acc}_P(f_X)\big) - a\,\Phi^{-1}\big(\mathrm{acc}_{P_\phi}(f_X)\big) \right| \le \epsilon \end{array} \right\}. \tag{34}$$

Then:

   (i) $\mathcal{W}_0$ has Lebesgue measure zero in $\mathbb{R}^{l \times l}$.

   (ii) For any $0 \le \epsilon_i \le \epsilon_j$, we have $\mathcal{W}_{\epsilon_i} \subseteq \mathcal{W}_{\epsilon_j}$.

In particular, as $\epsilon \to 0$ (i.e., perfect accuracy on the line), almost every shift is misspecified, and the Lebesgue measure of the set of well–specified shifts grows monotonically with $\epsilon$.

*Proof.* From Lemma 4 have the inequality

$$\left| \Phi^{-1}\big(\mathrm{acc}_P(f_X)\big) - a\,\Phi^{-1}\big(\mathrm{acc}_{P_\phi}(f_X)\big) \right| \ge C\Big( \|\mathrm{w}_{\mathrm{spu}}\| \sqrt{\log(1/\delta)}\, \|M\mu_{\mathrm{spu}} - \mu_{\mathrm{spu}}\| \Big) - |1-a| \max_{x \in [\alpha, 1-\alpha]} \left| \Phi^{-1}(x) \right|,$$

where $C$ is a positive constant (depending on the concentration bounds), and $L$ is the Lipschitz constant of $\Phi^{-1}$ on $[\alpha, 1-\alpha]$. Suppose

$$\left| \Phi^{-1}\big(\mathrm{acc}_P(f_X)\big) - a\,\Phi^{-1}\big(\mathrm{acc}_{P_\phi}(f_X)\big) \right| \le \epsilon,$$

then for $\epsilon$ sufficiently small, it must be that

$$\|M\mu_{\mathrm{spu}} - \mu_{\mathrm{spu}}\| \le \frac{\epsilon + |1-a| \max_{x \in [\alpha, 1-\alpha]} \left| \Phi^{-1}(x) \right|}{C\,\|\mathrm{w}_{\mathrm{spu}}\| \sqrt{\log(1/\delta)}}. \tag{35}$$

Thus, as $\epsilon \to 0$ we must have

$$\|M\mu_{\mathrm{spu}} - \mu_{\mathrm{spu}}\| = 0,$$

i.e.,

$$M\mu_{\mathrm{spu}} = \mu_{\mathrm{spu}} \implies \mathrm{w}_{\mathrm{spu}}^\top (M\mu_{\mathrm{spu}}) = \mathrm{w}_{\mathrm{spu}}^\top \mu_{\mathrm{spu}} \ge 0.$$

The second equality follows from $\mathrm{w}_{\mathrm{spu}}$ being the optimal contribution of $Z_{\mathrm{spu}}^{\mathrm{ID}}$ to $f_X^{P_{\mathrm{ID}}}$.

Let

$$S = \{ M \in \mathbb{R}^{l \times l} : M\mu_{\mathrm{spu}} = \mu_{\mathrm{spu}} \}.$$

Since $\mu_{\mathrm{spu}} \ne 0$, $S$ is an affine subspace of $\mathbb{R}^{l \times l}$ with dimension strictly less than $l^2$ and hence has Lebesgue measure zero (Sard's theorem; Sard (1942)). Since

$$\mathcal{W}_0 \subset S,$$

it follows that $\mathcal{W}_0$ has Lebesgue measure zero.

The monotonicity claim follows immediately from Equations 34-35: if $0 \le \epsilon_i \le \epsilon_j$, then by definition

$$\mathcal{W}_{\epsilon_i} \subseteq \mathcal{W}_{\epsilon_j}.$$

This completes the proof. □

## A.7 Example of Shifts with Accuracy on the Line that are Well-Specified

Let $Z_{\mathrm{spu}} \in \mathbb{R}^k$ be Gaussian with mean $\mu_{\mathrm{spu}} = \mathbb{E}[Z_{\mathrm{spu}}] \neq 0$, and covariance $\Sigma_{\mathrm{spu}}$. Fix $\mathrm{w}_{\mathrm{spu}} \in \mathbb{R}^k$ with $\mathrm{w}_{\mathrm{spu}}^\top \mu_{\mathrm{spu}} \neq 0$ and let $a > 0$. Assume the mapping is linear, i.e. $\phi(u) = M\,u$, so that

$$Z_{\mathrm{spu}}^{\mathrm{OOD}} = \phi(Z_{\mathrm{spu}}^{\mathrm{ID}}) = M\,Z_{\mathrm{spu}}^{\mathrm{ID}}.$$

Then for any $\epsilon > 0$ and $\delta \in (0,1)$, there exists a matrix $M \in \mathbb{R}^{k \times k}$ such that:

$$\mathrm{w}_{\mathrm{spu}}^\top M \mu_{\mathrm{spu}} + \sqrt{2\,\mathrm{w}_{\mathrm{spu}}^\top M \Sigma_{\mathrm{spu}} M^\top \mathrm{w}_{\mathrm{spu}}\, \log(1/\delta)} < 0, \tag{36}$$

$$\left| \Phi^{-1}\big(\mathrm{acc}_P(f_{\mathrm{X}})\big) - a\,\Phi^{-1}\big(\mathrm{acc}_{P_M}(f_{\mathrm{X}})\big) \right| \leq \epsilon, \tag{37}$$

with probability at least $1 - \delta$, where $\mathrm{acc}_P(f_{\mathrm{X}})$ and $\mathrm{acc}_{P_M}(f_{\mathrm{X}})$ are defined as in Lemma 4.

For the construction, let

$$v = \frac{\mathrm{w}_{\mathrm{spu}}}{\|\mathrm{w}_{\mathrm{spu}}\|}$$

be the unit vector in the direction of $\mathrm{w}_{\mathrm{spu}}$, and define its reflection matrix

$$R = I - 2vv^\top.$$

Note that $Rv = -v$ and $R^2 = I$. Choose a scalar $\alpha > 0$ and define

$$M = \alpha R.$$

Then, we compute:

$$\mathrm{w}_{\mathrm{spu}}^\top(M\mu_{\mathrm{spu}}) = \mathrm{w}_{\mathrm{spu}}^\top(\alpha R\mu_{\mathrm{spu}}) = \alpha\,(\mathrm{w}_{\mathrm{spu}}^\top R\mu_{\mathrm{spu}}) = -\alpha\,\mathrm{w}_{\mathrm{spu}}^\top\mu_{\mathrm{spu}},$$

$$\mathrm{w}_{\mathrm{spu}}^\top(M\Sigma_{\mathrm{spu}}M^\top)\mathrm{w}_{\mathrm{spu}} = \alpha^2\,\mathrm{w}_{\mathrm{spu}}^\top(R\Sigma_{\mathrm{spu}}R^\top)\mathrm{w}_{\mathrm{spu}} = \alpha^2\,\mathrm{w}_{\mathrm{spu}}^\top\Sigma_{\mathrm{spu}}\mathrm{w}_{\mathrm{spu}},$$

since $R\mathrm{w}_{\mathrm{spu}} = -\mathrm{w}_{\mathrm{spu}}$ and $R$ is orthogonal.

The spurious correlation reversal condition (36) becomes

$$-\alpha\,\mathrm{w}_{\mathrm{spu}}^\top\mu_{\mathrm{spu}} + \sqrt{2\,\alpha^2\,\mathrm{w}_{\mathrm{spu}}^\top\Sigma_{\mathrm{spu}}\mathrm{w}_{\mathrm{spu}}\,\log(1/\delta)} < 0.$$

This can be written as

$$\alpha\left(-\mathrm{w}_{\mathrm{spu}}^\top\mu_{\mathrm{spu}} + \alpha\,\sqrt{2\,\mathrm{w}_{\mathrm{spu}}^\top\Sigma_{\mathrm{spu}}\mathrm{w}_{\mathrm{spu}}\,\log(1/\delta)}\right) < 0.$$

In particular, since $\mathrm{w}_{\mathrm{spu}}^\top\mu_{\mathrm{spu}} > 0$, it suffices to choose $\alpha$ such that

$$\alpha > \frac{\sqrt{2\,\mathrm{w}_{\mathrm{spu}}^\top\Sigma_{\mathrm{spu}}\mathrm{w}_{\mathrm{spu}}\,\log(1/\delta)}}{\mathrm{w}_{\mathrm{spu}}^\top\mu_{\mathrm{spu}}}.$$

At the same time, we want the errors induced by $M$ to be small. Define the following error terms:

$$\epsilon_1 = \|M\mu_{\mathrm{spu}} - \mu_{\mathrm{spu}}\| = \|\alpha R\mu_{\mathrm{spu}} - \mu_{\mathrm{spu}}\|,$$

and

$$\epsilon_2 = |\alpha^2 - 1| \cdot \left|\mathrm{w}_{\mathrm{spu}}^\top\Sigma_{\mathrm{spu}}\mathrm{w}_{\mathrm{spu}}\right|.$$

We want to choose $\alpha$ close to 1 (so that $\epsilon_1$ and $\epsilon_2$ are small) while also satisfying the above inequality. Hence, we set

$$\alpha = \max\left\{1 + \eta,\ \frac{\sqrt{2\,\mathrm{w}_{\mathrm{spu}}^\top\Sigma_{\mathrm{spu}}\mathrm{w}_{\mathrm{spu}}\,\log(1/\delta)}}{\mathrm{w}_{\mathrm{spu}}^\top\mu_{\mathrm{spu}}}\right\},$$

for some small $\eta > 0$ chosen so that $|\alpha - 1|$ is below the desired threshold. By choosing $\alpha$ accordingly, we ensure that:

1. The spurious correlation reversal condition (36) holds.

2. The induced errors $\epsilon_1$ and $\epsilon_2$ are small enough so that, by Lemma 4, we have

$$\left| \Phi^{-1}(\text{acc}_P(f_X)) - a\,\Phi^{-1}(\text{acc}_{P_M}(f_X)) \right| \leq B\Big( \|\mathbf{w}_{\text{spu}}\|\epsilon_1 + C\sqrt{2\log(2/\delta)} + \sqrt{\epsilon_2} \Big) \leq \epsilon.$$

Thus, $M = \alpha R$ satisfies both conditions (36) and (37) with probability at least $1 - \delta$. Note, however, that the set of such $M$ has Lebesgue measure zero in $\mathbb{R}^{l \times l}$.

## B  Simulation Experiment Setup

**Simulation Experiments.**  We evaluate our results so far empirically. We define an initial distribution with $Z_{\text{dg}} \in \mathbb{R}^2$ as a Gaussian with mean $Y \cdot \mu_{\text{dg}}$, where $\mu_{\text{dg}} = [1; 1]$ and unit variance, and $Z_{\text{spu}}^{\text{ID}} \in \mathbb{R}^2$ as a Gaussian with mean $Y \cdot \mu_{\text{spu}}$, where $\mu_{\text{spu}} = [1; 1]$ and unit variance. The input $X \in \mathbb{R}^4$ and label $y \in \{0, 1\}$. We define a domain by $M$ where $Z_{\text{spu}}^{\text{OOD}} = M Z_{\text{spu}}^0$ and all other random variables' distribution is preserved. We consider settings where the training domain is (i) Gaussian and (ii) Sub-Gaussian (mixture of Gausians). We define a set of 50 Gaussian test domains defined by randomly sampled $M$.

We train two types of classifiers: *domain general*, which are logistic regression classifiers trained and evaluated with only $Z_{\text{dg}}$ features but still trained only on the training distribution, and *domain specific*, which are logistic regression classifiers trained an evaluated with $X$ features but still trained only on the training distribution. Details on the experiments can be found in Appendix B.

Figure 2a demonstrates the setting where the training domain is defined by $M = I_{[2]}$, i.e., $Z_{\text{spu}}^{\text{ID}}$ is a multivariate Gaussian. In this setting, we observe the expected behavior derived in Theorem 1. That is, when the spurious correlation reversal and controlled spurious feature variance hold out-of-distribution, the domain-general classifiers outperform the domain-specific classifiers.

Figure 2 demonstrates the setting where the training domain is a mixture of $M$'s, i.e., a mixture of Gaussians making a Sub-Gaussian distribution. Figure 2c demonstrates the setting where $M$ is unconstrained. Here, the test domains can be written as an interpolation of the training domains, i.e., there are positive and negative definite $M$'s mixed to create the training domain. In this setting, Rosenfeld et al. Rosenfeld et al. (2022b) show that the training domain empirical risk minimizer solves the worst-case domain generalization problem. Indeed, figure 2c's results show that there is not generally a difference in OOD performance between the domain-general and domain-specific classifiers.

Figure 2c demonstrates the setting where the testing domains are not a convex combination of the training domain – test domains can be outside the bounds of the training $M$'s. Here, there is a clear difference in the OOD performance between the domain-general and domain-specific classifiers. Furthermore, the expected conditions derived in Theorem 1 are observed. That is, when the spurious correlation reversal and controlled spurious feature variance hold out-of-distribution, the domain-general classifiers outperform the domain-specific classifiers.

Clearly, in natural datasets, it is often impractical to conduct such experiments to determine when a domain-general classifier achieves the best transfer accuracy on a benchmark. Typically, the domain-general features are unknown, and we lack the ability to manipulate natural datasets. However, we demonstrate below that the absence of a strong positive correlation between in- and out-of-distribution accuracy for arbitrary predictors—referred to as accuracy on the line (Miller et al., 2021), Definition 6—can identify well-specified benchmarks that reliably evaluate domain generalization via transfer accuracy. *We will show that well-specified domain generalization benchmarks exhibit either weak in- and out-of-distribution accuracy correlation or a strong inverse correlation.*

*Parameters.*  We use the following parameters across our experiments: $\mu = [1, 1]$, $\Sigma_{\text{dg}} = \text{diag}([1, 1])$, $\mu_{\text{spu}} = [1, 1]$. We expect our results to hold independent of these parameters. We chose these parameters for the ease of intuition of the results on the simulated dataset. We use a sample size of 1000 for each domain.

---

**Algorithm 1:** Generative Mechanism for ColoredMNIST

---

**Input** : MNIST dataset with grayscale images $z_{\mathrm{dg}}$ and binary labels $y \in \{0, 1\}$
**Output:** ColoredMNIST dataset with colorized images $x$ and labels $y$
Define color mapping probability $P(z_{\mathrm{spu}}|y)$ based on a chosen spurious correlation
Sample $y \sim P(y)$ from the original MNIST dataset
Sample grayscale image $z_{\mathrm{dg}}$ corresponding to $y$
`// Introduce spurious correlation`
With probability $p$, assign color $z_{\mathrm{spu}}$ based on $P(z_{\mathrm{spu}}|y)$
With probability $1 - p$, assign color $z_{\mathrm{spu}}$ randomly (breaking correlation)
Apply color transformation $T(z_{\mathrm{dg}}, z_{\mathrm{spu}})$ to obtain $x$
**return** $(x, y)$

---

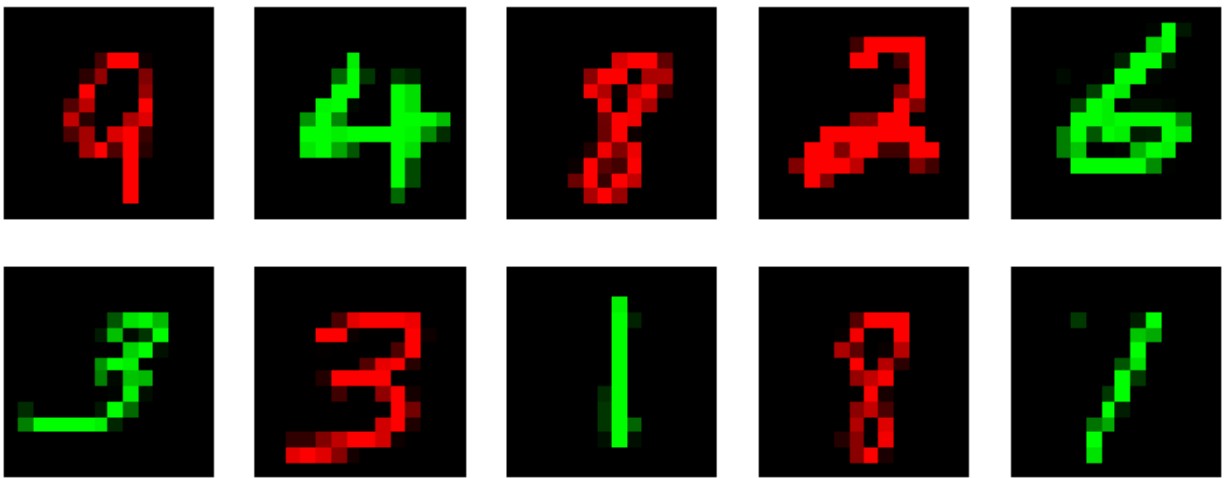

Figure 6: Colored MNIST image examples

$$P_M = \begin{cases} Y & = \mathrm{Bern}(0.5) \\ Z_{\mathrm{dg}} & = \mathcal{N}\big(Y \cdot \mu_{\mathrm{dg}}, \Sigma_{\mathrm{dg}}\big) \\ Z_{\mathrm{spu}} & = \mathcal{N}\big(Y \cdot M\mu_{\mathrm{dg}}, M\Sigma_{\mathrm{spu}}M^\top\big) \end{cases} \tag{38}$$

In Figure 2a, We pick a $P_I$ as our train domain and then randomly sample $M$'s to construct $P_M$'s. We then train a domain-specific logistic regression classifier for $f_X : \mathcal{X} \mapsto \mathcal{Y}$ and a domain-general logistic regression classifier for $f_{\mathrm{dg}} : \mathcal{Z}_{\mathrm{dg}} \mapsto \mathcal{Y}$ for $P_I$. We retrain each classifier

### B.1 ColoredMNIST Case Study

The ColoredMNIST dataset (Arjovsky et al., 2019) intuitively illustrates the complexity of benchmarking domain generalization. ColoredMNIST modifies the gray-scale MNIST (Deng, 2012) dataset by adding color as a spurious correlation. The digits labels are binary with $+1$ when 'digit $\geq 5$' and $-1$ otherwise. The observed (training) labels, however, contain 25% label noise, i.e., a predictor that uses digit information can achieve 75% accuracy at most, in/out-of-distribution. Additionally, the digit images are colored. The color of the digit matches the noisy observed labels with probability $p_e$, inducing a *spurious correlation* or *shortcut* of strength $p_e$. $p_e$ defines a distinct distribution.

Since the observed labels are noisy versions of the true digit labels, the color potentially correlates more with the observed labels than the digit itself. For example, consider a training domain where $p_e^i = P_i(Y = +1 \mid$

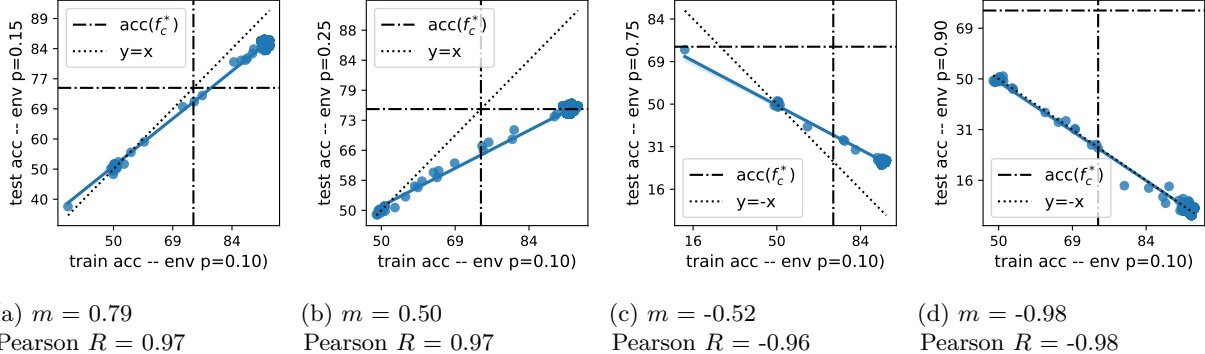

(a) $m = 0.79$
Pearson $R = 0.97$

(b) $m = 0.50$
Pearson $R = 0.97$

(c) $m = $ -0.52
Pearson $R = $ -0.96

(d) $m = $ -0.98
Pearson $R = $ -0.98

Figure 7: Correlations between classifier performance In-Distribution vs. Out-of-Distribution on ColoredM-NIST variations. $m$ is the slope of the line, and $R$ is the Pearson correlation coefficient. The axis-parallel dashed lines denote the maximum within-domain accuracy of 75%, and $y = x$ represents invariant performance across training and test (target) domains. Classifiers achieving above 75% accuracy use color as a predictor. Figures 7a and 7b represent shifts where color-based predictors achieve the highest OOD accuracy—above 75% accuracy. Without domain knowledge, one might conclude that the best ERM solution is the most domain-general. However, Figures 7c and 7d show that these classifiers are not domain-general; some features that improve ID accuracy hurt OOD performance.

color $= +1) = 0.9$. A color-based predictor would achieve 90% accuracy in-domain, while a domain-general predictor that ignores color would achieve 75% accuracy at a maximum. Furthermore, under a shift where $p_e^j = P_j(Y = 1 \mid \text{color} = +1) > 0.75$, a color-based classifier trained on $p_e^i$ classifier will still outperform the domain-general classifier in OOD accuracy. However, when $p_e^j = P_j(Y = 1 \mid \text{color} = +1) < 0.75$, the same color-based classifier will transfer worse than the domain-general classifier. This simple example underscores that for a domain generalization benchmark to be well-specified, with respect to OOD accuracy, spurious correlations from training to test domains must change enough for the domain-general classifier to achieve the highest possible OOD accuracy.

Figure 7 demonstrates that domain-general classifiers need not transfer the best OOD. To demonstrate this, we test a set of classifiers on a ColoredMNIST training domain where $P_{\text{tr}}(Y = 1 \mid \text{color} = \text{green}) = 0.1$ and across various test domains with $P_{\text{te}}(Y = 1 \mid \text{color} = \text{green}) = p_e^j$. The observation of the variance in the domain-general and color-based classifier transfer gap in Figure 7 underscores this work's key question on which ID-OOD shifts allow for reliable domain-generalization evaluation.

We leverage a ConvNet architecture for the ColoredMNIST dataset (Table 4); we vary hyperparameters enumerated in (Gulrajani & Lopez-Paz, 2020). We vary the hyperparameters in Table 2 and whether or not we use data augmentation.

## C  Additional Results and Discussion

### C.1  Model Training

**Data Augmentation.**  When data augmentation is applied, the transformation consists of a series of preprocessing steps applied to images before they are used for training. First, the image undergoes a **random resized crop** to a size of $224 \times 224$ pixels, with a scaling factor ranging from 70% to 100% of the original size. Next, a **random horizontal flip** is applied to introduce variability in orientation. The transformation also includes **color jittering**, which adjusts brightness, contrast, saturation, and hue with a factor of 0.3 each, followed by **random grayscale conversion**, which randomly turns images into grayscale with a certain probability.

Table 2: Models Generation

| Hyperparameter | Range |
|---|---|
| Learning Rate ($lr$) | $10^{-5}$ to $10^{-3.5}$ |
| Weight Decay | $10^{-6}$ to $10^{-2}$ |
| Batch Size | $2^3$ (8) to $2^{5.5}$ ($\approx 45$) |
| Data Augmentation | {True, False} |
| Transfer Learning | {True, False} |
| Model Architecture | {ResNet18, ResNet50, DenseNet121, ViT-B-16, and ConvNeXt_Tiny} |
| Dropout | $\{0.0, 0.1, 0.5\}$ |
| Epoch | – |

**Experimental Setup.** We follow the following general experimental procedure. When experiments deviate from this, it is specified in their respective sections.

Each dataset consists of $E$ domains, each corresponding to a unique data distribution. Our experiments involve ID/OOD splits using a leave-one-domain-out approach. Specifically, for each domain indexed as $i \in [1..E]$, we train on the subset $\mathcal{E}^i_{\text{train}} = \{e_1, \ldots, e_{i-1}, e_{i+1}, \ldots, e_E\}$ and test on the held-out domain $\mathcal{E}^i_{\text{test}} = \{e_i\}$.

For each $i$, we train the following classifiers on $P^{\mathcal{E}^i_{\text{train}}}$: ResNet18, ResNet50, DenseNet121, and ConvNeXt_Tiny. For each classifier, we consider ImageNet pretrained variants: (i) Fine-tuned – end-to-end training on $P^{\mathcal{E}^i_{\text{train}}}$ and (ii) Transfer learning – retraining only the last layer on $P^{\mathcal{E}^i_{\text{train}}}$. classifiers are generated with hyperparameters in Table 2; we also take classifiers at different checkpoints during training.

Table 3: Dataset environments with minimum samples such that change in ID and OOD accuracies correlation changes less than 1% with new models, and total samples included. We do this for each dataset and ID/OOD split. Our analysis accounts for sampling error and ensures this threshold is met with 95% confidence, determined by bootstrapping with 1000 resamplings.

| Dataset | OOD | Minimum Samples | Total Samples |
|---|---|---|---|
| CivilComments | Env 0 | 6,710 | 10,350 |
| CivilComments | Env 1 | 7,210 | 10,350 |
| CivilComments | Env 2 | 7,610 | 10,350 |
| CivilComments | Env 3 | 10,310 | 10,350 |
| CivilComments | Env 4 | 6,210 | 10,350 |
| CivilComments | Env 5 | 9,810 | 10,350 |
| CivilComments | Env 6 | 9,810 | 10,350 |
| CivilComments | Env 7 | 10,310 | 10,350 |
| CivilComments | Env 8 | 10,310 | 10,350 |
| CivilComments | Env 9 | 4,910 | 10,350 |
| CivilComments | Env 10 | 5,910 | 10,350 |
| CivilComments | Env 11 | 5,510 | 10,350 |
| CivilComments | Env 12 | 4,810 | 10,350 |
| CivilComments | Env 13 | 5,610 | 10,350 |
| CivilComments | Env 14 | 2,510 | 10,350 |
| CivilComments | Env 15 | 5,910 | 10,350 |
| ColoredMNIST | Env 0 | 14,110 | 19,758 |
| ColoredMNIST | Env 1 | 3,110 | 20,130 |
| ColoredMNIST | Env 2 | 1,810 | 20,020 |
| Covid-CXR | Env 0 | 3,410 | 7,140 |
| Covid-CXR | Env 1 | 1,410 | 7,140 |
| Covid-CXR | Env 2 | 3,310 | 7,140 |

Table 3 continued from previous page

| Dataset | OOD | Minimum Samples | Total Samples |
|---|---|---|---|
| Covid-CXR | Env 3 | 4,110 | 7,140 |
| Covid-CXR | Env 4 | 4,610 | 7,140 |
| PACS | Env 0 | 2,310 | 4,407 |
| PACS | Env 1 | 3,110 | 4,761 |
| PACS | Env 2 | 910 | 5,043 |
| PACS | Env 3 | 1,210 | 4,806 |
| SpawriousM2M Easy | Env 0 | 1,310 | 6,630 |
| SpawriousM2M Easy | Env 1 | 3,010 | 6,630 |
| SpawriousM2M Easy | Env 2 | 1,610 | 6,732 |
| SpawriousM2M Hard | Env 0 | 5,610 | 5,916 |
| SpawriousM2M Hard | Env 1 | 2,910 | 5,916 |
| SpawriousM2M Hard | Env 2 | 2,910 | 5,916 |
| SpawriousO2O Easy | Env 0 | 4,810 | 6,630 |
| SpawriousO2O Easy | Env 1 | 5,610 | 6,630 |
| SpawriousO2O Easy | Env 2 | 5,610 | 6,630 |
| SpawriousO2O Hard | Env 0 | 4,810 | 5,916 |
| SpawriousO2O Hard | Env 1 | 4,310 | 5,916 |
| SpawriousO2O Hard | Env 2 | 5,910 | 5,916 |
| Terra Incognita | Env 0 | 910 | 3,186 |
| Terra Incognita | Env 1 | 1,410 | 3,186 |
| Terra Incognita | Env 2 | 1,010 | 3,135 |
| Terra Incognita | Env 3 | 1,710 | 3,165 |
| WILDS Camelyon | Env 0 | 1,610 | 8,400 |
| WILDS Camelyon | Env 1 | 2,010 | 9,228 |
| WILDS Camelyon | Env 2 | 3,110 | 7,408 |
| WILDS Camelyon | Env 3 | 2,810 | 8,548 |
| WILDS Camelyon | Env 4 | 4,710 | 7,480 |
| WILDS FMoW | Env 0 | 810 | 27,340 |
| WILDS FMoW | Env 1 | 2,310 | 27,245 |
| WILDS FMoW | Env 2 | 710 | 28,725 |
| WILDS FMoW | Env 3 | 4,110 | 25,550 |
| WILDS FMoW | Env 4 | 4,310 | 23,930 |
| WILDS FMoW | Env 5 | 7,010 | 26,540 |
| WILDS Waterbirds | Env 0 | 2,010 | 8,823 |
| WILDS Waterbirds | Env 1 | 510 | 9,027 |

## C.2 ColoredMNIST

**ColoredMNIST (Arjovsky et al., 2019).** A variant of the MNIST handwritten digit classification dataset (LeCun, 1998). Domain $d \in \{0.1, 0.2, 0.9\}$ contains a disjoint set of digits colored either red or blue. The label is a noisy function of the digit and color, such that color bears a correlation of $d$ with the label and the digit bears a correlation of 0.75 with the label. This dataset contains 70,000 examples of dimension (2, 28, 28) and 2 classes.

**Experimental Details.** We leverage a ConvNet architecture for the ColoredMNIST dataset (Table 4).

Table 4: MNIST ConvNet architecture.

| # | Layer |
|---|---|
| 1 | Conv2D (in=d, out=64) |
| 2 | ReLU |
| 3 | GroupNorm (groups=8) |
| 4 | Conv2D (in=64, out=128, stride=2) |
| 5 | ReLU |
| 6 | GroupNorm (groups=8) |
| 7 | Conv2D (in=128, out=128) |
| 8 | ReLU |
| 9 | GroupNorm (groups=8) |
| 10 | Conv2D (in=128, out=128) |
| 11 | ReLU |
| 12 | GroupNorm (8 groups) |
| 13 | Global average-pooling |

**Discussion.** Despite colored MNIST's apparent simplicity, the spurious correlation between color and the label is quite strong—particularly generalization to text environment 2, going from domains with spurious correlation probability of $0.1, 0.2 \to 0.9$. in Gulrajani & Lopez-Paz (2020)'s evaluation of standard domain generalization methods at the time, they found that no model could mitigate the effect of this spurious correlation. We note that this ID/OOD split has a strong accuracy on the inverse line. In test environment 1, we observe that the training distributions are such that the spurious correlations cancel out (0.1 vs. 0.9), and the domain-general model is also the best ID empirical risk minimizer.

Knowledge of the spurious correlation mechanism in each domain makes it relatively easy to identify the type of features a model uses due to the predictability of expected accuracy between models that use color and those that don't. Due to the potential ambiguity of benchmarking results when spurious correlation mechanisms are unknown, semisynthetic benchmarks are vital in the evaluation process.

Table 5: ColoredMNIST ID vs. OOD properties.

| OOD | slope | intercept | Pearson R | p-value | standard error |
|---|---|---|---|---|---|
| Env 0 acc | 1.90 | -0.58 | 0.82 | 0.00 | 0.01 |
| Env 1 acc | 0.96 | 0.01 | 0.94 | 0.00 | 0.00 |
| Env 2 acc | -1.56 | 0.47 | -0.74 | 0.00 | 0.01 |

Table 6: ColoredMNIST ID vs. OOD properties.

| OOD | ID | slope | intercept | Pearson R | p-value | standard error |
|---|---|---|---|---|---|---|
| Env 0 acc | Env 1 acc | 1.23 | -0.12 | 0.99 | 0.00 | 0.00 |
| Env 0 acc | Env 2 acc | -0.73 | 0.87 | -0.36 | 0.00 | 0.02 |
| Env 1 acc | Env 0 acc | 0.91 | 0.04 | 0.98 | 0.00 | 0.00 |
| Env 1 acc | Env 2 acc | 0.67 | 0.16 | 0.72 | 0.00 | 0.01 |
| Env 2 acc | Env 0 acc | -1.34 | 0.53 | -0.82 | 0.00 | 0.01 |
| Env 2 acc | Env 1 acc | -1.65 | 0.29 | -0.64 | 0.00 | 0.02 |

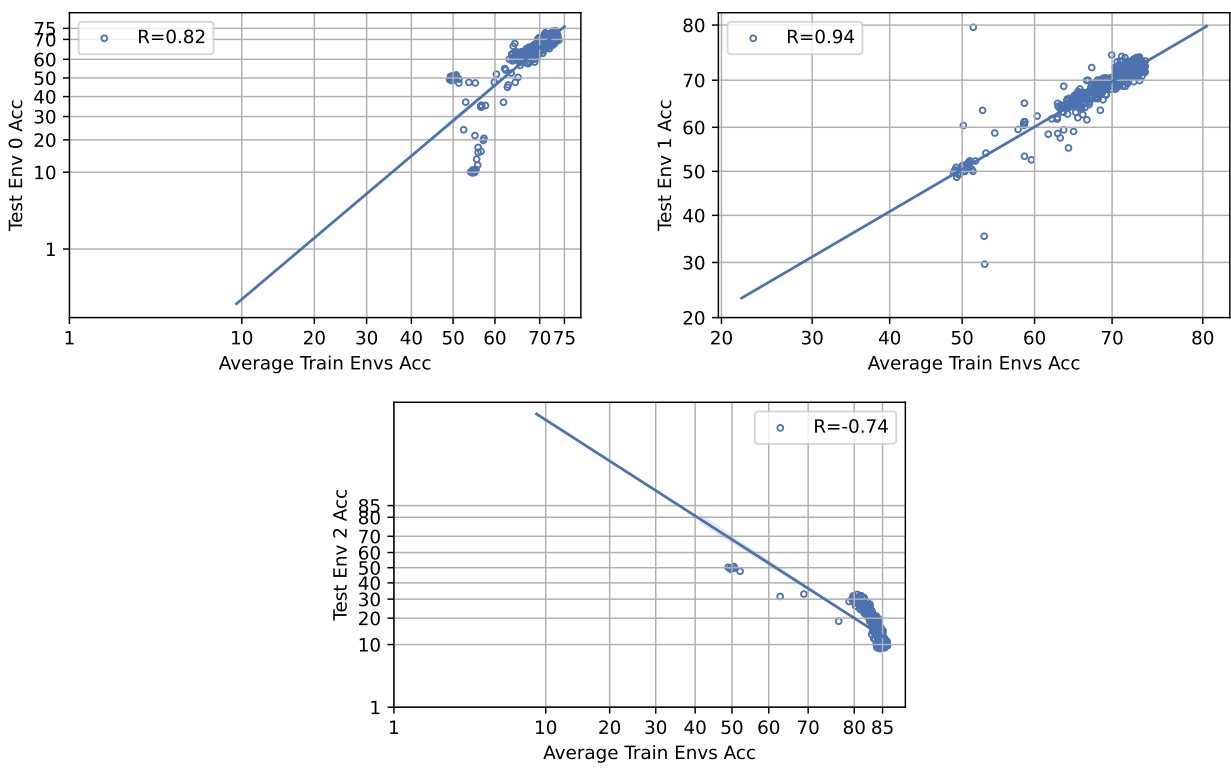

Figure 8: ColoredMNIST: Average train Env Accuracy vs. Test Env Accuracy.

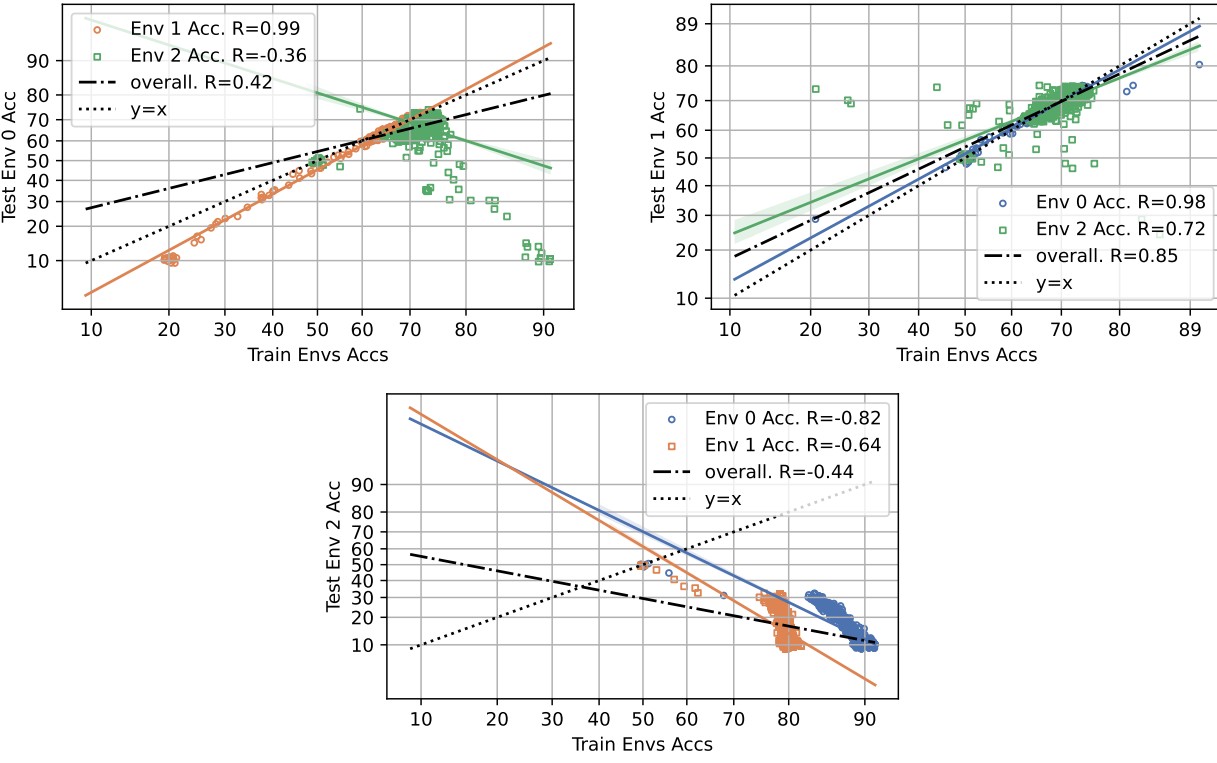

Figure 9: ColoredMNIST: Train Env Accuracy vs. Test Env Accuracy.

### C.3 Spawrious

**Spawrious (Lynch et al., 2023).** The Spawrious image classification benchmark suite consists of six different datasets, including one-to-one (O2O) spurious correlations, where a single spurious attribute correlates with a binary label, and many-to-many (M2M) spurious correlations across multiple classes and spurious attributes. Each benchmark task is proposed with three difficulty levels: Easy, Medium, and Hard. The dataset contains images of four dog breeds $c \in \{bulldog, dachshund, labrador, corgi\}$ found in six backgrounds $b \in \{beach, desert, dirt, jungle, mountain, sand\}$. Images are generated using text-to-image models and filtered using an image-to-text model for quality control. This benchmark suite consists of 152,064 images of dimensions (3, 224, 224).

For the O2O task, the class (dog breed) and background combinations are sampled such that $\mu\%$ of the images per class contain a spurious background $b^{sp}$ and $(100 - \mu)\%$ contain a generic background $b^{ge}$. While the generic background is held constant for each class, each spurious background is observed in only one class ($p_{train}(b_i^{sp} \mid c_j) = 1$ if $i = j$ and 0 if $i \neq j$). Two separate training domains are defined by varying the value of $\mu$. These induced spurious correlations are reverted to yield a test domain with an unseen class-background pair for each class ($p_{test}(b_i \mid c_i) = 1$).

For the M2M task, disjoint class and background groups are constructed $\mathcal{B}_1, \mathcal{B}_2, \mathcal{C}_1, \mathcal{C}_2$, each with two elements. To introduce the training domains, class-background combinations $(c, b)$ are selected with $c \in \mathcal{C}_i$ and $b \in \mathcal{B}_i$. Each training domain consists of a single background per class such that $p_{train}^e(b_k \mid c_k) = e$, with domain index $e \in \{0, 1\}$, $b_k \in \mathcal{B}_i$, $c_k \in \mathcal{C}_i$. In contrast, the test domain is generated by selecting combinations from $c \in \mathcal{C}_i$ and $b \in \mathcal{B}_j$ with $i \neq j$ and sampling backgrounds such that $p_{test}(b_1 \mid c_k) = p_{test}(b_2 \mid c_k) = 0.5$ for $c_k \in \mathcal{C}_i$, $\{b_1, b_2\} = \mathcal{B}_j$.

The difficulty level (Easy, Medium, Hard) differs due to the splits in the available class-background combinations. These splits were empirically determined, and the full details of the final data combinations are found in Table 2 of Lynch et al. (2023).

**Discussion.** We observed that test environments 1 and 2 have a strong correlation between ID and OOD accuracy and have a slope of 1, making them misspecified for benchmarking domain generalization. Appropriately, Lynch et al. (2023) propose transferring to test environment '0' as the spurious correlation task. The correlation for test environment 0 is much weaker than the others, indicating that ID improvement does not as strongly imply OOD improvement. While there is still a positive linear correlation, the interpretation of these benchmarking results is informative because of the knowledge of the spurious correlation mechanism. Lynch et al. (2023) give examples of informative analysis of benchmarking results on this dataset. Notably, the O2O_easy setting has a weaker correlation by design, and the accuracy on the line strength increases. We see similar behavior for the M2M_ setting. However, this task is much harder than the O2O task, which is reflected in weaker accuracy on the line.

### C.3.1 SpawriousO2O Easy

Table 7: SpawriousO2O_easy ID vs. OOD properties.

| OOD | slope | intercept | Pearson R | p-value | standard error |
|---|---|---|---|---|---|
| Env 0 acc | 0.48 | -0.29 | 0.74 | 0.00 | 0.04 |
| Env 1 acc | 1.05 | -0.13 | 0.98 | 0.00 | 0.02 |
| Env 2 acc | 0.95 | -0.11 | 0.97 | 0.00 | 0.02 |

Table 8: SpawriousO2O_easy ID vs. OOD properties.

| OOD | ID | slope | intercept | Pearson R | p-value | standard error |
|---|---|---|---|---|---|---|
| Env 0 acc | Env 1 acc | 0.50 | -0.33 | 0.75 | 0.00 | 0.04 |
| Env 0 acc | Env 2 acc | 0.47 | -0.23 | 0.72 | 0.00 | 0.04 |
| Env 1 acc | Env 0 acc | 1.09 | -0.33 | 0.93 | 0.00 | 0.04 |
| Env 1 acc | Env 2 acc | 1.01 | 0.11 | 0.98 | 0.00 | 0.02 |
| Env 2 acc | Env 0 acc | 0.93 | -0.12 | 0.94 | 0.00 | 0.03 |
| Env 2 acc | Env 1 acc | 0.94 | -0.02 | 0.98 | 0.00 | 0.01 |

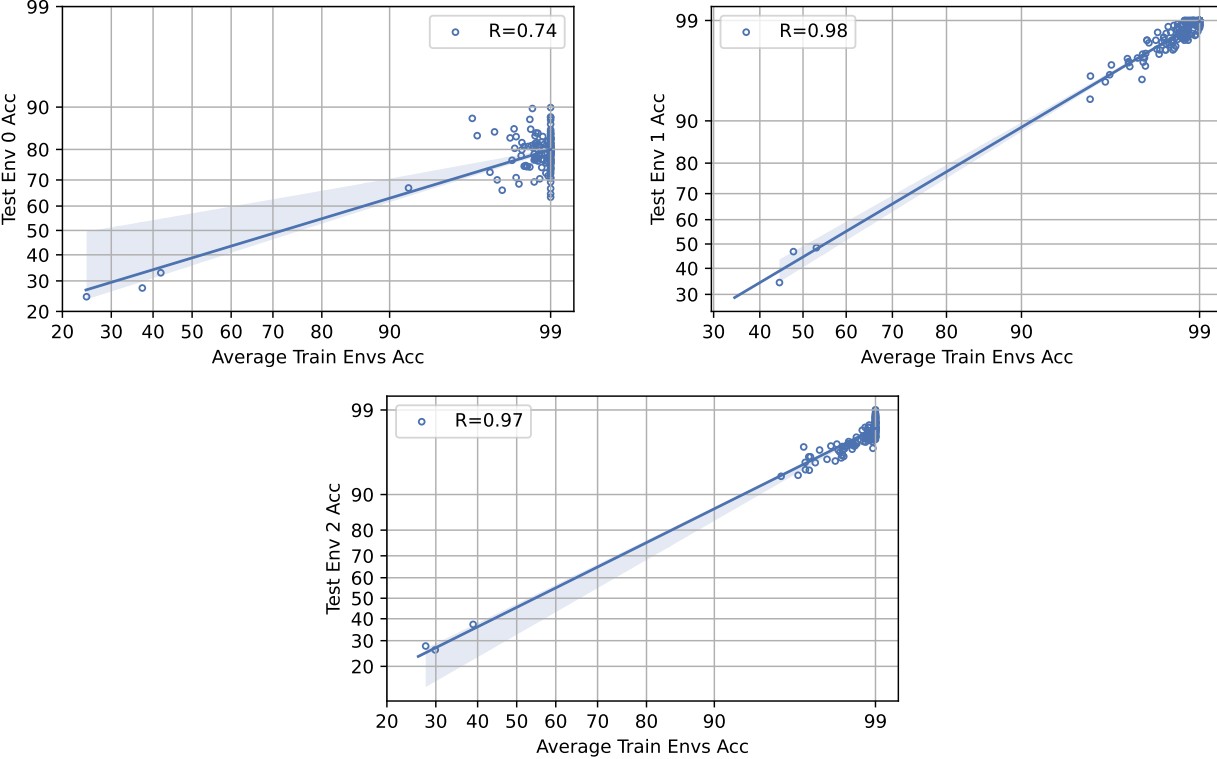

Figure 10: SpawriousO2O easy: Average train Env Accuracy vs. Test Env Accuracy.

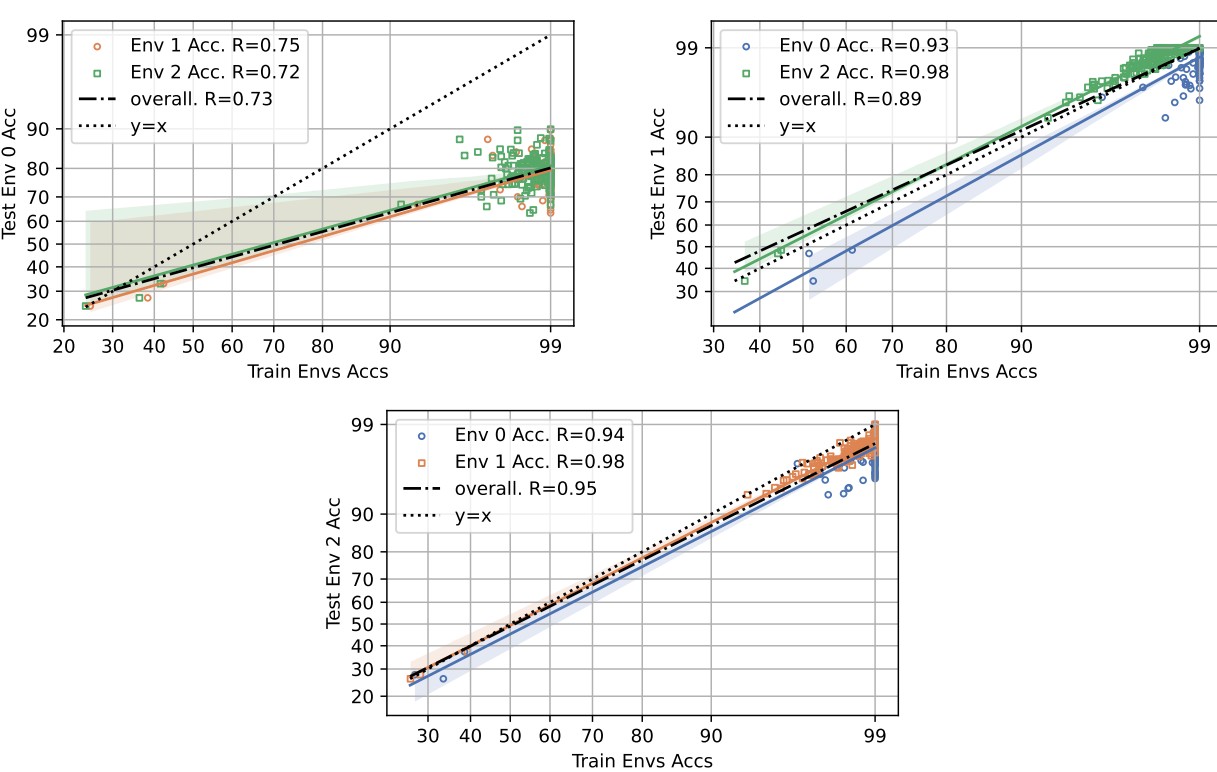

Figure 11: SpawriousO2O easy: Train Env Accuracy vs. Test Env Accuracy.

### C.3.2 SpawriousO2O Hard

Table 9: SpawriousO2O_hard ID vs. OOD properties.

| OOD | slope | intercept | Pearson R | p-value | standard error |
|---|---|---|---|---|---|
| Env 0 acc | 0.32 | -0.21 | 0.50 | 0.00 | 0.05 |
| Env 1 acc | 0.98 | 0.06 | 0.96 | 0.00 | 0.02 |
| Env 2 acc | 0.94 | -0.07 | 0.96 | 0.00 | 0.02 |

Table 10: SpawriousO2O_hard ID vs. OOD properties.

| OOD | ID | slope | intercept | Pearson R | p-value | standard error |
|---|---|---|---|---|---|---|
| Env 0 acc | Env 1 acc | 0.32 | -0.23 | 0.49 | 0.00 | 0.05 |
| Env 0 acc | Env 2 acc | 0.32 | -0.19 | 0.50 | 0.00 | 0.04 |
| Env 1 acc | Env 0 acc | 0.90 | 0.14 | 0.89 | 0.00 | 0.04 |
| Env 1 acc | Env 2 acc | 1.01 | 0.12 | 0.97 | 0.00 | 0.02 |
| Env 2 acc | Env 0 acc | 0.92 | -0.05 | 0.93 | 0.00 | 0.03 |
| Env 2 acc | Env 1 acc | 0.92 | 0.01 | 0.97 | 0.00 | 0.02 |

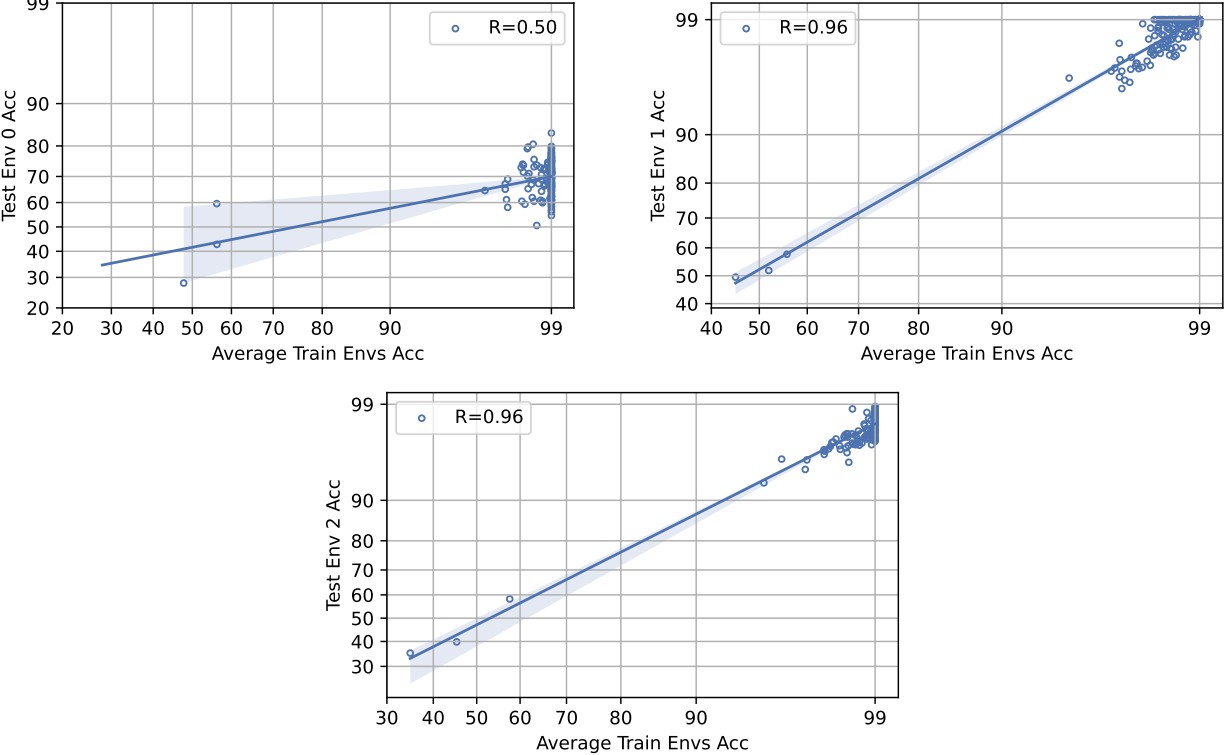

Figure 12: SpawriousO2O hard: Average train Env Accuracy vs. Test Env Accuracy.

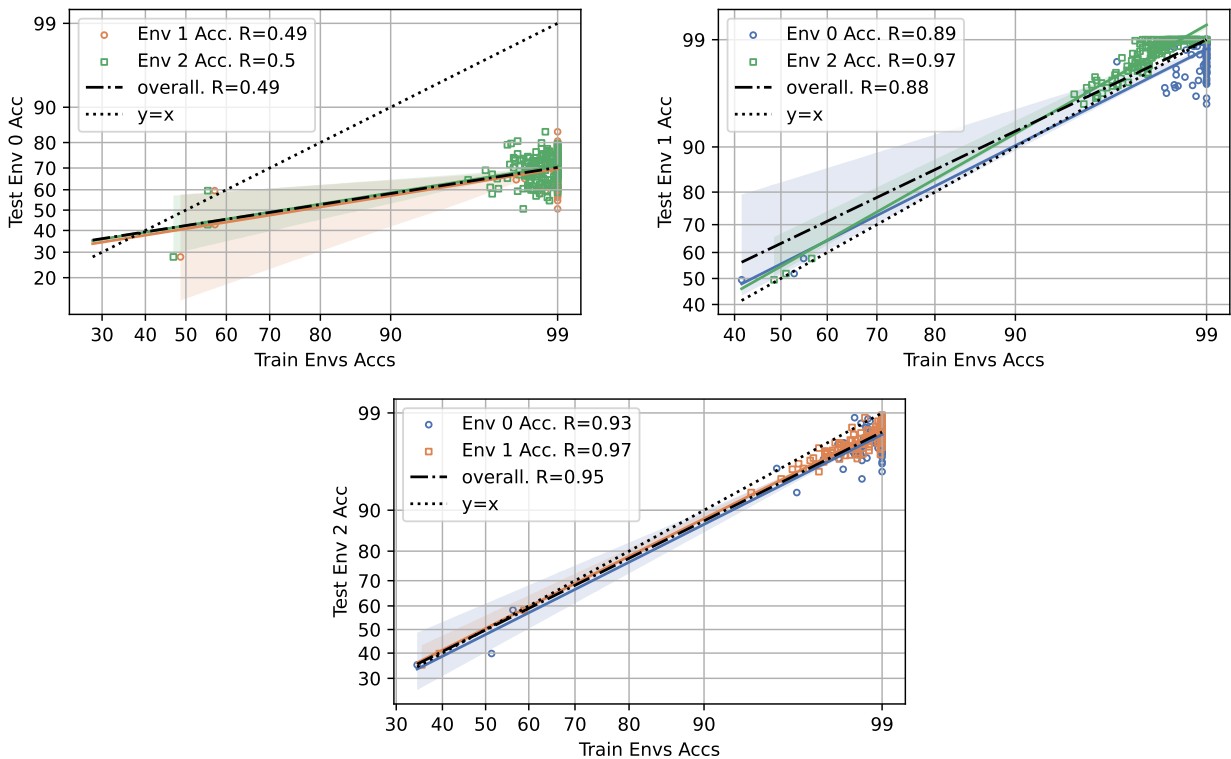

Figure 13: SpawriousO2O hard: Train Env Accuracy vs. Test Env Accuracy.

### C.3.3 SpawriousM2M Easy

Table 11: SpawriousM2M_easy ID vs. OOD properties.

| OOD | slope | intercept | Pearson R | p-value | standard error |
|---|---|---|---|---|---|
| Env 0 acc | 0.34 | 0.26 | 0.60 | 0.00 | 0.01 |
| Env 1 acc | 0.65 | -0.08 | 0.95 | 0.00 | 0.00 |
| Env 2 acc | 0.65 | 0.02 | 0.93 | 0.00 | 0.00 |

Table 12: SpawriousM2M_easy ID vs. OOD properties.

| OOD | ID | slope | intercept | Pearson R | p-value | standard error |
|---|---|---|---|---|---|---|
| Env 0 acc | Env 1 acc | 0.35 | 0.23 | 0.61 | 0.00 | 0.01 |
| Env 0 acc | Env 2 acc | 0.32 | 0.29 | 0.58 | 0.00 | 0.01 |
| Env 1 acc | Env 0 acc | 0.64 | -0.09 | 0.95 | 0.00 | 0.00 |
| Env 1 acc | Env 2 acc | 0.63 | -0.05 | 0.94 | 0.00 | 0.00 |
| Env 2 acc | Env 0 acc | 0.67 | -0.02 | 0.94 | 0.00 | 0.00 |
| Env 2 acc | Env 1 acc | 0.61 | 0.08 | 0.90 | 0.00 | 0.01 |

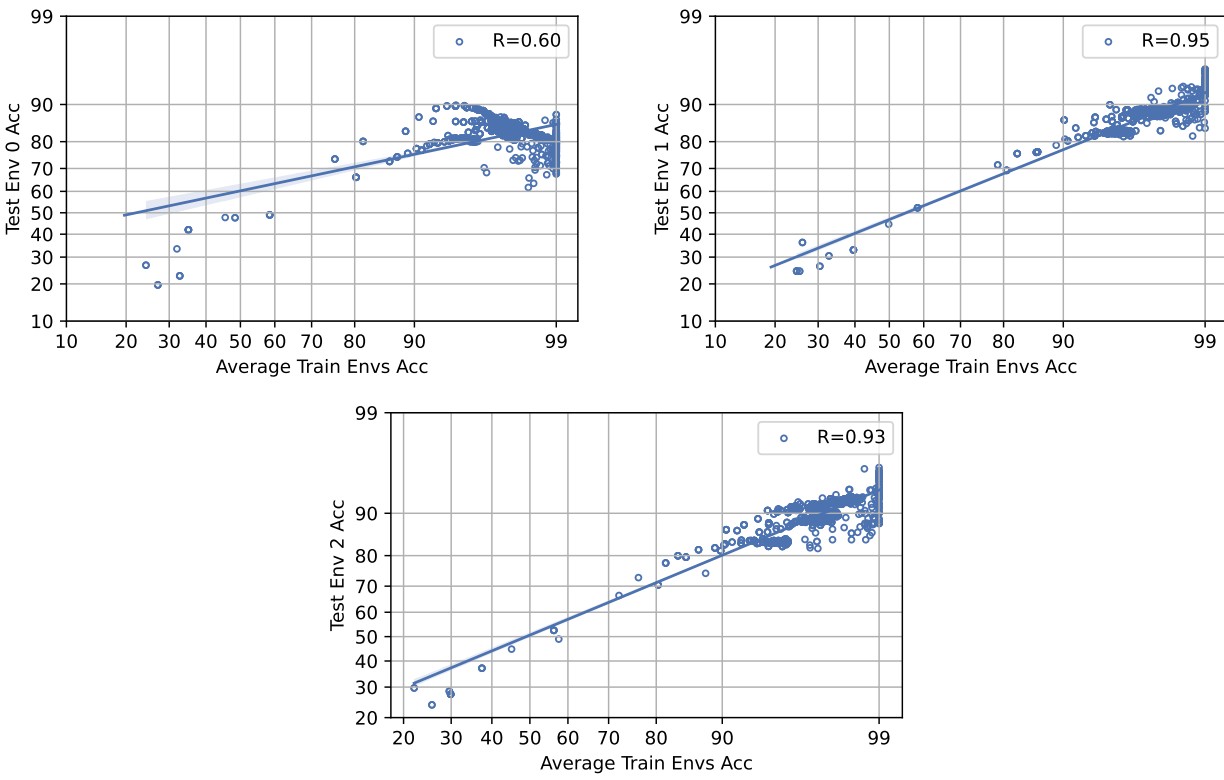

Figure 14: SpawriousO2O easy: Average train Env Accuracy vs. Test Env Accuracy.

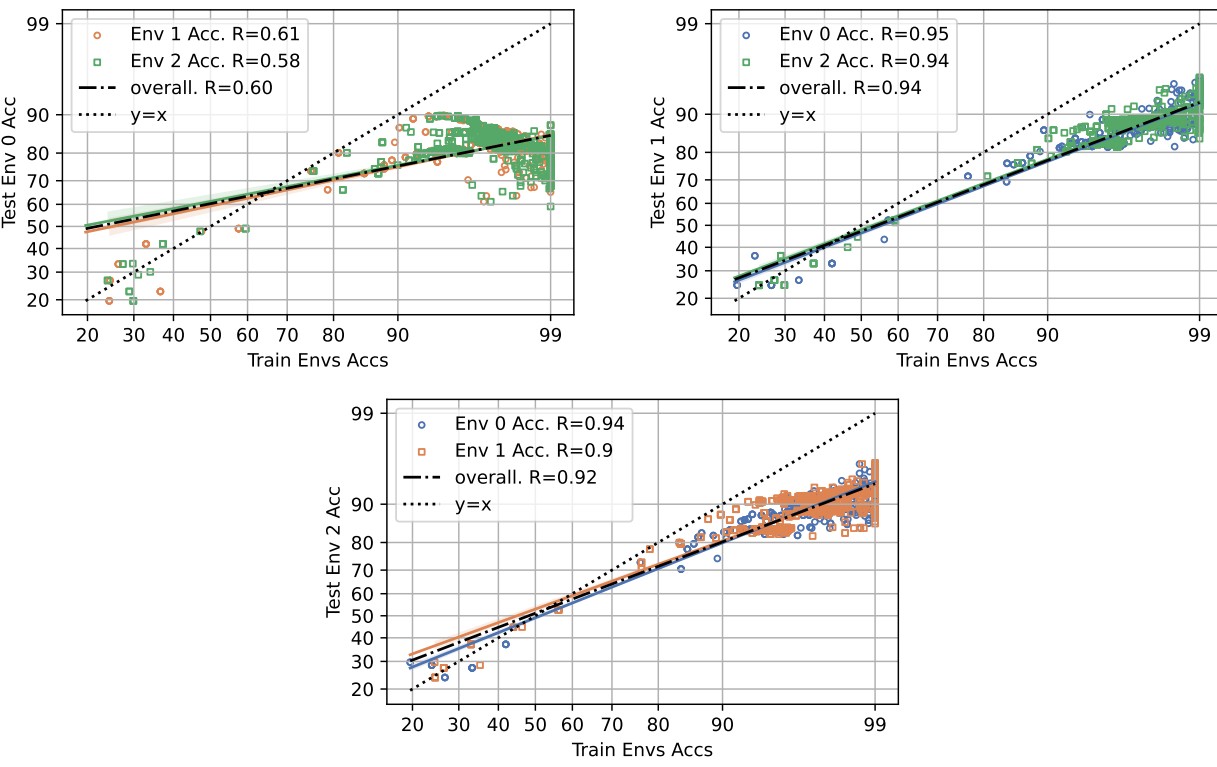

Figure 15: SpawriousO2O easy: Train Env Accuracy vs. Test Env Accuracy.

### C.3.4 SpawriousM2M Hard

Table 13: SpawriousM2M_hard ID vs. OOD properties.

| OOD | slope | intercept | Pearson R | p-value | standard error |
|---|---|---|---|---|---|
| Env 0 acc | 0.16 | -0.04 | 0.29 | 0.00 | 0.01 |
| Env 1 acc | 0.76 | -0.26 | 0.94 | 0.00 | 0.01 |
| Env 2 acc | 0.66 | -0.10 | 0.91 | 0.00 | 0.01 |

Table 14: SpawriousM2M_hard ID vs. OOD properties.

| OOD | ID | slope | intercept | Pearson R | p-value | standard error |
|---|---|---|---|---|---|---|
| Env 0 acc | Env 1 acc | 0.17 | -0.07 | 0.33 | 0.00 | 0.01 |
| Env 0 acc | Env 2 acc | 0.14 | -0.02 | 0.27 | 0.00 | 0.01 |
| Env 1 acc | Env 0 acc | 0.78 | -0.27 | 0.95 | 0.00 | 0.00 |
| Env 1 acc | Env 2 acc | 0.73 | -0.23 | 0.92 | 0.00 | 0.01 |
| Env 2 acc | Env 0 acc | 0.68 | -0.11 | 0.92 | 0.00 | 0.01 |
| Env 2 acc | Env 1 acc | 0.62 | -0.08 | 0.89 | 0.00 | 0.01 |

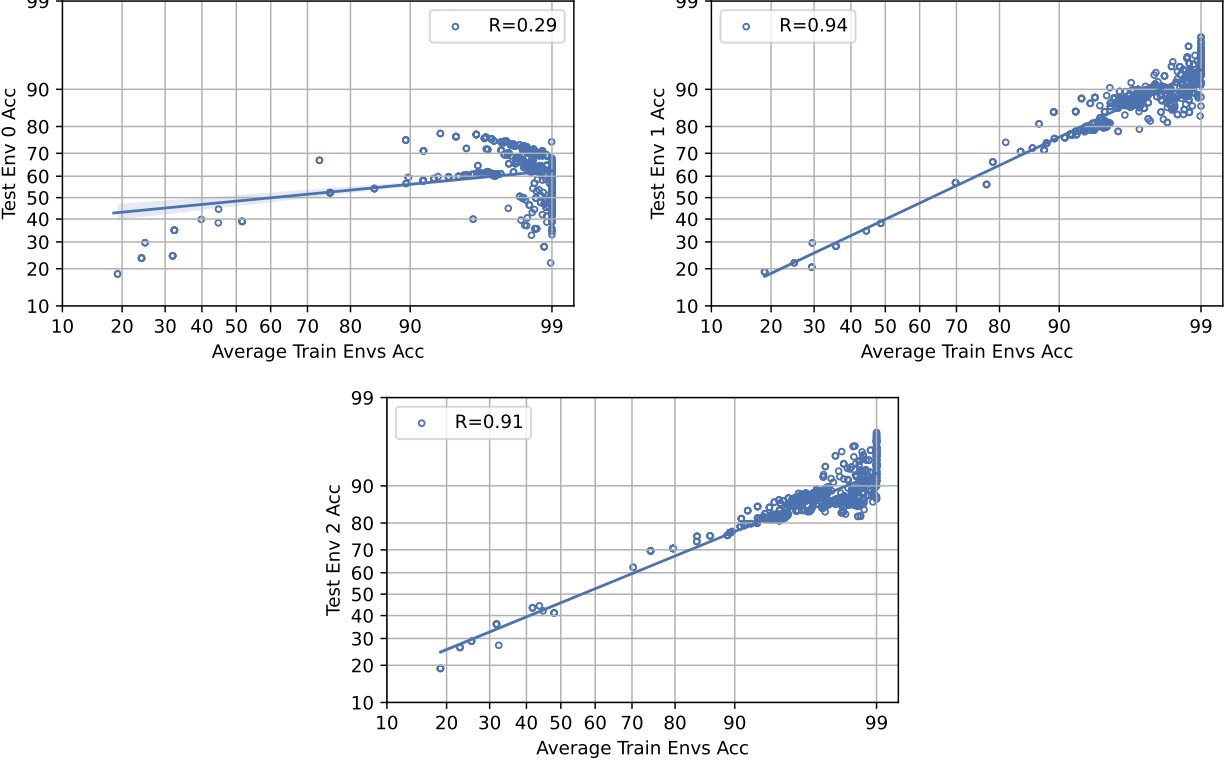

Figure 16: SpawriousM2M Hard: Average train Env Accuracy vs. Test Env Accuracy.

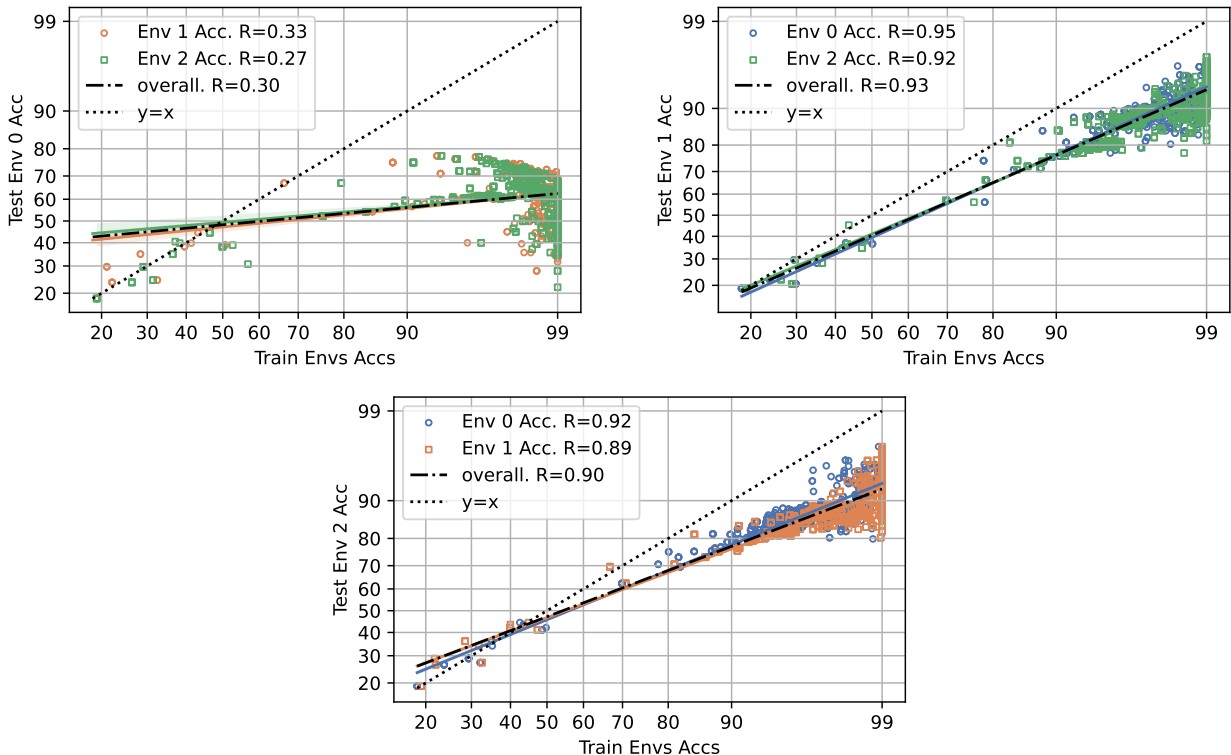

Figure 17: SpawriousM2M Hard: Train Env Accuracy vs. Test Env Accuracy.

## C.4 PACS

**PACS (Li et al., 2017).** A dataset comprised of four domains $d \in \{art, cartoons, photos, sketches\}$. This dataset contains 9,991 examples of dimension (3, 224, 224) and 7 classes.

**Discussion.** In general, we find that PACS does not strongly represent worst-case shifts for any split. Our results suggest that this benchmark may not accurately benchmark an algorithm's ability to give models free of spurious correlations.

Table 15: PACS ID vs. OOD properties.

| OOD | slope | intercept | Pearson R | p-value | standard error |
|---|---|---|---|---|---|
| Env 0 acc | 0.74 | -0.31 | 0.98 | 0.00 | 0.00 |
| Env 1 acc | 0.68 | -0.68 | 0.84 | 0.00 | 0.01 |
| Env 2 acc | 1.00 | 0.32 | 0.86 | 0.00 | 0.01 |
| Env 3 acc | 0.76 | -0.87 | 0.86 | 0.00 | 0.01 |

Table 16: PACS ID vs. OOD properties.

| OOD | ID | slope | intercept | Pearson R | p-value | standard error |
|-----|----|-------|-----------|-----------|---------|----------------|
| Env 0 acc | Env 1 acc | 0.71 | -0.10 | 0.96 | 0.00 | 0.01 |
| Env 0 acc | Env 2 acc | 0.64 | -0.47 | 0.91 | 0.00 | 0.01 |
| Env 0 acc | Env 3 acc | 0.64 | 0.14 | 0.90 | 0.00 | 0.01 |
| Env 1 acc | Env 0 acc | 0.75 | -0.59 | 0.89 | 0.00 | 0.01 |
| Env 1 acc | Env 2 acc | 0.51 | -0.67 | 0.71 | 0.00 | 0.01 |
| Env 1 acc | Env 3 acc | 0.71 | -0.29 | 0.98 | 0.00 | 0.00 |
| Env 2 acc | Env 0 acc | 1.04 | 0.23 | 0.87 | 0.00 | 0.01 |
| Env 2 acc | Env 1 acc | 0.90 | 0.45 | 0.82 | 0.00 | 0.02 |
| Env 2 acc | Env 3 acc | 0.78 | 0.76 | 0.75 | 0.00 | 0.02 |
| Env 3 acc | Env 0 acc | 0.76 | -0.79 | 0.83 | 0.00 | 0.01 |
| Env 3 acc | Env 1 acc | 0.80 | -0.75 | 0.92 | 0.00 | 0.01 |
| Env 3 acc | Env 2 acc | 0.50 | -0.83 | 0.64 | 0.00 | 0.02 |

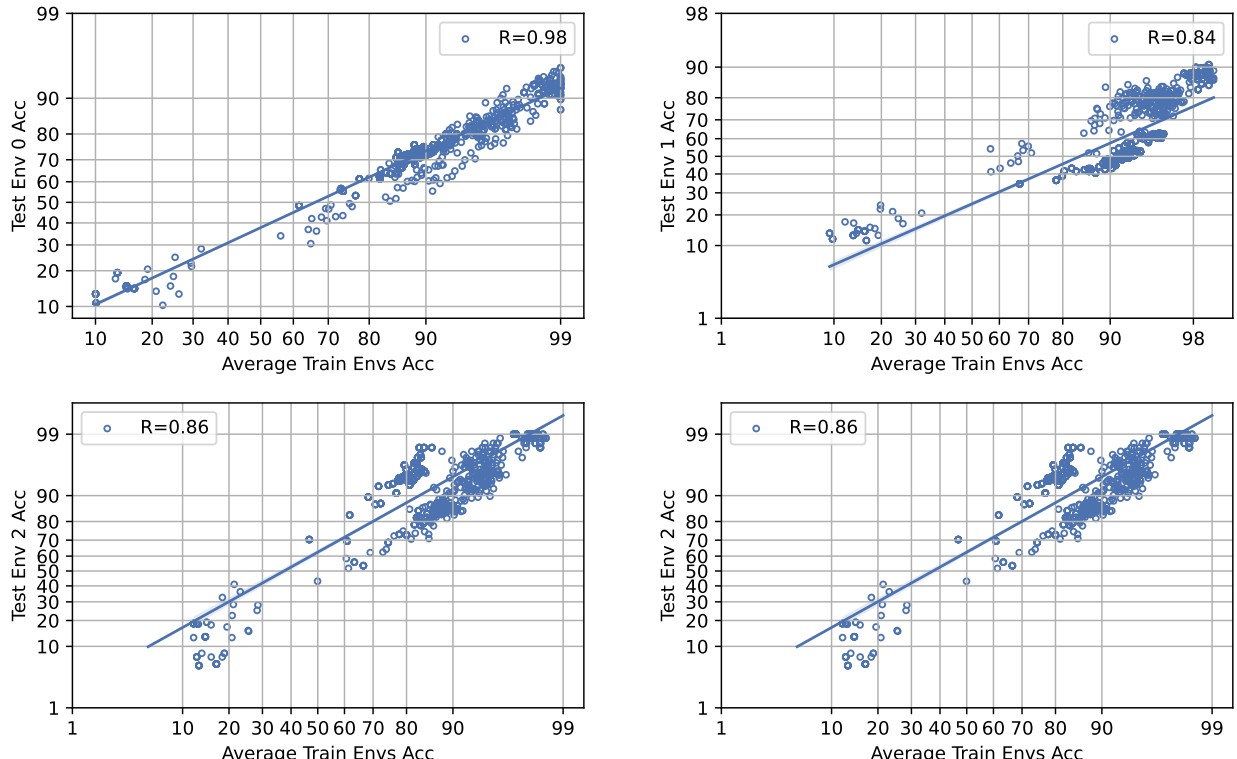

Figure 18: PACS: Average train Env Accuracy vs. Test Env Accuracy.

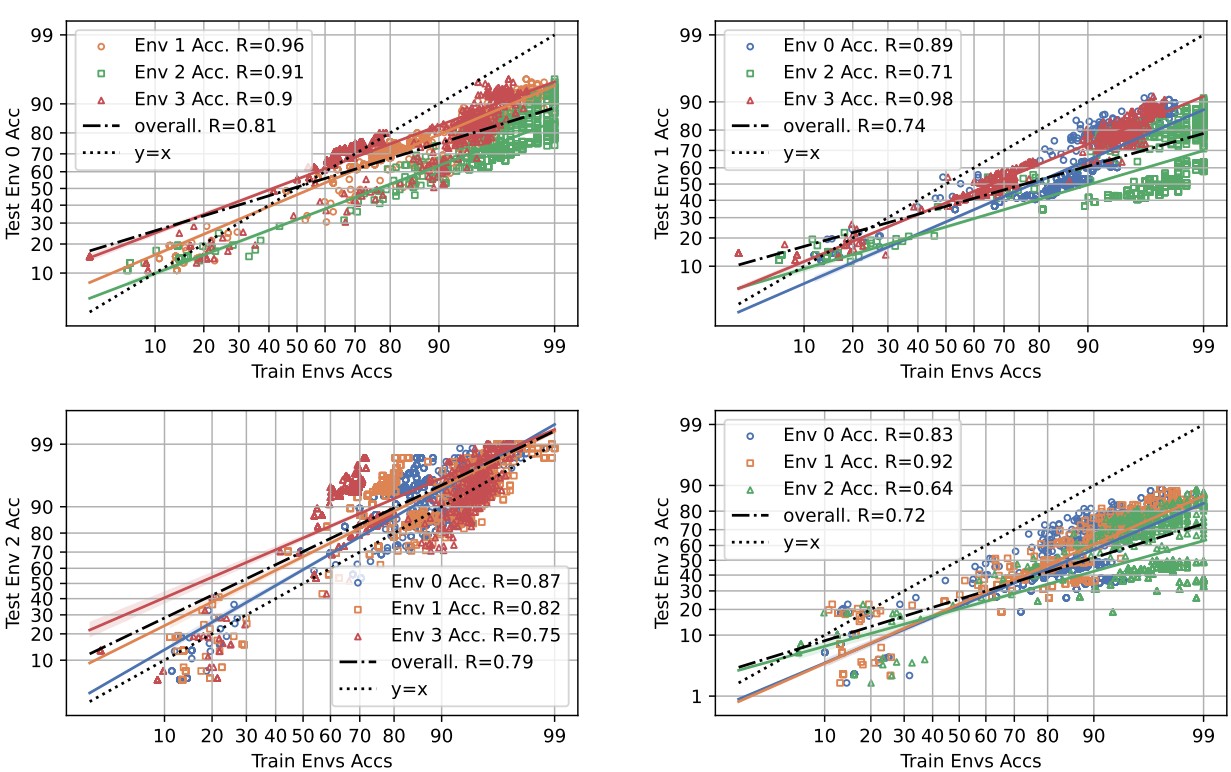

Figure 19: PACS: Train Env Accuracy vs. Test Env Accuracy.

### C.5 Terra Incognita

**Terra Incognita (Beery et al., 2018).** A dataset that contains photographs of wild animals taken by camera traps at locations $d \in \{L100, L38, L43, L46\}$. This dataset contains 24,788 examples of dimensions (3, 224, 224) and 10 classes: Bird, Bobcat, Cat, Coyote, Dog, Empty, Fox, Horse, Mouse, Opossum, Rabbit, Raccoon, Rat, Skunk, Squirrel, Weasel.

**Discussion.** In general, we find that Terra Incognita does not strongly represent worst-case shifts for any split. Ahuja et al. (2021) consider Terra Incognita domain-general features to be fully informative, i.e., labels did not need to rely on spurious features such as the background to generate labels. Our results suggest that this benchmark may not accurately benchmark an algorithm's ability to give models free of spurious correlations. For Env 1, we observe that the slope of the line varies quite a bit. Particularly, there is a near-zero slope for models greater than 80% accuracy.

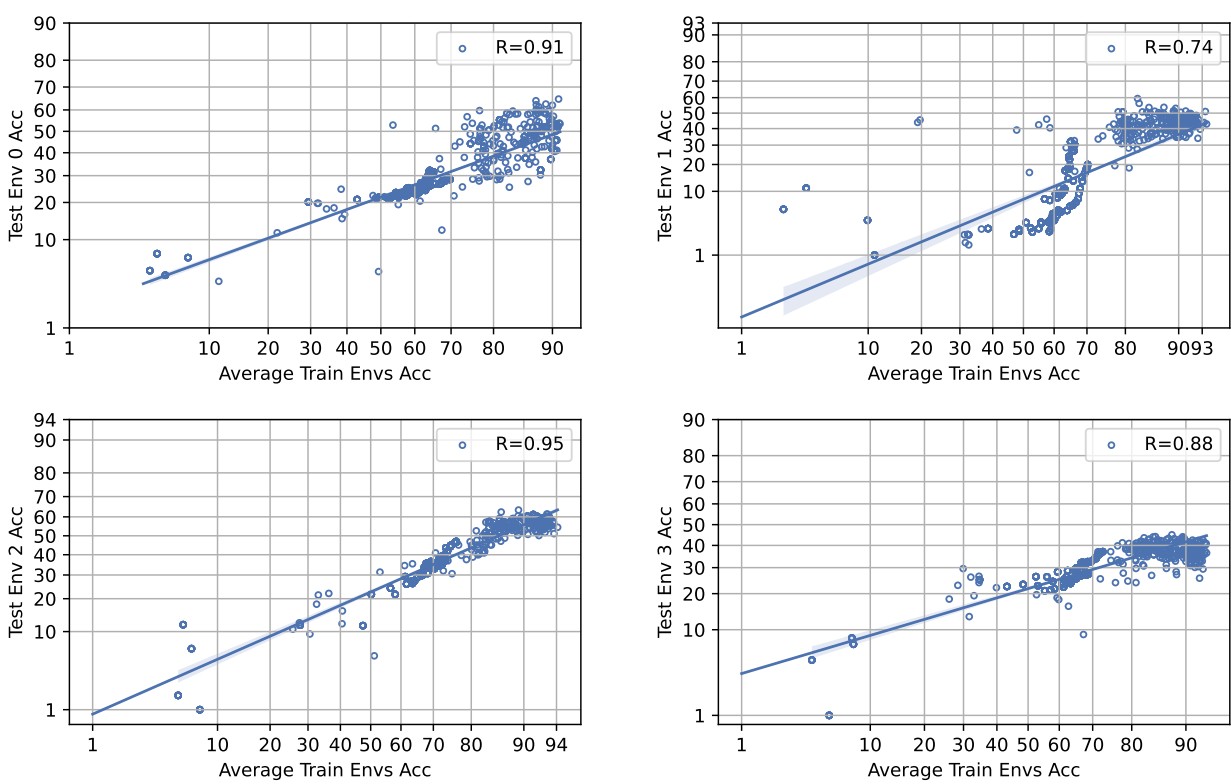

Figure 20: Terra Incognita: Average train Env Accuracy vs. Test Env Accuracy.

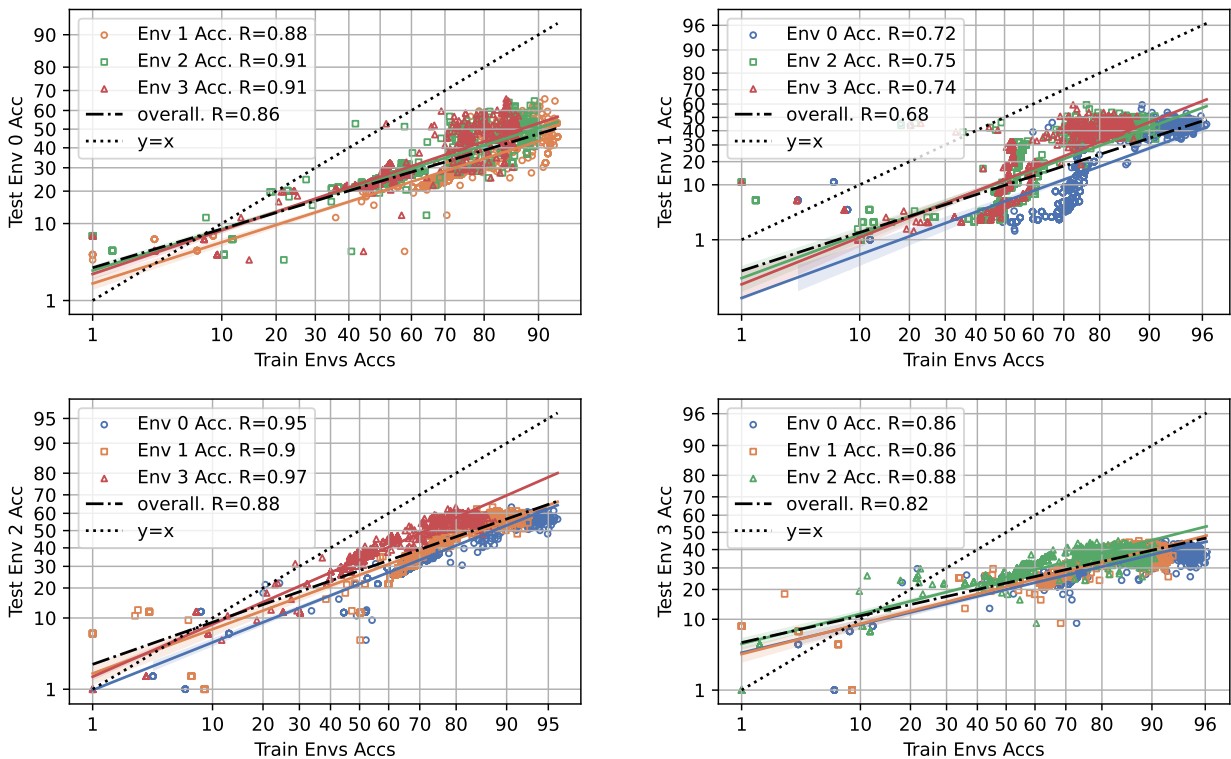

Figure 21: Terra Incognita: Train Env Accuracy vs. Test Env Accuracy.

## C.6 Camelyon

**Camelyon (Bandi et al., 2018; Koh et al., 2021).** A dataset that contains histopathological images of lymph node tissue, collected from two hospitals, denoted as Hospital A, Hospital B. This dataset contains 327,680 examples of dimension (3, 96, 96) and 2 classes (tumor, non-tumor).

**Discussion.** We find that overall, there is a strong correlation between ID and OOD accuracy. However, we observe that for some ID/OOD splits, a regime of training accuracy has a negative correlation (environments 0 and 2), suggesting that within a certain accuracy range, these splits may be well-specified for benchmarking spurious correlations for models in the regime with negative correlation. This highlights the importance of qualitative evaluation as opposed to quantitative evaluation.

Table 17: WILDSCamelyon ID vs. OOD properties.

| OOD | slope | intercept | Pearson R | p-value | standard error |
|---|---|---|---|---|---|
| Env 0 acc | 0.78 | 0.33 | 0.90 | 0.00 | 0.01 |
| Env 1 acc | 0.71 | -0.00 | 0.88 | 0.00 | 0.01 |
| Env 2 acc | 0.62 | 0.49 | 0.78 | 0.00 | 0.01 |
| Env 3 acc | 0.63 | 0.49 | 0.88 | 0.00 | 0.01 |
| Env 4 acc | 0.63 | 0.40 | 0.78 | 0.00 | 0.01 |

Table 18: WILDSCamelyon ID vs. OOD properties.

| OOD | ID | slope | intercept | Pearson R | p-value | standard error |
|---|---|---|---|---|---|---|
| Env 0 acc | Env 1 acc | 0.73 | 0.44 | 0.88 | 0.00 | 0.01 |
| Env 0 acc | Env 2 acc | 0.79 | 0.25 | 0.90 | 0.00 | 0.01 |
| Env 0 acc | Env 3 acc | 0.83 | 0.17 | 0.90 | 0.00 | 0.01 |
| Env 0 acc | Env 4 acc | 0.74 | 0.28 | 0.91 | 0.00 | 0.01 |
| Env 1 acc | Env 0 acc | 0.71 | -0.00 | 0.89 | 0.00 | 0.01 |
| Env 1 acc | Env 2 acc | 0.69 | 0.02 | 0.85 | 0.00 | 0.01 |
| Env 1 acc | Env 3 acc | 0.74 | -0.05 | 0.89 | 0.00 | 0.01 |
| Env 1 acc | Env 4 acc | 0.69 | -0.03 | 0.89 | 0.00 | 0.01 |
| Env 2 acc | Env 0 acc | 0.64 | 0.41 | 0.81 | 0.00 | 0.01 |
| Env 2 acc | Env 1 acc | 0.59 | 0.58 | 0.74 | 0.00 | 0.01 |
| Env 2 acc | Env 3 acc | 0.67 | 0.37 | 0.79 | 0.00 | 0.01 |
| Env 2 acc | Env 4 acc | 0.61 | 0.44 | 0.81 | 0.00 | 0.01 |
| Env 3 acc | Env 0 acc | 0.66 | 0.41 | 0.90 | 0.00 | 0.01 |
| Env 3 acc | Env 1 acc | 0.57 | 0.64 | 0.84 | 0.00 | 0.01 |
| Env 3 acc | Env 2 acc | 0.63 | 0.47 | 0.87 | 0.00 | 0.01 |
| Env 3 acc | Env 4 acc | 0.61 | 0.45 | 0.88 | 0.00 | 0.01 |
| Env 4 acc | Env 0 acc | 0.63 | 0.35 | 0.79 | 0.00 | 0.01 |
| Env 4 acc | Env 1 acc | 0.60 | 0.49 | 0.74 | 0.00 | 0.01 |
| Env 4 acc | Env 2 acc | 0.62 | 0.39 | 0.79 | 0.00 | 0.01 |
| Env 4 acc | Env 3 acc | 0.67 | 0.29 | 0.78 | 0.00 | 0.01 |

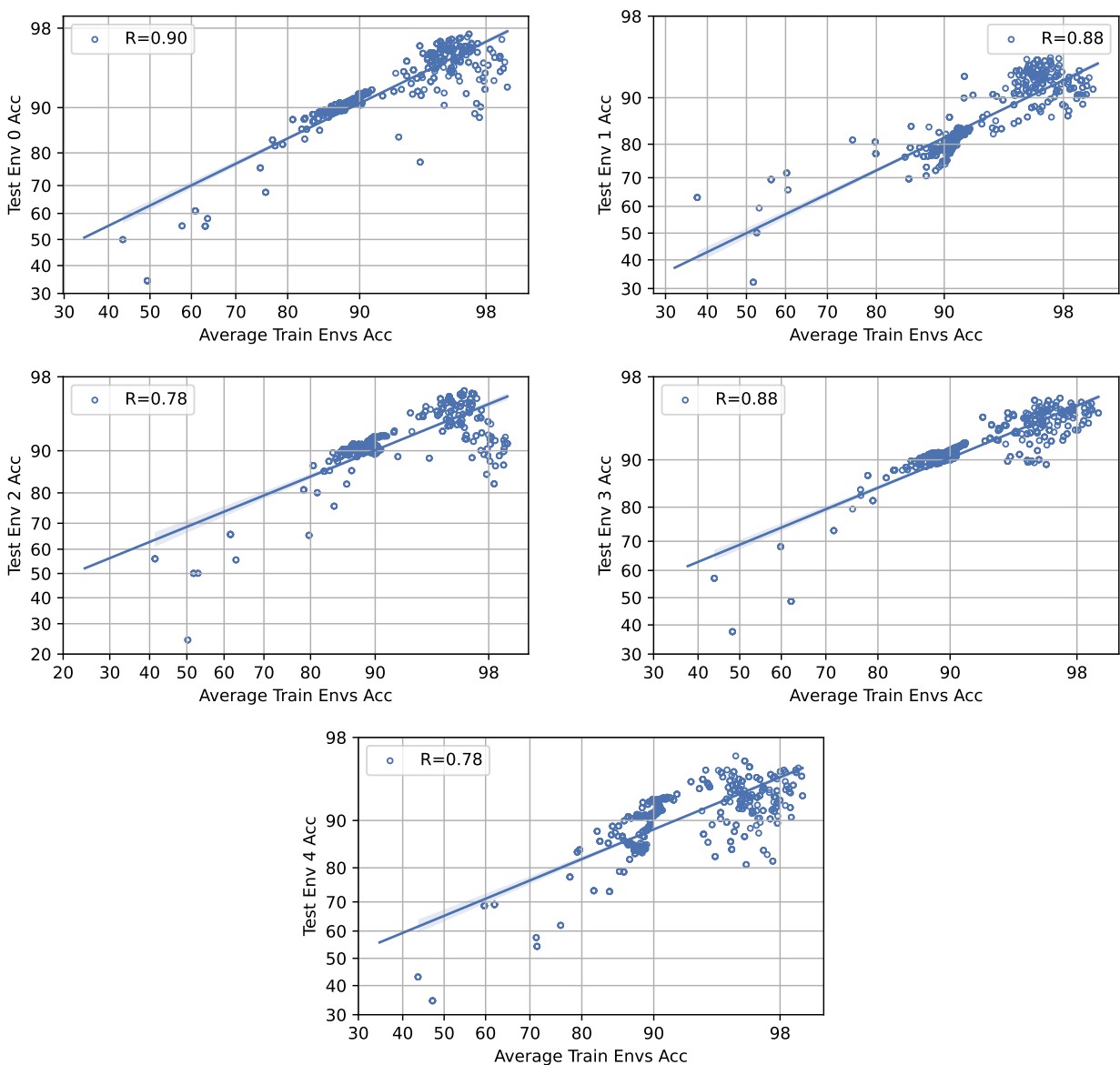

Figure 22: Camelyon: Average train Env Accuracy vs. Test Env Accuracy.

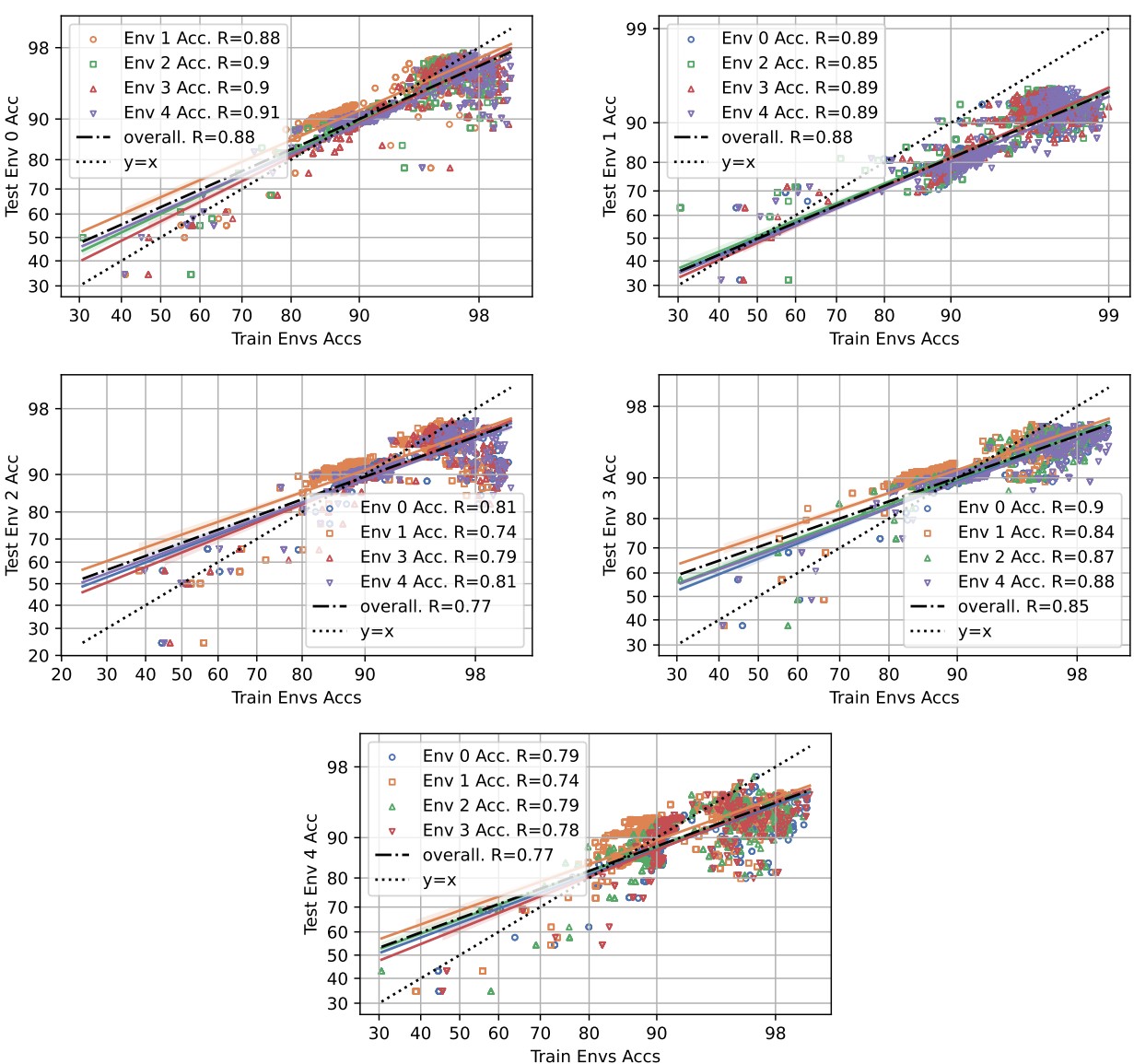

Figure 23: Camelyon: Train Env Accuracy vs. Test Env Accuracy.

## C.7 Covid-CXR

**Covid-CXR (Alzate-Grisales et al., 2022; Cohen et al., 2020b; Tabik et al., 2020; Tahir et al., 2021; Suwalska et al., 2023).** A dataset that aggregates five different benchmark Covid-19 datasets, represented as the domain $d \in \{$Cov-Caldas (Columbia), Covid-Chest-X-Ray (Global), Covid-GR (Spain), Covid-Qu-Ex (Global), and PolCovid (Poland)$\}$. These open-source Covid-19 datasets are collected from around the globe, accessible online. We are provided with *chest X-ray image (x)* and *binary diagnosis (y)*. The objective is to maintain consistent performance across different datasets $d$, which may incorporate data from singular or multiple sources.

| Dataset | # Train | # Test | % Pos Train | % Neg Train | % Pos Test | % Neg Test |
|---|---|---|---|---|---|---|
| Cov-Caldas | 3247 | 688 | 0.566 | 0.434 | 0.565 | 0.435 |
| Covid-Chest-X-Ray | 625 | 157 | 0.376 | 0.624 | 0.376 | 0.624 |
| Covid-GR | 681 | 171 | 0.501 | 0.499 | 0.497 | 0.503 |
| Covid-Qu-Ex | 21715 | 6788 | 0.353 | 0.647 | 0.353 | 0.647 |
| PolCovid | 4343 | 450 | 0.249 | 0.751 | 0.333 | 0.667 |

Table 19: Composition for Covid-19 Datasets.

- Cov-Caldas (Columbia, Alzate-Grisales et al. (2022)): This dataset was sourced from a single institution, S.E.S. Hospital Universitario de Caldas, located in the State of Caldas, Colombia. Labels were assigned based on positive results from conventional laboratory tests, such as PCR.

- Covid-Chest-X-Ray (Global, Cohen et al. (2020b)): This dataset, compiled from web sources, publications, and volunteer contributions, includes data on five types of pneumonia and Covid-19, along with metadata such as sex, age, and symptoms. The images are sourced from medical websites and are part of an open-source public project, where contributors can submit pull requests to add new images. The data is compiled from 138 unique locations.

- Covid-GR (Spain, Tabik et al. (2020)): In collaboration with four expert radiologists from Hospital Universitario Clínico San Cecilio in Granada, Spain, the authors developed a protocol for selecting and annotating chest X-ray (CXR) images for the dataset. A CXR image is labeled as Covid-19 positive if both the RT-PCR test and the radiologist's assessment confirm the diagnosis within 24 hours.

- Covid-Qu-Ex (Global, Tahir et al. (2021)): This dataset aggregates chest X-rays from six subdatasets, including the Covid-19 CXR dataset, RSNA CXR dataset (non-Covid infections and normal CXRs), Chest-Xray-Pneumonia dataset, PadChest dataset, Montgomery and Shenzhen CXR lung mask datasets, and QaTa-Cov19 CXR infection mask dataset. Designed to serve as a benchmark, it combines multiple publicly available datasets and repositories, which were originally dispersed and formatted differently. The authors performed quality control to ensure consistency. This dataset, specifically portions of their cited Covid-19 CXR dataset, overlaps with Covid-Chest-X-Ray (Env 1), thus deviating slightly from our standard experimental procedure of leave-one-domain-out.

- PolCovid (Poland, Suwalska et al. (2023)): Chest X-rays were collected from 15 Polish hospitals using a variety of devices and parameters due to differences in equipment between medical centers. The dataset includes patients with Covid-19, pneumonia, and healthy controls.

**Experimental Details.** Using the DomainBed suite by Gulrajani & Lopez-Paz (2020), we employ the ResNet-50 architecture (with and without AugMix data augmentation), along with ResNet-18, DenseNet-121, and ConvNeXt-Tiny, on the Covid-CXR dataset. Each of the five datasets is treated as a distinct domain, $d$. For each model, we apply two pretrained variants using ImageNet: (i) Fine-tuned and (ii) Transfer learning.

**Discussion.** Covid-CXR is a real-world medical dataset containing chest X-ray images of Covid-19 patients, where spurious correlations emerge naturally due to the complex and multifaceted nature of medical data. These correlations are not explicitly observed or intentionally introduced but arise from underlying factors such as imaging artifacts, patient demographics, or co-occurring medical conditions that are often unmeasured or unaccounted for in the dataset. As observed, there exist strong, weak, and inverse correlations between ID and OOD performance. While environments 0, 2, and 4 are primarily positively correlated in OOD performance, environments 1 and 3 demonstrate the inverse accuracy-on-the-line phenomenon. Additionally, several observations show weak correlations, with slopes close to zero. Existing literature suggests that a horizontal line (i.e. slope of 0) is indicative of a severe distribution shift, where it prevents any meaningful transfer learning between the training data and the OOD data Teney et al. (2024). This could be attributed to the significantly more severe distribution shifts present in this dataset. These shifts, as discussed by Cohen et al. (2020a), may arise due to errors in labeling, discrepancies between institutions and radiologists, biases in clinical practices, and interobserver variability.

Moreover, Covid-Qu-Ex (Env 3), the largest and most diverse dataset of those explored in this work, demonstrates low OOD transfer accuracy. For OOD performance for ID Env 3, we observe slopes closest to 0, indicating a weak or near-zero correlation. The high ID accuracy (up to $\approx 99\%$) suggests that the model may be learning misleading features in OOD domains. However, for environments evaluated on this dataset, a strongly negative relationship is present, suggesting that improved ID performance may be associated with reduced OOD accuracy. In this dataset, the authors scale by concatenating existing chest X-Ray datasets. But, as discussed in Cohen et al. (2020a), Shen et al. (2024), and Teney et al. (2024), simply increasing the amount of data may not address the core issue of distribution shift, as more data could exacerbate overfitting to domain-specific artifacts or noise, rather than improving generalization. Shen et al. (2024) coins this as the *Data Addition Dilemma*, where adding data can both improve and worsen performance.

Table 20: CXR ID vs. OOD properties.

| OOD | slope | intercept | Pearson R | p-value | standard error |
|---|---|---|---|---|---|
| Env 0 acc | 0.54 | -0.27 | 0.55 | 0.00 | 0.02 |
| Env 1 acc | -0.38 | 0.13 | -0.50 | 0.00 | 0.02 |
| Env 2 acc | 0.44 | 0.05 | 0.54 | 0.00 | 0.02 |
| Env 3 acc | -0.60 | 0.56 | -0.48 | 0.00 | 0.03 |
| Env 4 acc | 0.53 | -0.04 | 0.31 | 0.00 | 0.04 |

Table 21: CXR ID vs. OOD properties.

| OOD | ID | slope | intercept | Pearson R | p-value | standard error |
|---|---|---|---|---|---|---|
| Env 0 acc | Env 1 acc | 0.56 | -0.23 | 0.57 | 0.00 | 0.02 |
| Env 0 acc | Env 2 acc | 0.31 | -0.21 | 0.41 | 0.00 | 0.02 |
| Env 0 acc | Env 3 acc | 0.16 | -0.26 | 0.84 | 0.00 | 0.00 |
| Env 0 acc | Env 4 acc | 0.43 | -0.39 | 0.68 | 0.00 | 0.01 |
| Env 1 acc | Env 0 acc | -0.43 | 0.09 | -0.61 | 0.00 | 0.01 |
| Env 1 acc | Env 2 acc | -0.16 | 0.06 | -0.27 | 0.00 | 0.01 |
| Env 1 acc | Env 3 acc | -0.07 | 0.06 | -0.51 | 0.00 | 0.00 |
| Env 1 acc | Env 4 acc | -0.21 | 0.12 | -0.46 | 0.00 | 0.01 |
| Env 2 acc | Env 0 acc | 0.39 | 0.07 | 0.54 | 0.00 | 0.01 |
| Env 2 acc | Env 1 acc | 0.36 | 0.06 | 0.46 | 0.00 | 0.02 |
| Env 2 acc | Env 3 acc | 0.09 | 0.06 | 0.61 | 0.00 | 0.00 |
| Env 2 acc | Env 4 acc | 0.28 | -0.04 | 0.51 | 0.00 | 0.01 |
| Env 3 acc | Env 0 acc | -0.68 | 0.54 | -0.49 | 0.00 | 0.03 |
| Env 3 acc | Env 1 acc | -0.47 | 0.51 | -0.46 | 0.00 | 0.02 |
| Env 3 acc | Env 2 acc | -0.38 | 0.51 | -0.37 | 0.00 | 0.02 |
| Env 3 acc | Env 4 acc | -0.26 | 0.54 | -0.29 | 0.00 | 0.02 |
| Env 4 acc | Env 0 acc | -0.03 | 0.19 | -0.02 | 0.48 | 0.04 |
| Env 4 acc | Env 1 acc | 0.56 | -0.01 | 0.40 | 0.00 | 0.03 |
| Env 4 acc | Env 2 acc | 0.47 | -0.07 | 0.37 | 0.00 | 0.03 |
| Env 4 acc | Env 3 acc | 0.10 | 0.04 | 0.36 | 0.00 | 0.01 |

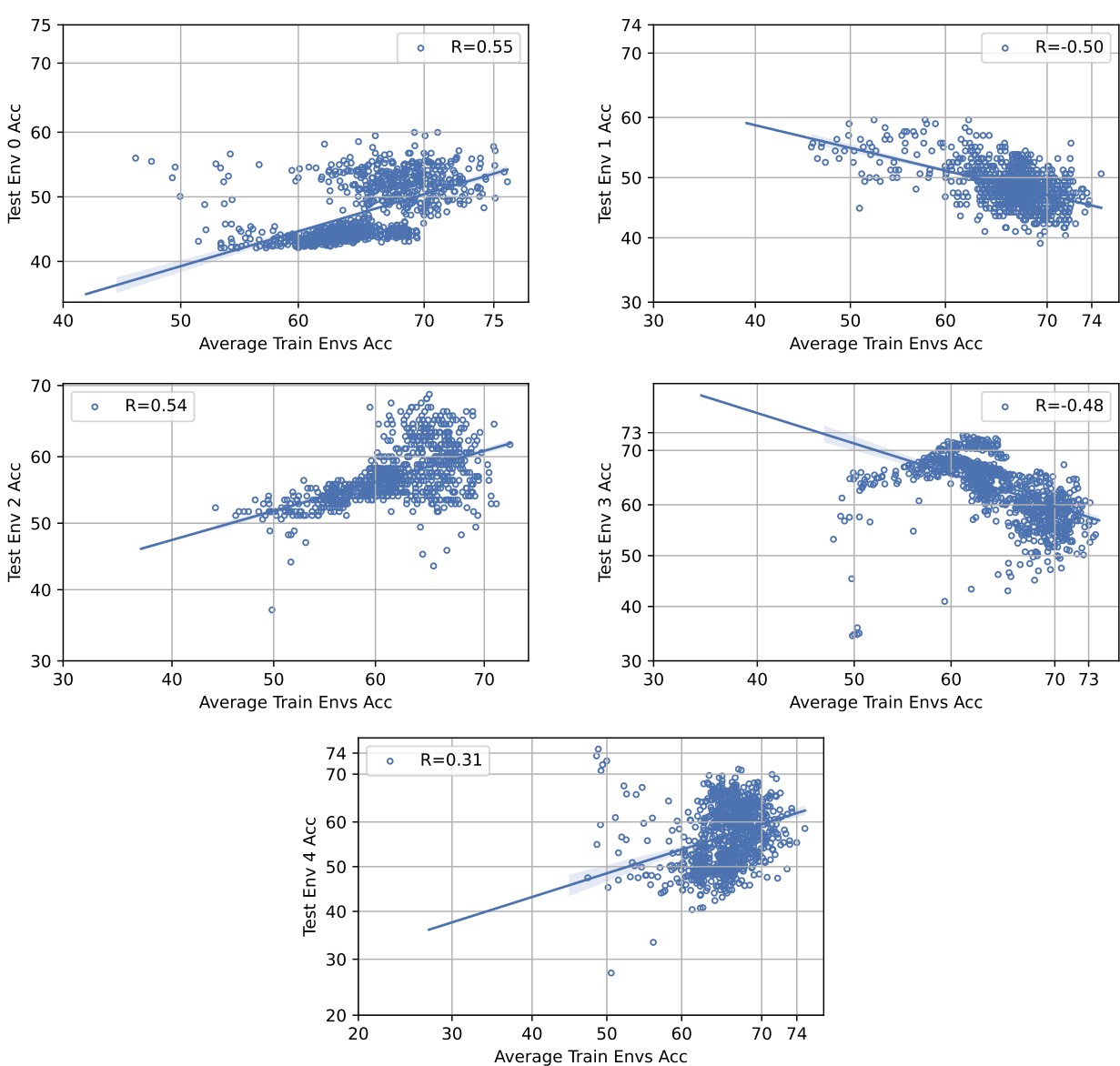

Figure 24: Covid-CXR: Average train Env Accuracy vs. Test Env Accuracy.

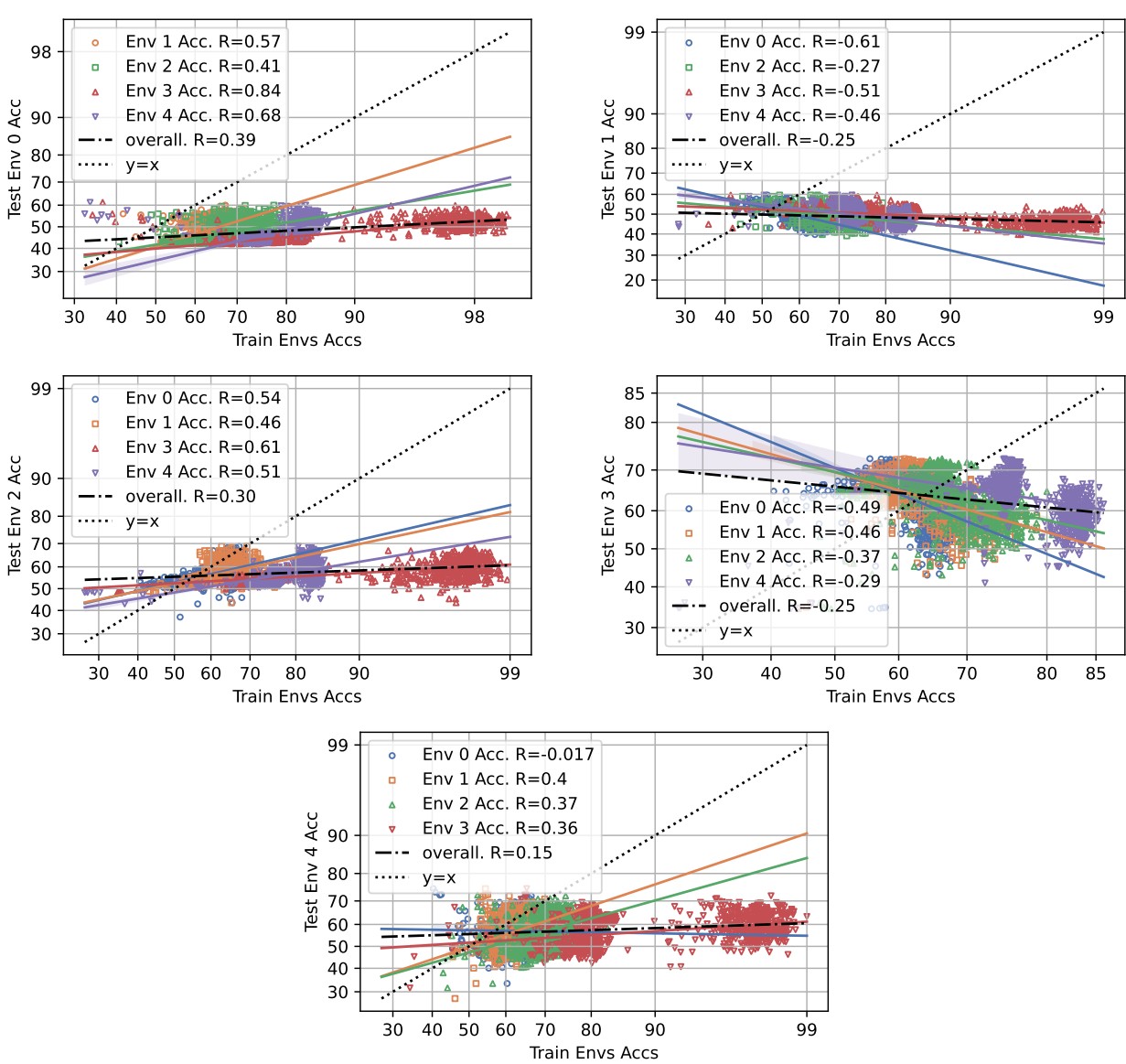

Figure 25: Covid-CXR: Train Env Accuracy vs. Test Env Accuracy.

## C.8 FMoW

**FMoW (Bandi et al., 2018; Koh et al., 2021).** A dataset consisting of 141,696 RGB satellite images from 2022 - 2017 (resized to 224 x 224 pixels), where the label is one of 62 building or land use categories. This dataset simultaneously considers a domain generalization task, where two domains are defined by the year of image acquisition $t \in \{before2016, after2016\}$, and a subpopulation shift task, where the domains are denoted by the geographic region of the image $r \in \{Africa, Americas, Oceana, Asia, Europe\}$.

**Discussion.** We find that WILDFMoW has accuracy on the line for all splits, suggesting that this benchmark may be misspecified for benchmarking spurious correlations.

Table 22: WILDSFMoW ID vs. OOD properties.

| OOD | slope | intercept | Pearson R | p-value | standard error |
|-----|-------|-----------|-----------|---------|----------------|
| Env 0 acc | 0.75 | -0.62 | 0.98 | 0.00 | 0.00 |
| Env 1 acc | 0.78 | -0.34 | 0.96 | 0.00 | 0.00 |
| Env 2 acc | 0.65 | -0.48 | 0.94 | 0.00 | 0.00 |
| Env 3 acc | 0.83 | -0.34 | 0.99 | 0.00 | 0.00 |
| Env 4 acc | 0.96 | -0.18 | 0.99 | 0.00 | 0.00 |
| Env 5 acc | 0.76 | -0.61 | 0.87 | 0.00 | 0.01 |

Table 23: WILDSFMoW ID vs. OOD properties.

| OOD | ID | slope | intercept | Pearson R | p-value | standard error |
|---|---|---|---|---|---|---|
| Env 0 acc | Env 1 acc | 0.78 | -0.52 | 0.98 | 0.00 | 0.00 |
| Env 0 acc | Env 2 acc | 0.69 | -0.72 | 0.97 | 0.00 | 0.00 |
| Env 0 acc | Env 3 acc | 0.71 | -0.52 | 0.97 | 0.00 | 0.00 |
| Env 0 acc | Env 4 acc | 0.48 | -0.89 | 0.96 | 0.00 | 0.00 |
| Env 0 acc | Env 5 acc | 0.43 | -1.11 | 0.94 | 0.00 | 0.00 |
| Env 1 acc | Env 0 acc | 0.75 | -0.26 | 0.97 | 0.00 | 0.00 |
| Env 1 acc | Env 2 acc | 0.78 | -0.43 | 0.96 | 0.00 | 0.00 |
| Env 1 acc | Env 3 acc | 0.80 | -0.20 | 0.98 | 0.00 | 0.00 |
| Env 1 acc | Env 4 acc | 0.53 | -0.62 | 0.95 | 0.00 | 0.00 |
| Env 1 acc | Env 5 acc | 0.49 | -0.88 | 0.94 | 0.00 | 0.00 |
| Env 2 acc | Env 0 acc | 0.58 | -0.50 | 0.94 | 0.00 | 0.00 |
| Env 2 acc | Env 1 acc | 0.70 | -0.47 | 0.93 | 0.00 | 0.00 |
| Env 2 acc | Env 3 acc | 0.64 | -0.49 | 0.92 | 0.00 | 0.00 |
| Env 2 acc | Env 4 acc | 0.43 | -0.80 | 0.91 | 0.00 | 0.00 |
| Env 2 acc | Env 5 acc | 0.39 | -1.01 | 0.92 | 0.00 | 0.00 |
| Env 3 acc | Env 0 acc | 0.74 | -0.30 | 0.98 | 0.00 | 0.00 |
| Env 3 acc | Env 1 acc | 0.91 | -0.25 | 0.99 | 0.00 | 0.00 |
| Env 3 acc | Env 2 acc | 0.79 | -0.49 | 0.97 | 0.00 | 0.00 |
| Env 3 acc | Env 4 acc | 0.53 | -0.67 | 0.96 | 0.00 | 0.00 |
| Env 3 acc | Env 5 acc | 0.49 | -0.96 | 0.92 | 0.00 | 0.00 |
| Env 4 acc | Env 0 acc | 0.89 | -0.13 | 0.98 | 0.00 | 0.00 |
| Env 4 acc | Env 1 acc | 1.02 | -0.08 | 0.99 | 0.00 | 0.00 |
| Env 4 acc | Env 2 acc | 0.92 | -0.33 | 0.98 | 0.00 | 0.00 |
| Env 4 acc | Env 3 acc | 0.95 | -0.08 | 0.99 | 0.00 | 0.00 |
| Env 4 acc | Env 5 acc | 0.54 | -0.87 | 0.90 | 0.00 | 0.00 |
| Env 5 acc | Env 0 acc | 0.74 | -0.56 | 0.89 | 0.00 | 0.01 |
| Env 5 acc | Env 1 acc | 0.82 | -0.54 | 0.85 | 0.00 | 0.01 |
| Env 5 acc | Env 2 acc | 0.70 | -0.74 | 0.84 | 0.00 | 0.01 |
| Env 5 acc | Env 3 acc | 0.74 | -0.55 | 0.85 | 0.00 | 0.01 |
| Env 5 acc | Env 4 acc | 0.52 | -0.90 | 0.86 | 0.00 | 0.00 |

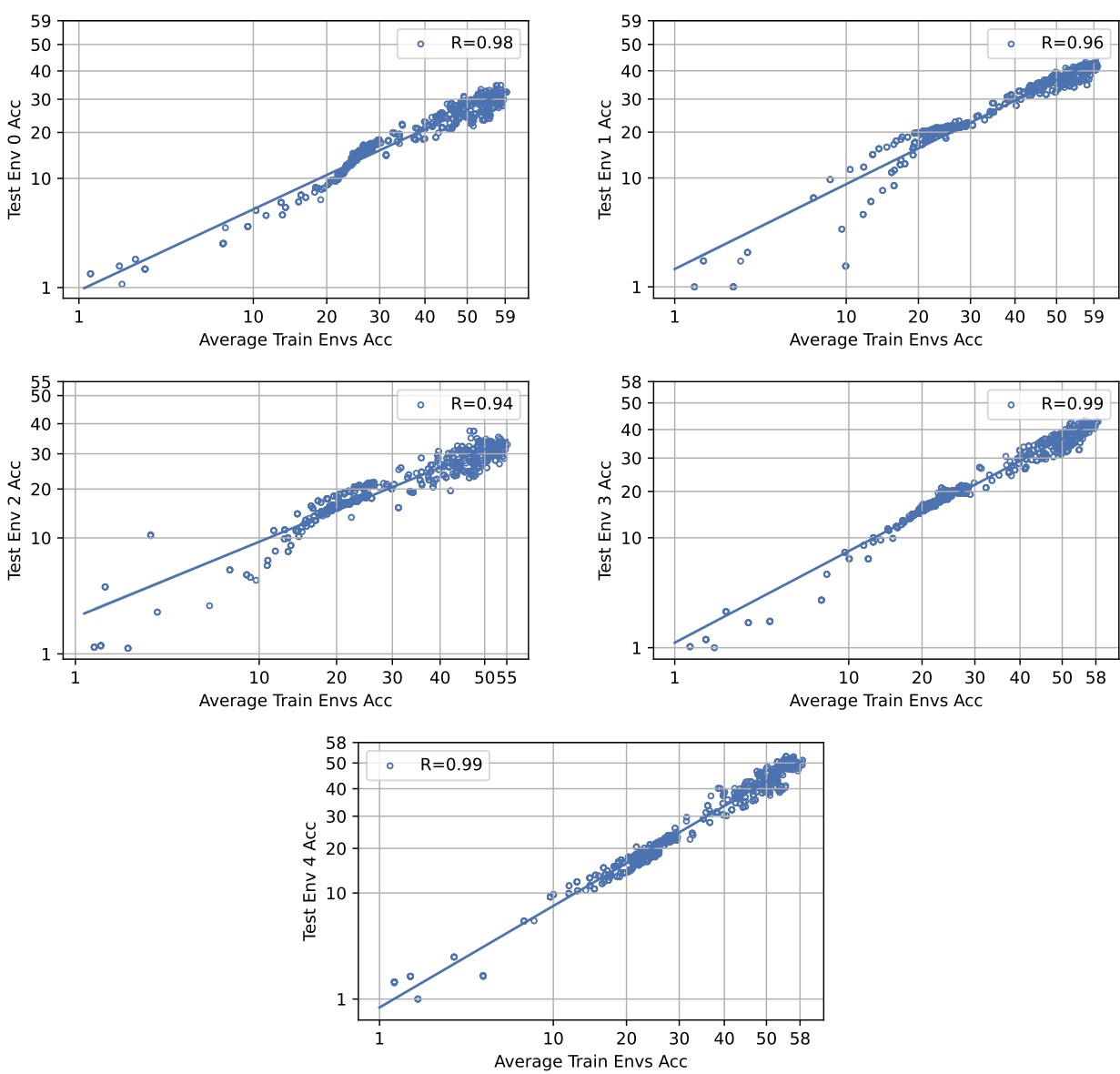

Figure 26: FMoW: Average train Env Accuracy vs. Test Env Accuracy.

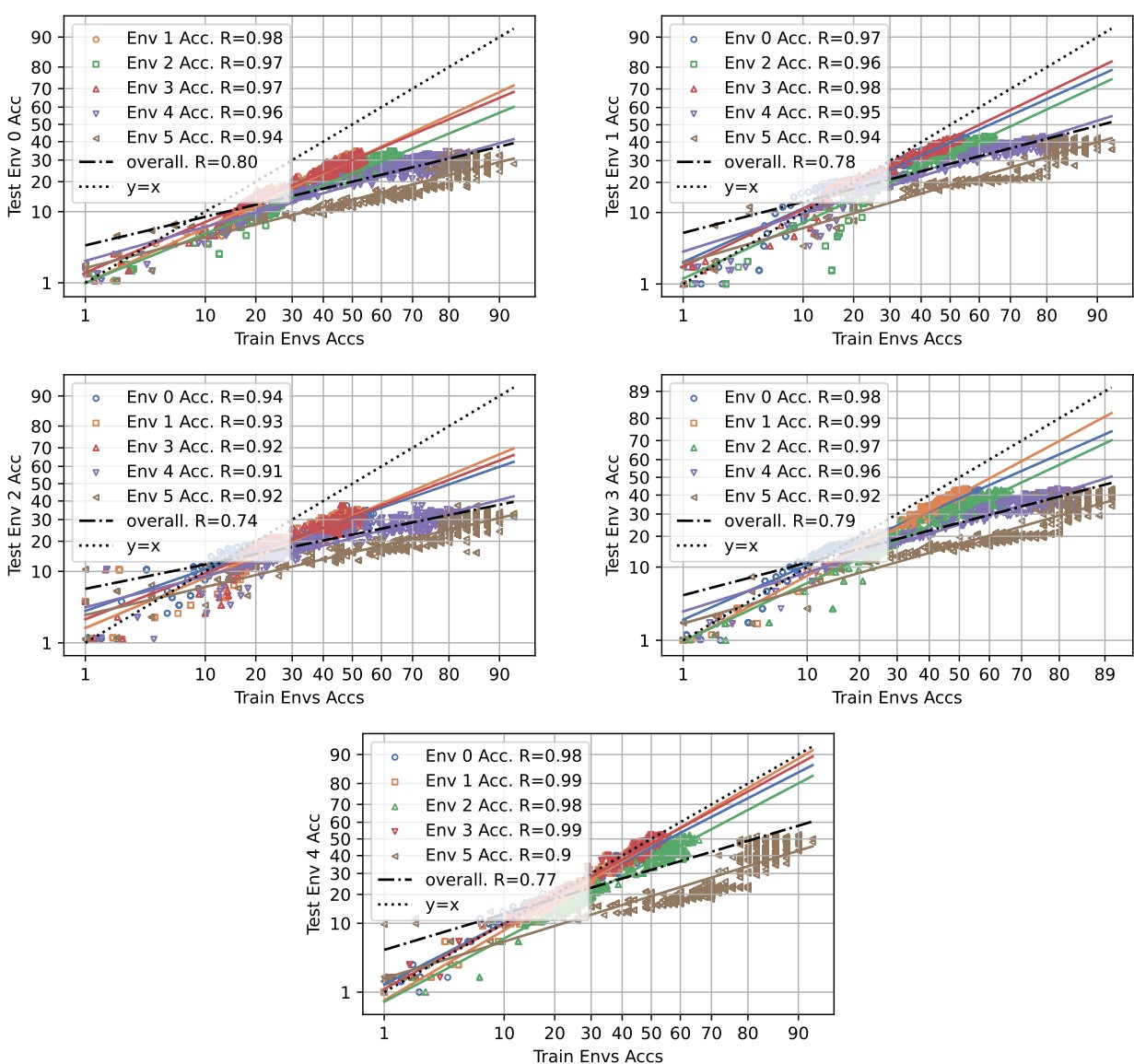

Figure 27: FMoW: Train Env Accuracy vs. Test Env Accuracy.

## C.9 Waterbirds

**Waterbirds (Sagawa et al., 2019; Koh et al., 2021).** The Waterbirds dataset is a modification of the CUB dataset (Welinder et al., 2010) constructed to induce a subpopulation shift in the association between bird type $b \in \{waterbird, landbird\}$ in the foreground and the image background $c \in \{water, land\}$. Waterbird labels are assigned to seabirds and waterfowl; all other bird types are labeled as landbirds. Image backgrounds are obtained from the Places dataset (Zhou et al., 2016) and subset to include water backgrounds (categories: ocean or natural lake) and land backgrounds (categories: bamboo forest or broadleaf forest). The training set consists of 95% of all waterbirds with a water background and the remaining 5% with a land background. Similarly, 95% of all landbirds are displayed on a land background with the remaining 5% on a water background. The validation and test sets include an equal distribution of waterbirds and landbirds on each background. This dataset consists of 11,788 examples of size (3 x 224 x 224).

**Experimental Details.** Environment 0 consists of the full training split from the Waterbirds dataset and Environment 1 is a concatenation of the validation and test splits from the Waterbirds dataset (Sagawa et al., 2019). In addition to plotting the average ID vs. OOD accuracies for each test environment, we also include average ID vs. group-specific accuracies and pairwise combinations of group-specific ID vs. group-specific OOD accuracy.

**Discussion.** In our evaluation of average ID and OOD performance, we find that the Waterbirds dataset does not strongly represent worst-case shifts. However, plotting group-specific OOD accuracies, we find no linear correlation for Environment 1 group ($y = 0, a = 1$), representing a degradation in worst-group accuracy (WGA) with improvements to Environment 0 ID average accuracy. We examine the pairwise group-specific accuracies for each environment and generally find that OOD group accuracy positively correlates with ID accuracy for groups sharing the same label. Since Waterbirds groups are defined by label (bird), attribute (background) pairs, this result is consistent with the studies that suggest worst-class accuracy (WCA) is a good proxy for WGA when group membership is unknown (Yang et al., 2023). Similarly, OOD group accuracy negatively correlates with ID accuracy for groups of the opposite label, demonstrating the trade-off between majority and minority group/class performance under subpopulation shifts.

Table 24: WILDSWaterbirds average ID vs. OOD properties.

| ID | OOD | slope | intercept | Pearson R | p-value | standard error |
|---|---|---|---|---|---|---|
| Env 0 acc | Env 1 acc | 0.83 | 0.35 | 0.92 | 0.00 | 0.01 |
| Env 1 acc | Env 0 acc | 0.47 | 0.16 | 0.69 | 0.00 | 0.02 |

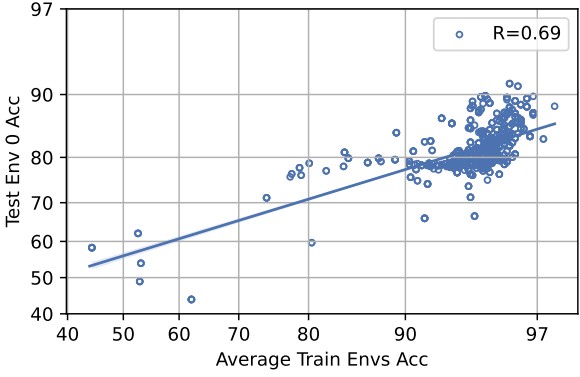 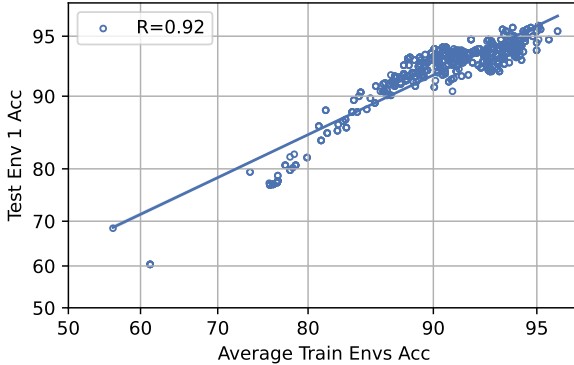

Figure 28: Waterbirds average ID vs. average OOD accuracies

Table 25: Waterbirds average ID vs. group-specific OOD properties.

| ID | OOD | slope | intercept | Pearson R | p-value | standard error |
|---|---|---|---|---|---|---|
| Env 0 avg acc | Env 1 y=0,a=0 acc | -0.13 | 1.58 | -0.13 | 0.00 | 0.03 |
| Env 0 avg acc | Env 1 y=0,a=1 acc | 0.00 | 1.32 | 0.00 | 0.95 | 0.03 |
| Env 0 avg acc | Env 1 y=1,a=0 acc | 0.22 | 1.17 | 0.84 | 0.00 | 0.00 |
| Env 0 avg acc | Env 1 y=1,a=1 acc | 0.32 | 1.13 | 0.81 | 0.00 | 0.01 |
| Env 1 avg acc | Env 0 y=0,a=0 acc | 0.70 | -0.02 | 0.65 | 0.00 | 0.03 |
| Env 1 avg acc | Env 0 y=0,a=1 acc | -0.07 | 1.61 | -0.11 | 0.00 | 0.02 |
| Env 1 avg acc | Env 0 y=1,a=0 acc | 0.27 | 1.63 | 0.56 | 0.00 | 0.01 |
| Env 1 avg acc | Env 0 y=1,a=1 acc | 0.30 | 1.21 | 0.60 | 0.00 | 0.01 |

Table 26: Waterbirds pairwise group-specific ID vs. OOD properties.

| ID | OOD | slope | intercept | Pearson R | p-value | standard error |
|---|---|---|---|---|---|---|
| Env 0 y=0,a=0 acc | Env 1 y=0,a=0 acc | 0.76 | 0.58 | 0.86 | 0.00 | 0.01 |
| Env 0 y=0,a=1 acc | Env 1 y=0,a=0 acc | 0.72 | 0.73 | 0.73 | 0.00 | 0.02 |
| Env 0 y=1,a=0 acc | Env 1 y=0,a=0 acc | -0.15 | 2.19 | -0.43 | 0.00 | 0.01 |
| Env 0 y=1,a=1 acc | Env 1 y=0,a=0 acc | -0.13 | 2.17 | -0.36 | 0.00 | 0.01 |
| Env 0 y=0,a=0 acc | Env 1 y=0,a=1 acc | 0.45 | 1.05 | 0.42 | 0.00 | 0.03 |
| Env 0 y=0,a=1 acc | Env 1 y=0,a=1 acc | 0.65 | 0.72 | 0.54 | 0.00 | 0.03 |
| Env 0 y=1,a=0 acc | Env 1 y=0,a=1 acc | -0.03 | 1.96 | -0.06 | 0.08 | 0.01 |
| Env 0 y=1,a=1 acc | Env 1 y=0,a=1 acc | -0.05 | 1.99 | -0.10 | 0.00 | 0.01 |
| Env 0 y=0,a=0 acc | Env 1 y=1,a=0 acc | -1.16 | 3.05 | -0.33 | 0.00 | 0.10 |
| Env 0 y=0,a=1 acc | Env 1 y=1,a=0 acc | -0.96 | 2.58 | -0.25 | 0.00 | 0.12 |
| Env 0 y=1,a=0 acc | Env 1 y=1,a=0 acc | 1.24 | 0.13 | 0.93 | 0.00 | 0.02 |
| Env 0 y=1,a=1 acc | Env 1 y=1,a=0 acc | 1.34 | 0.12 | 0.92 | 0.00 | 0.02 |
| Env 0 y=0,a=0 acc | Env 1 y=1,a=1 acc | -0.70 | 2.02 | -0.32 | 0.00 | 0.07 |
| Env 0 y=0,a=1 acc | Env 1 y=1,a=1 acc | -0.87 | 2.24 | -0.34 | 0.00 | 0.08 |
| Env 0 y=1,a=0 acc | Env 1 y=1,a=1 acc | 0.84 | 0.19 | 0.95 | 0.00 | 0.01 |
| Env 0 y=1,a=1 acc | Env 1 y=1,a=1 acc | 0.92 | 0.20 | 0.97 | 0.00 | 0.01 |
| Env 1 y=0,a=0 acc | Env 0 y=0,a=0 acc | 0.77 | 0.55 | 0.96 | 0.00 | 0.01 |
| Env 1 y=0,a=1 acc | Env 0 y=0,a=0 acc | 0.16 | 2.18 | 0.35 | 0.00 | 0.01 |
| Env 1 y=1,a=0 acc | Env 0 y=0,a=0 acc | -0.07 | 2.23 | -0.18 | 0.00 | 0.01 |
| Env 1 y=1,a=1 acc | Env 0 y=0,a=0 acc | 0.02 | 2.26 | 0.05 | 0.10 | 0.01 |
| Env 1 y=0,a=0 acc | Env 0 y=0,a=1 acc | 0.54 | -0.49 | 0.39 | 0.00 | 0.04 |
| Env 1 y=0,a=1 acc | Env 0 y=0,a=1 acc | 0.73 | 0.23 | 0.88 | 0.00 | 0.01 |
| Env 1 y=1,a=0 acc | Env 0 y=0,a=1 acc | -0.13 | 0.61 | -0.23 | 0.00 | 0.02 |
| Env 1 y=1,a=1 acc | Env 0 y=0,a=1 acc | -0.50 | 1.25 | -0.69 | 0.00 | 0.02 |
| Env 1 y=0,a=0 acc | Env 0 y=1,a=0 acc | -0.33 | 0.50 | -0.18 | 0.00 | 0.05 |
| Env 1 y=0,a=1 acc | Env 0 y=1,a=0 acc | -0.40 | -0.01 | -0.32 | 0.00 | 0.04 |
| Env 1 y=1,a=0 acc | Env 0 y=1,a=0 acc | 0.61 | 0.17 | 0.84 | 0.00 | 0.01 |
| Env 1 y=1,a=1 acc | Env 0 y=1,a=0 acc | 0.88 | -1.21 | 0.83 | 0.00 | 0.02 |
| Env 1 y=0,a=0 acc | Env 0 y=1,a=1 acc | 0.05 | 1.08 | 0.03 | 0.38 | 0.05 |
| Env 1 y=0,a=1 acc | Env 0 y=1,a=1 acc | -0.50 | 1.50 | -0.47 | 0.00 | 0.03 |
| Env 1 y=1,a=0 acc | Env 0 y=1,a=1 acc | 0.44 | 1.44 | 0.60 | 0.00 | 0.02 |
| Env 1 y=1,a=1 acc | Env 0 y=1,a=1 acc | 0.94 | 0.13 | 0.96 | 0.00 | 0.01 |

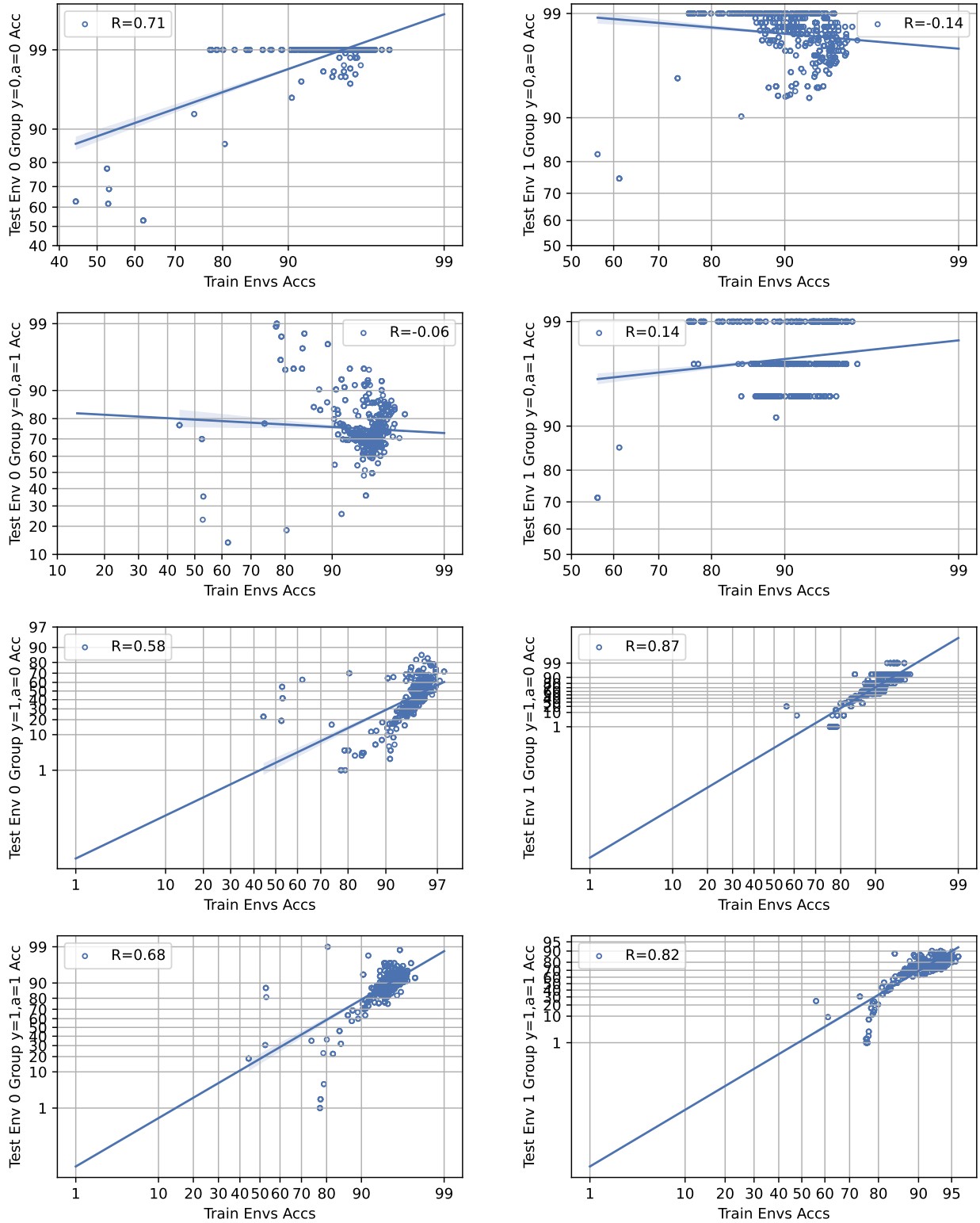

Figure 29: Waterbirds average ID vs. group-specific OOD accuracies

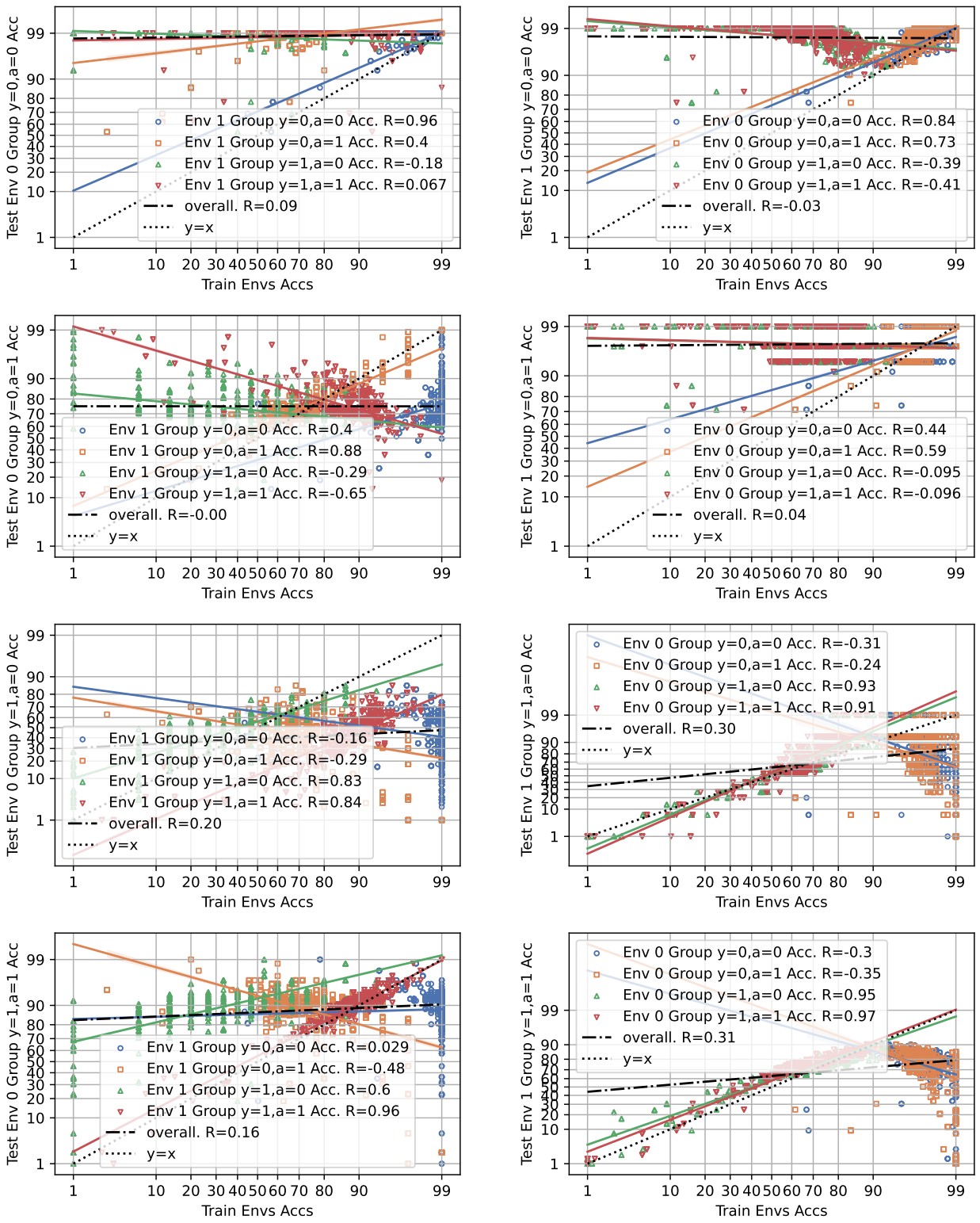

Figure 30: Waterbirds pairwise group-specific accuracies

### C.10 CivilComments

**CivilComments (Koh et al., 2021)** A dataset comprised of multiple subpopulations, which correspond to different demographic identities. The domain $d$ is a multi-dimensional binary vector with 8 dimensions, each corresponding to whether a given comment mentions one of the 8 demographic identities: *male, female, LGBTQ, Christian, Muslim, other religions, Black*, and *White*. For each of the 8 identities, we form 2 groups corresponding to the toxicity label, generating a total of 16 groups (e.g., male_toxic, male_non-toxic, etc.). By construction, since each comment may belong to more than one group, this experimental procedure differs slightly from the standard subpopulation shift framework discussed in this work. Regardless, the experimental results with and without group overlaps display similar accuracy on the line and accuracy on the inverse line patterns.

**Experimental Details.** We leverage both BERT and DistilBERT architectures for the CivilComments dataset.

**Discussion.** The CivilComments dataset exhibits both an accuracy on a line and an accuracy on the inverse line phenomenon across all test environments in the Leave-One-Domain-Out procedure. The spurious correlation, which involves the presence or absence of 8 different demographic identities (e.g., male, white, Christian, Muslim, etc.), is rather strongly correlated with the toxicity label. Empirically, the correlation coefficient ($R$) values range from as low as -0.47 to as high as 0.43. There appear to be many ID/OOD splits that can be derived from this dataset for benchmarking domain generalization.

Table 27: CivilComments ID vs. OOD properties.

| OOD | slope | intercept | Pearson R | p-value | standard error |
|---|---|---|---|---|---|
| Env 0 acc | 0.36 | -0.29 | 0.28 | 0.00 | 0.04 |
| Env 1 acc | -0.49 | 0.16 | -0.47 | 0.00 | 0.03 |
| Env 2 acc | 0.39 | -0.17 | 0.28 | 0.00 | 0.04 |
| Env 3 acc | -0.49 | 0.12 | -0.42 | 0.00 | 0.03 |
| Env 4 acc | 0.58 | 0.20 | 0.43 | 0.00 | 0.04 |
| Env 5 acc | -0.54 | 0.00 | -0.43 | 0.00 | 0.04 |
| Env 6 acc | 0.46 | -0.04 | 0.30 | 0.00 | 0.05 |
| Env 7 acc | -0.50 | 0.16 | -0.43 | 0.00 | 0.03 |
| Env 8 acc | 0.49 | 0.03 | 0.36 | 0.00 | 0.04 |
| Env 9 acc | -0.49 | 0.06 | -0.41 | 0.00 | 0.03 |
| Env 10 acc | 0.53 | 0.17 | 0.34 | 0.00 | 0.05 |
| Env 11 acc | -0.57 | -0.09 | -0.46 | 0.00 | 0.03 |
| Env 12 acc | 0.46 | 0.05 | 0.30 | 0.00 | 0.05 |
| Env 13 acc | -0.54 | -0.04 | -0.45 | 0.00 | 0.03 |
| Env 14 acc | 0.41 | -0.10 | 0.29 | 0.00 | 0.04 |
| Env 15 acc | -0.52 | 0.06 | -0.43 | 0.00 | 0.04 |

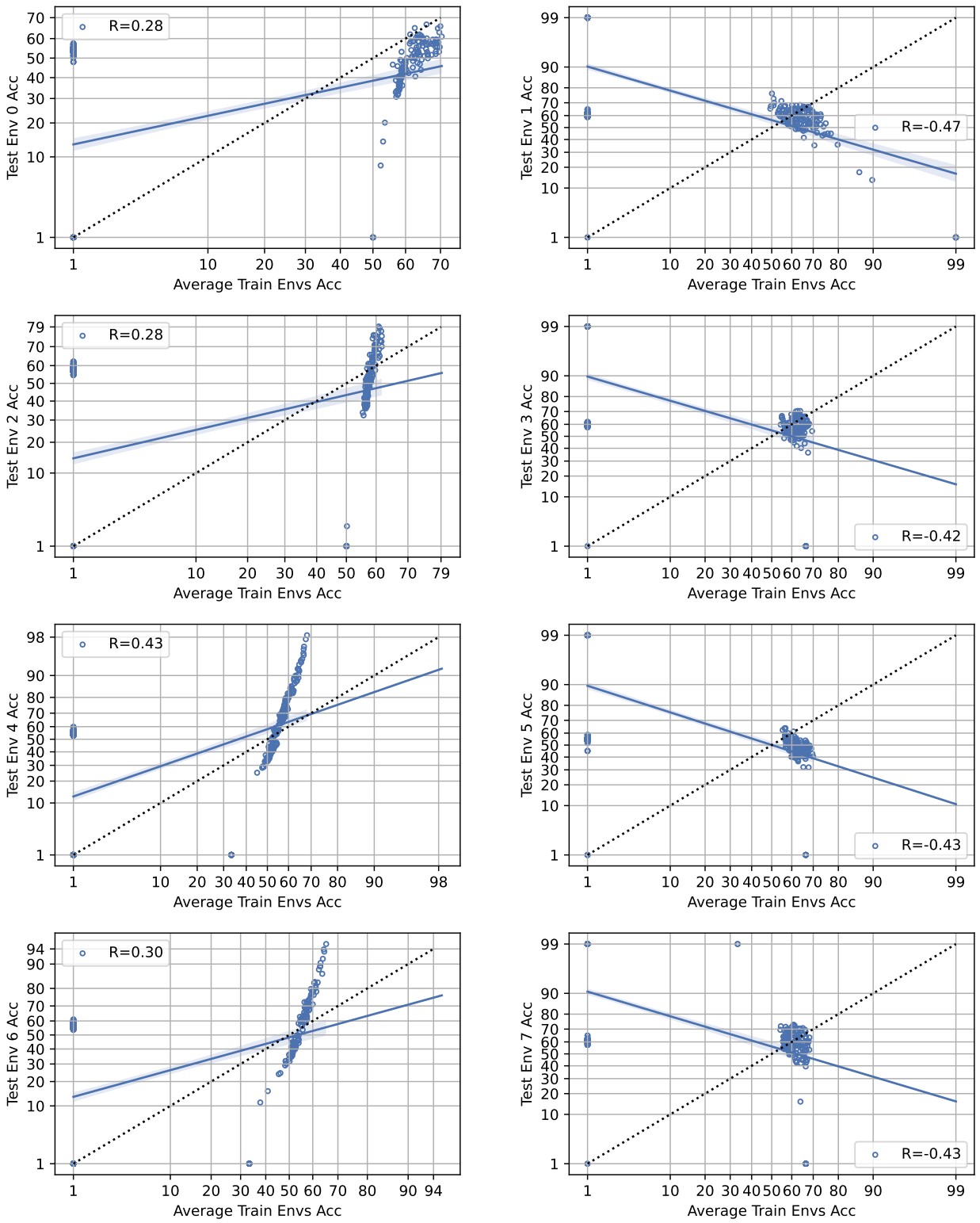

Figure 31: CivilComments: Average train Env Accuracy vs. Test Env Accuracy.

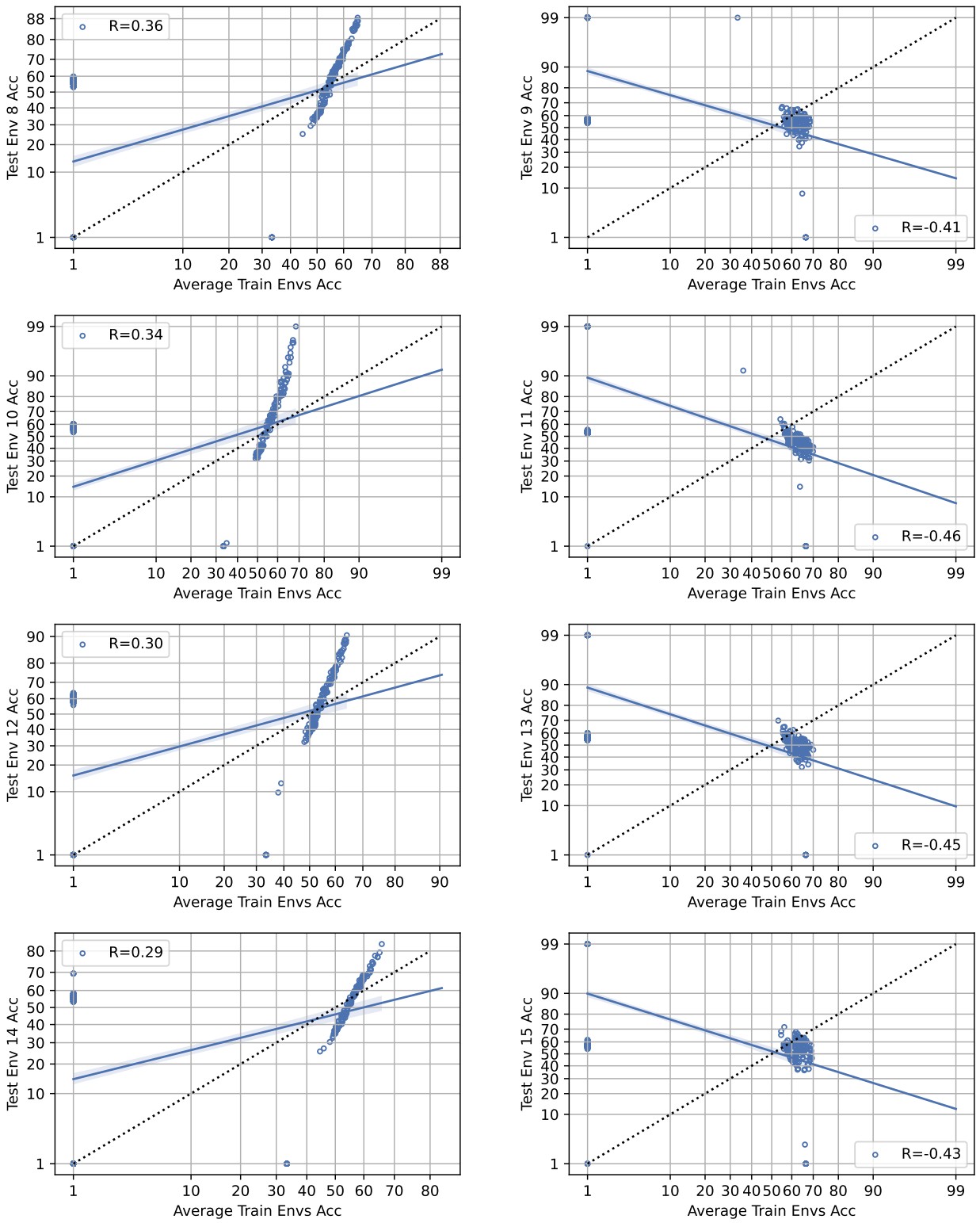

Figure 32: CivilComments: Average train Env Accuracy vs. Test Env Accuracy.

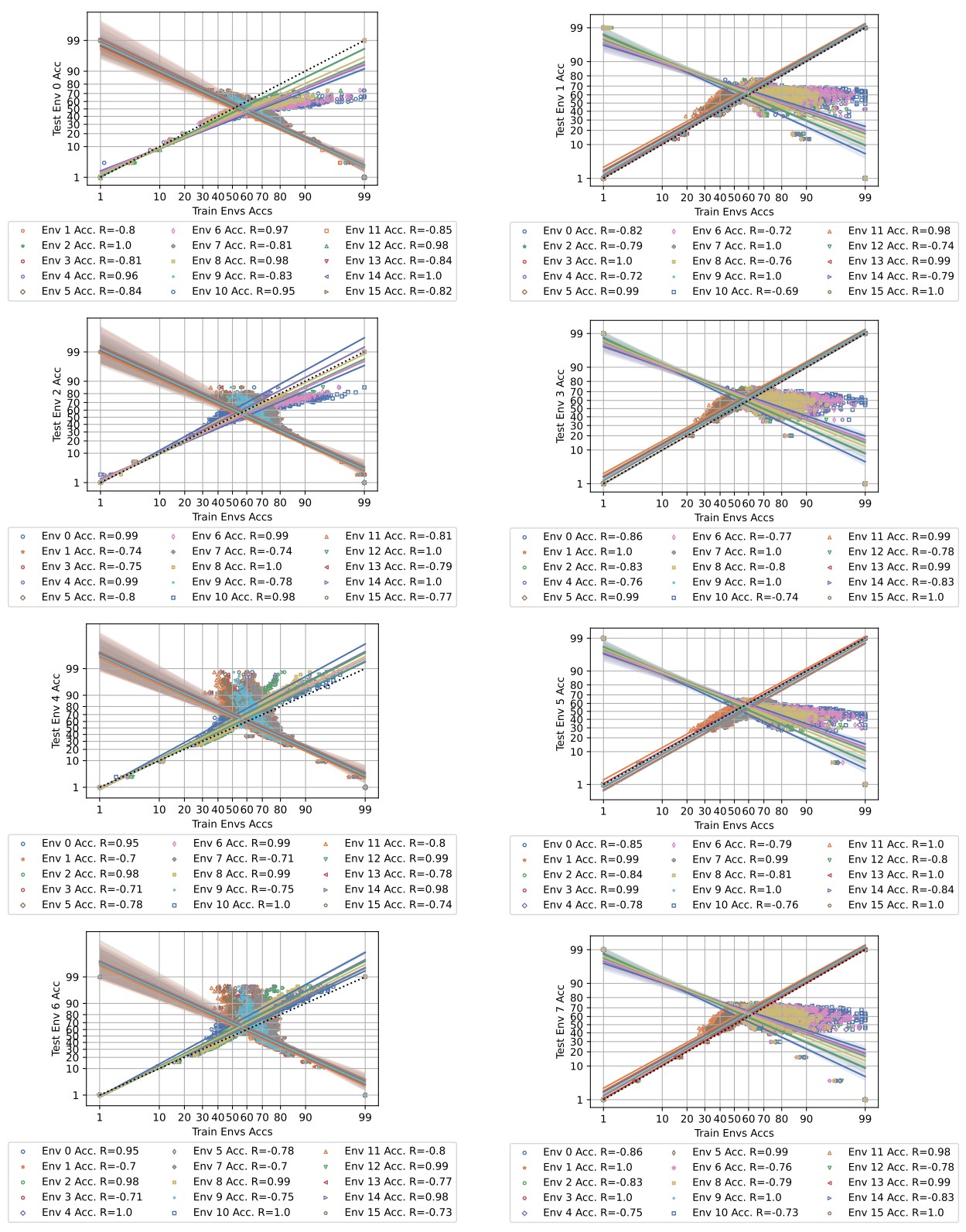

Figure 33: CivilComments: Train Env Accuracy vs. Test Env Accuracy.

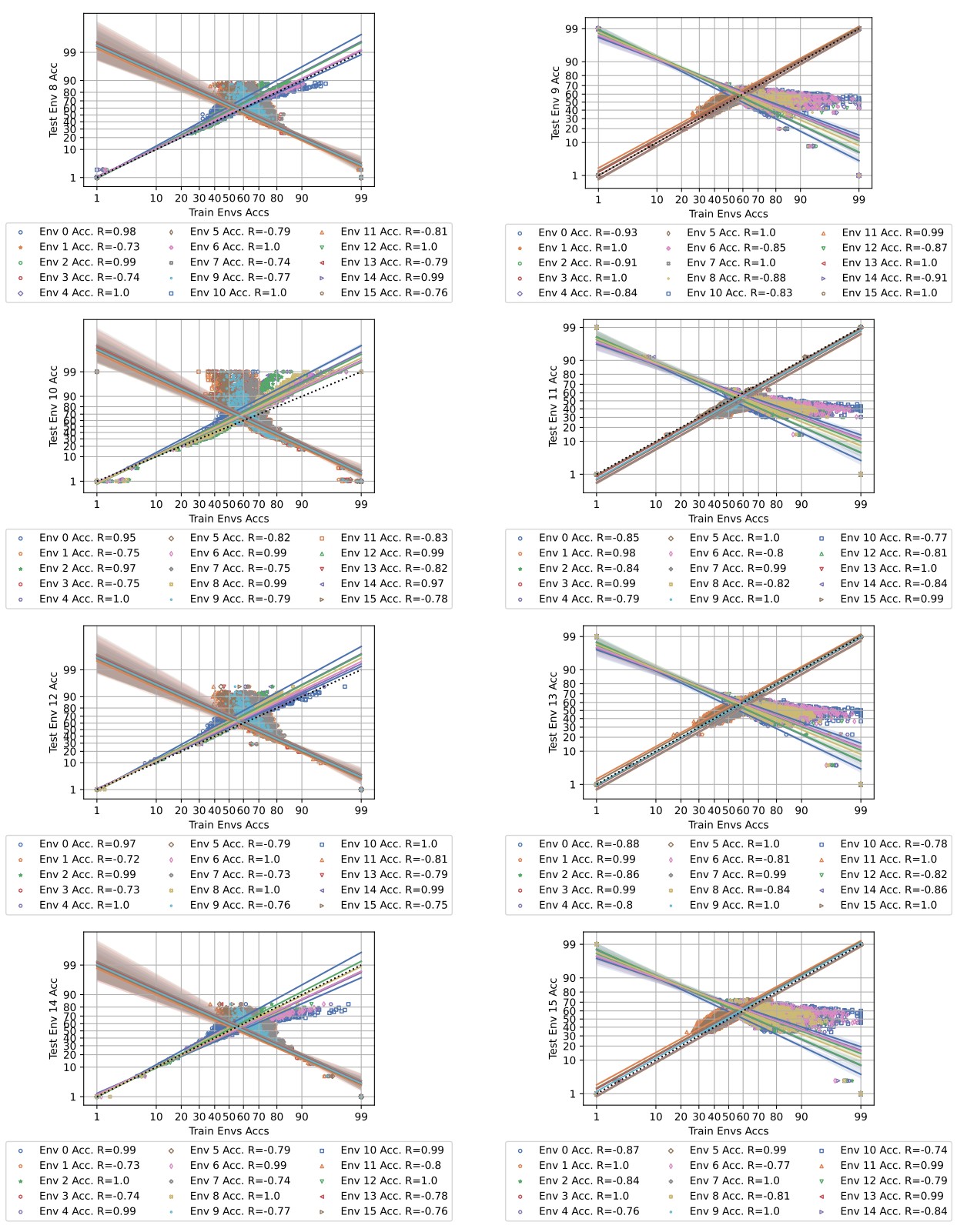

Figure 34: CivilComments: Train Env Accuracy vs. Test Env Accuracy.

## Acknowledgements

OS was partly supported by the UIUC Beckman Institute Graduate Research Fellowship, NSF-NRT 1735252, GEM Associate Fellowship, and the Alfred P. Sloan MPhD Program. SK acknowledges support from NSF 2046795 and 2205329, the MacArthur Foundation, Stanford HAI, and Google Inc. We thank A. Anas Chentouf, Tyler LaBonte, Vivian Nastl, Haoran Zhang, and Yibo Zhang for their comments on an earlier draft. We also thank the TMLR reviewers and the action editor for their valuable feedback and suggestions, which helped improve this work.

