# OpenReview forum: "Are Domain Generalization Benchmarks with Accuracy on the Line Misspecified?"
_TMLR — Accepted by TMLR_

### Review · Reviewer_ULfo · 2025-04-24

**Summary Of Contributions:**

This paper considers the problem of evaluation in domain generalization. Motivated by the observation that ERM achieves the best OOD accuracy in many benchmarks, it studies when the best domain generalizable model achieves the best OOD accuracy and the behavior of the OOD accuracy relative to the in-distribution (ID) accuracy. It shows that when the OOD spurious correlations are sufficiently misaligned with the ID spurious correlations, the best domain generalizable model achieves the best OOD accuracy and the relationship between the OOD accuracy and the ID accuracy is non-linear with high probability. Therefore, the paper proposes to use the correlation between the OOD and ID accuracies as a test for whether a domain generalization benchmark is well-specified. Experiments on some common neural networks and datasets indicate that in general, semisynthetic (like colored MNIST) and subpopulation shift datasets are well-specified.

**Audience:**

Yes

**Broader Impact Concerns:**

No broader impact statement is necessary.

**Claims And Evidence:**

Yes

**Requested Changes:**

Critical:
- Please provide some guidance on how many and what type of models need to be evaluated to carry out the test on the correlation between the OOD and ID accuracies.
- Can you expand on why the assumption on the disentanglement of the domain-general and spurious features is reasonable? (Last full paragraph of page 2.)

Minor:
- What are the obstacles to obtaining sufficient theoretical guarantees?

**Strengths And Weaknesses:**

Strengths:
- The paper contains theoretical guarantees and experiments.
- The problem is well-motivated. Domain generalization and the accurate evaluation of models are crucial for real world application.
- The paper is clear about the limitations of the proposed test and recommends a holistic and well-hedged approach for real world problems.

Weaknesses:
- The theoretical results are necessary, but not sufficient. Ideally, they should be sufficient as it allows conclusions to be drawn with more confidence. Currently, the results allow us to reject the hypothesis that the dataset is well-specified if the correlation between the OOD and ID accuracies is sufficiently high. However, this implies that some datasets may be prematurely discarded.
- It is unclear in practice how many and what type of models are needed to be evaluated to obtain the correlation between the OOD and ID accuracies. Are the ones used in the experiments sufficient?
- The theory assumes that the domain-general and the spurious features are disentangled and combine linearly to get the target variable. I feel that this disentanglement is a strong assumption, and the authors do not go into detail why it is reasonable. The linearity assumption seems reasonable given results on the neural tangent kernel, for example.

---

> ### Author Response · Authors · 2025-05-29
> **Thanks for your review (ULfo); we address your concerns here.**
>
> We thank the reviewer for the helpful feedback. We are pleased that the reviewer found our work to be well-motivated and crucial for real-world applications, although there are some questions about the theoretical analysis, which we address below. We group our replies by the requested changes.
>
> `“Please provide some guidance on how many and what type of models need to be evaluated to carry out the test on the correlation between the OOD and ID accuracies.”`
>
> We expand previous work’s evaluation of a diverse set of independently-trained models, varying architecture, random seed, data order, and hyperparameters, to avoid a single-model bias. In practice, we adopt the heuristic: keep adding models until the ID-OOD correlation changes by < 1%. On all benchmarks in Table 4, this threshold was reached well before we exhausted our model pool, and we still trained orders of magnitude more models than prior studies (see Table 2 for exact counts). This conservative choice ensures the correlation estimate is stable. We discuss this in the paragraph "Selecting Number of Models".
>
> `“Can you expand on why the assumption on the disentanglement of the domain-general and spurious features is reasonable? (Last full paragraph of page 2.)”`
>
> We have updated our results to accommodate nonlinear shifts in the spurious correlations.
>
> Regarding disentanglement, this assumption is widely used in the literature (cited in original submission), with growing empirical support. For example, several studies show that finetuning a single linear layer can recover performance lost to spurious correlations (Rosenfeld et al., 2022). Additional work on last-layer retraining likewise boosts subgroup accuracy (Liu et al., 2021; LaBonte et al., 2023). These findings suggest that the non-spurious and spurious representations learned by deep models are often linearly separable, aligning with our assumption.
>
> We acknowledge, however, that this remains a strong assumption backed primarily by empirical evidence. We state this caveat explicitly for transparency.
>
> `“What are the obstacles to obtaining sufficient theoretical guarantees?”`
>
> To get necessary and sufficient conditions, we need a lower bound on P(\we^T\Ze^OOD \ge 0). Such a lower bound does not exist except with stronger distributional assumptions on symmetry. We have added Theorem 2 in the revision to have necessary and sufficient results with the assumption of symmetry; thanks for the suggestion. Some distributions with this assumption are Gaussians, local-scale families, two-sided sub-Gaussians, etc. Notably, this class is more restrictive than those that give only sufficient but not necessary conditions (most random variables with bounded moments). But, including this in the result is more complete. Thank you for this feedback.

---

> > ### Comment · Reviewer_ULfo · 2025-06-07
> >
> > Thanks to the authors for your response. I have no further questions.

---

### Review · Reviewer_jq8n · 2025-05-05

**Summary Of Contributions:**

This paper examines the limitations of current domain generalization (DG) benchmarks, specifically on a phenomenon where in-distribution (ID) accuracy strongly correlates with out-of-distribution (OOD) accuracy.

The authors argue that these benchmarks may be misspecified for evaluating the true robustness of models to spurious correlations, which are statistical associations that do not generalize across domains.

Through both theoretical analysis and empirical experiments, the paper introduces the concept of well-specified benchmarks and shows that many popular DG benchmarks fail to capture meaningful distribution shifts.

The results suggest that for robust evaluation, benchmarks should not show a strong positive correlation between ID and OOD accuracy.

**Audience:**

Yes

**Claims And Evidence:**

No

**Requested Changes:**

-  What if the correlation does not follow this assumption, which can be common in many real-world scenarios? More discussion and analysis on this aspect is needed.

- Does the property and conclusion of this paper hold on some larger-scale domain generalization settings, such as between ImageNet and ImageNet-R, ImageNet-A, ImageNet-C, or other larger-scale domain generalization dataset?

- More in-depth discussion on how the findings can influence the development of new DG algorithms or the improvement of existing ones is needed.

- More feature space visualization on the different types of features from a model on ID and ODD scenarios is needed, such as t-SNE, to help the understanding of the relationship.

- An exploration on alternative evaluation metrics may provide a better understanding of model robustness to spurious correlations.

- Please provide a detailed comparison with other recent domain generalization methods that specifically address spurious correlation removal.

- Please include an ablation study to assess how the accuracy on the line phenomenon varies under different configurations of domain splits and model types.

- The paper introduces the concept of well-specified benchmarks, but it does not provide sufficient guidelines for practitioners to easily create or adapt their own datasets according to this framework. A more detailed toolkit or practical advice on how to curate well-specified benchmarks would be valuable for future research.

- Is table 3 necessary in the main text? Or can be put into the supplementary material?

**Strengths And Weaknesses:**

Strength:

+ The theoretical framework is solid.

+ The theoretical framework developed in the paper offers valuable insights.

+ The paper proposes a new framework for assessing DG benchmarks and provides guidelines for constructing more effective evaluation protocols for domain generalization tasks.

+ Introduces the concept of well-specified benchmarks.

Weakness:

-  One of the major concerns is that, the spurious correlations follow the Gaussian assumption. What if the correlation does not follow this assumption, which can be common in many real-world scenarios? More discussion and analysis on this aspect is needed.

- Does the property and conclusion of this paper hold on some larger-scale domain generalization settings, such as between ImageNet and ImageNet-R, ImageNet-A, ImageNet-C, or other larger-scale domain generalization dataset?

- More in-depth discussion on how the findings can influence the development of new DG algorithms or the improvement of existing ones is needed.

- More feature space visualization on the different types of features from a model on ID and ODD scenarios is needed, such as t-SNE, to help the understanding of the relationship.

- An exploration on alternative evaluation metrics may provide a better understanding of model robustness to spurious correlations. Metrics such as area under the ROC curve (AUC), average worst-case performance, or robustness to adversarial perturbations could be used in conjunction to offer a more nuanced assessment.

- The paper does a good job of critiquing existing benchmarks, but it does not provide a detailed comparison with other recent domain generalization methods that specifically address spurious correlation removal. A more comprehensive comparison would position the paper’s contributions within the broader DG research landscape.

- The paper does not include an ablation study to assess how the accuracy on the line phenomenon varies under different configurations of domain splits and model types. Such an experiment would help validate the framework's robustness and further illustrate its practical utility.

- The paper introduces the concept of well-specified benchmarks, but it does not provide sufficient guidelines for practitioners to easily create or adapt their own datasets according to this framework. A more detailed toolkit or practical advice on how to curate well-specified benchmarks would be valuable for future research.

- Is table 3 necessary in the main text? Or can be put into the supplementary material?

---

> ### Author Response · Authors · 2025-05-29
> **Thanks for your review (jq8n); we address your concerns here.**
>
> We thank the reviewer for the helpful feedback. We are pleased that the reviewer found our theoretical analysis to be solid, although there was a misunderstanding on the restrictiveness of assumptions and requests for additional empirical analysis, which we address below. We group our replies by the requested changes.
>
> `“What if the correlation does not follow this assumption, which can be common in many real-world scenarios?.”`
>
> Our theorems do not require Gaussianity. Importantly, our results are relatively general (see Remark 1). Our first result is sufficient with the sub-Gaussian assumptions to give explicit conditions; however, random variables with bounded moments or other distribution assumptions will also give similar results with different constants. Hence, the implications of these results hold quite generally, and the original submission addresses this concern. We added theorem 2 with necessary and sufficient conditions, which require symmetric random variables (Gaussians are one example, but not the only example).
>
> `“...ImageNet and ImageNet-R, ImageNet-A, ImageNet-C, or other larger-scale domain generalization dataset…”`
>
> We note that ImageNet-R/A/C and similar “ImageNet-variant” are mostly corruptions in X that leave the class label Y unchanged. T the DG setting we analyze focuses on shifts where X-Y correlations (i.e., spurious correlations) differ across domains. DomainBed, WILDS, and the other datasets we analyze are the standard.
>
> Nevertheless, existing work has evaluated ImageNet → ImageNet-R/A/C and observed the same “accuracy-on-the-line” pattern (e.g., Miller et al., 2020; Taori et al., 2020). See revised Related Works section.
>
> `“...how the findings can influence the development of new DG algorithms or the improvement of existing ones is needed…”`
>
> Paragraph “Implications on Key Domain Generalization and Evaluation Practices.” in section 4.2 now highlights two directions, in addition to the directions originally in the submission. In summary, our work gives a test for when domain generalization is even needed and helps identify effective algorithms and practices.
>
> `“...feature space visualization...”`
>
> Please see Figure 4-5 and the supporting discussion in Appendix C.1 UMAP visualization, a more appropriate method in this setting than t-SNE. We find strong alignment between the embeddings in-distribution vs. out-of-distribution for datasets we find are misspecified, e.g., PACS. We see a strong misalignment in the datasets our analysis suggests are misspecified, e.g., Waterbirds, Spawrious, and ColoredMNIST. Thanks for the suggestion and please see Paragraph "Embeddings Analysis".
>
> `“An exploration on alternative evaluation metrics...”`
>
> Indeed, we already evaluate additional metrics. For datasets with group labels, we report both average accuracy and worst-group accuracy; see Appendix C9—we show that while the average ID-OOD accuracy exhibits a positive “accuracy-on-the-line” correlation, the worst-group accuracy can be weak or even negative.
>
> We agree that there are interesting questions regarding other metrics; we discuss related work on this topic in the Related Works section. We agree that a formal analysis of other metrics merits its own study in future work.
>
> `“...detailed comparison with other recent domain generalization methods...”`
>
> We believe the reviewer may have been referring to algorithmic baselines. However, our paper does not introduce a new domain-generalization method; it focuses solely on evaluating benchmarks. Consequently, a head-to-head comparison with spurious-correlation removal algorithms falls outside our scope, particularly since these results exist in a multitude of papers, and our primary findings are that few datasets satisfy our conditions. That said, we now add a brief discussion of algorithms originally evaluated and whether those benchmarks meet our well-specified criterion on paragraph "State-of-the-Art Algorithms"; we hope this clarifies the distinction.
>
> `“...ablation study to assess how the accuracy on the line phenomenon varies under different configurations of domain splits and model types.”`
>
> Our empirical study already enumerates all possible domain splits for every dataset (Table 2) and evaluates diverse architecture families. Additionally, the accuracy on the line result is a global property not tied to a specific model family.
>
> `“...advice on how to curate well-specified benchmarks...”`
>
> There are examples of developing these benchmarks in semi-synthetic settings. The main question is about fully ‘natural’ datasets. This requires some trial-and-error; our results offer an evidence-backed call-to-action for the community to advance evaluation science for developing these benchmarks. Our work gives some guiding principles and a test that the curation process for such benchmarks can be grounded on. Please see our revised text on “Constructing Benchmarks Without Positive Accuracy on the Line” in section 4.2.
>
> `“...table 3…”`
>
> Moved' thanks!

---

> > ### Comment · Reviewer_jq8n · 2025-06-07
> > **Re: Thanks for your review (jq8n); we address your concerns here.**
> >
> > Thanks the authors for the response letter, which not only addressed my concerns but also clarified my misunderstandings.
> >
> > Therefore, I believe it is good enough for publication.

---

### Review · Reviewer_svrV · 2025-05-21

**Summary Of Contributions:**

This work investigates the limitations of domain generalization benchmarks in evaluating model robustness to spurious correlations under distribution shifts. The authors argue that many existing benchmarks are misspecified, as they fail to include shifts in spurious correlations that meaningfully degrade out-of-distribution (OOD) generalization. To address this, they propose theoretical conditions for well-specified benchmarks, showing that a strong positive correlation between in-distribution (ID) and OOD accuracy—known as "accuracy on the line"—indicates that a benchmark is unsuitable for assessing robustness to spurious correlations. Through extensive analysis of over 40 ID/OOD splits across 12 popular benchmarks, they demonstrate that most existing benchmarks exhibit accuracy on the line, limiting their utility for domain generalization evaluation. The findings emphasize the need for rethinking benchmark design to better assess robustness to spurious correlations.

**Audience:**

Yes

**Claims And Evidence:**

Yes

**Requested Changes:**

1. Although the discussion is meaningful, it does not follow the real-world data generation process, limiting the impact. More specifically, if the assumption and the data model on the spurious features does not reflect the trends in real-world datasets such as those from DomainBed and Wilds, perhaps we need to reconsider better data models instead of the benchmarks. Otherwise, the gap between the methodology development in OOD generalization and the real-world practice will gradually lose the relevance.

2. It's also worth discussing the correlations in terms of other metrics. For example, mean accuracy and worst case accuracy will reflect different trends. It's also encouraged to consider other metrics in the Wilds benchmark.

3. A number of related works have also discussed this issue:
- Rich feature construction for the optimization generalization dilemma, ICML'22.
- Pareto invariant risk minimization, ICLR'23.
- Understanding and Improving Feature Learning for Out-of-Distribution Generalization, NeurIPS'23.
- Spurious Feature Diversification Improves Out-of-distribution Generalization, ICLR'24.

**Strengths And Weaknesses:**

## Strengths:

(+) This work focuses on a fundamental challenge in evaluating the OOD generalization benchmarks.

(+) The authors provide detailed theoretical discussion on the influence of spurious features, and how it's related to ID-OOD performance correlations.

(+) The experiments are quite extensive and comprehensive.

## Weakness

(-) Although the discussion is meaningful, it does not follow the real-world data generation process, limiting the impact.

(-) It's also worth discussing the correlations in terms of other metrics.

(-) A number of related works have also discussed this issue.

---

> ### Author Response · Authors · 2025-05-29
> **Thanks for your review (svrV); we address your concerns here.**
>
> We thank the reviewer for the helpful feedback and for recognizing the significance of studying the evaluation of robustness to spurious correlations, as well as acknowledging the comprehensiveness of our empirical results. We are also pleased that the reviewer found our work to be meaningful, although they expressed some concerns about its alignment with the real world, which we address, along with other concerns, below. We group our replies by the three requested changes.
>
> `“The assumption of the data model on the spurious features does not reflect trends in real-world datasets, …, perhaps we need to reconsider better data models instead of the benchmarks.”`
>
> Our analysis specifically aligns with real-world pathologies.
>
> We consider two classes of features:
> (i) domain-general signals that are stable across domains, and
> (ii) domain-specific (spurious) cues that can shift across domains.
>
> Real-world instances illustrate this distinction. Physiological signs in chest X-ray diagnosis are domain-general, whereas hospital-specific markings and text overlays are spurious; models latch onto those markings and fail to generalize OOD (Zech et al., 2018). A similar issue arises with scanner-specific artefacts in medical imaging, which vanish when hospitals upgrade to new machines (DeGrave et al., 2021). An example of a benchmark in this domain that may be mis-specified for these types of challenges is WILDSCamelyon (Section C.6). In recidivism prediction, demographic information can proxy race, spurious in that it correlates with historical policing bias rather than causal risk, leading to poor OOD performance (ProPublica, 2016). An example of a benchmark in this domain is CivilComments (well-specified), where demographic attributes are spurious correlates (Section C.10).
>
> Our analysis matches precisely this setting: spurious correlations can carry only a fraction of the predictive power of an in-distribution-optimal model, yet that fraction is enough to trigger failures when it shifts. Hence, our data model reflects genuine real-world trends. Many “real-world” benchmarks, however, do not exhibit such shifts; they show “accuracy-on-the-line” behavior and are thus misspecified for evaluating algorithms meant to combat the failures above---as our theoretical analysis demonstrates. We formalize conditions for well-specified benchmarks and show that datasets meeting these conditions are currently under-represented, highlighting a critical gap in today’s evaluation landscape.
>
> We have added these examples to the revision.
>
> `“It's also worth discussing the correlations in terms of other metrics.”`
>
> Indeed, we already evaluate additional metrics in the original submission. For datasets with group labels, we report both average accuracy and worst-group accuracy. In Waterbirds—see Section C.9—we show that while the average ID-OOD accuracy exhibits a positive “accuracy-on-the-line” correlation, the worst-group accuracy can be weak or even negative. Interpreting each subgroup as its own environment makes these findings consistent with the multi-environment results in our other datasets.
>
> We agree that there are interesting questions regarding other metrics; we also discuss related work with preliminary findings in the Related Works section of the original submission. We agree that a formal analysis of other metrics merits its own study in future work.
>
> `“A number of related works have also discussed this issue.”`
>
> We have cited the four papers suggested by the reviewer in our revision. However, we highlight that these four papers study a related but fundamentally different problem. They focus on algorithms for domain generalization. By clarifying where evaluation of this task itself breaks down, our work complements and strengthens the empirical claims of all four algorithmic papers. Key distinctions:
>
>
> RFC (Bonsai) alternates between discovery and synthesis episodes to construct rich features, boosting OOD performance across several methods, including on ColoredMNIST. PAIR reframes IRM as a multi-objective optimization problem, improving stability and accuracy on DomainBed. FeAT argues that ERM already finds invariant features but fails to retain them, and improves generalization by enforcing feature reuse. BANG demonstrates that diversifying spurious features in ensembles can outperform suppression, yielding gains on controlled datasets such as MultiColorMNIST.
>
> While each method proposes algorithmic solutions to improve robustness, our work asks a prerequisite question: are the benchmarks even capable of exposing spurious correlation failures in the first place? This is a core limitation in aligning progress in a generalizable way. Notably, papers like Zhang et al., 2022 focus on benchmarks that we also find to be well-specified, e.g., ColoredMNIST.
>
> We believe the new experiments, clarifications, and citations fully address the reviewer’s concerns and further strengthen the practical relevance of our study.

---

> > ### Comment · Reviewer_svrV · 2025-06-07
> >
> > Thanks the authors for the detailed responses to my comments, which addressed most of my concerns.
> >
> > Certainly, there exist spurious features in the real-world scenarios, however, evaluating the OOD generalization methods may not necessarily be focused on the removal of spurious features, as the additional regularization imposed by the OOD generalization methods will also affect the learning of domain-general features.
> >
> > In other words, although a number (or arguably most) of OOD generalization benchmarks demonstrate "accuracy-on-the-line" phenomenon, it does not invalidate their usefulness in the evaluation, as the setup in those benchmarks (e.g., those from wilds benchmark) aligns with the real-world use of ML methods.
> >
> > I believe the aforementioned point should be carefully addressed in the discussion of this work. Thanks!

---

> ### Author Response · Authors · 2025-06-07
> **Addressing the utility of benchmarks with accuracy on the line**
>
> We thank Reviewer svrV for their thoughtful engagement. We are happy to clarify this point further—this discussion already appears in Section 4.2. There, we explicitly note that benchmarks such as DomainBed and WILDS capture important “natural shifts” that are valuable to evaluate, even though they may not be well-suited for assessing robustness to spurious correlations. We agree these benchmarks are useful; our contribution is to identify a subset of settings where such benchmarks are provably misspecified for evaluating robustness to spurious correlations. These settings remain underexplored, yet state-of-the-art methods often rely on these benchmarks, which can lead to misleading conclusions.
>
> We also emphasize that dataset availability reflects selection bias. The frequency and importance of real-world scenarios where spurious correlations degrade OOD performance—particularly in high-stakes domains like healthcare, policing, and finance (see previously cited examples)—cannot be dismissed simply because they are underrepresented in public benchmarks. In these domains, spurious correlations often stem from historical biases and can lead to harmful outcomes if left unmitigated. However, privacy constraints frequently limit the release of relevant datasets.
>
> Finally, we appreciate the implicit reminder that our current draft could better acknowledge the significant contributions of DomainBed, WILDS, and related efforts. We will revise the paper to more clearly highlight their importance.
>
> We hope this clarifies our position and addresses your concerns.

---

> > ### Comment · Reviewer_svrV · 2025-06-07
> >
> > Thank the authors for the follow-up response! It clarifies my last concern!

---

### Decision · Action_Editor_pfWi · 2025-07-02

**Recommendation:** Accept as is

**Audience:**

Yes

**Audience Explanation:**

The paper directly addresses a critical issue in the machine learning community—how to meaningfully evaluate robustness to distribution shifts, especially spurious correlations. As domain generalization remains a key research area, this work will be of strong interest to researchers working on domain generalization and benchmark design. Moreover, the diagnostic test proposed for benchmark validity and the criteria for well-specified benchmarks are practical contributions that will benefit the audience beyond domain generalization.

**Claims And Evidence:**

Yes

**Claims Explanation:**

The authors have provided both theoretical and empirical evidence supporting their central claim: that many widely used domain generalization (DG) benchmarks are misspecified for evaluating robustness to spurious correlations. The theoretical framework is rigorously developed and now includes necessary and sufficient conditions. Empirical results span over 40 ID/OOD splits and diverse model configurations, providing robust support for the observed "accuracy-on-the-line" phenomenon. Furthermore, the added qualitative analyses (e.g., embedding visualizations) and expanded discussions strengthen the credibility of the findings. Reviewer concerns were thoroughly addressed in a transparent and technically sound manner. At the end of the discussion, all reviewers unanimously recommend acceptance.